# COExpander: Adaptive Solution Expansion for Combinatorial Optimization

**Jiale Ma** [* 1 2]   **Wenzheng Pan** [* 1]   **Yang Li** [1 2]   **Junchi Yan** [1 2]

## Abstract

Despite rapid progress in neural combinatorial optimization (NCO) for solving CO problems (COPs), as the problem scale grows, several bottlenecks persist: 1) solvers of the Global Prediction (GP) paradigm struggle in long-range decisions where the overly smooth intermediate heatmaps impede effective decoding, and 2) solvers of the Local Construction (LC) paradigm are time-consuming and incapable of tackling large instances due to the onerous auto-regressive process. Observing these challenges, we propose a new paradigm named Adaptive Expansion (AE) with its instantiation COExpander, positioned to leverage both advantages of GP and LC. COExpander utilizes informative heatmaps generated by a global predictor, which is learned under the guidance of locally determined partial solutions, to in turn direct the expansion of determined decision variables with adaptive step-sizes. To ensure transparent evaluation, we further take the lead to canonicalize 29 benchmarks spanning 6 popular COPs (MIS, MCl, MVC, MCut, TSP, ATSP) and various scales (50-10K nodes), upon which experiments demonstrate concrete SOTA performance of COExpander over these tasks.

## 1. Introduction

Combinatorial optimization problems (COPs) possess extensive applications in diverse fields including logistics (Wang & Tang, 2021), transportation (Baty et al., 2024), supply chain management (Singh & Rizwanullah, 2022), and network design (Paschos, 2014), in search of optimal solutions within discrete states. However, due to the inherent NP-hardness for most (especially large-scale) COPs, approach-

ing (near) optimal solutions within reasonable time remains a daunting challenge. Over the past decade, the neural combinatorial optimization (NCO) community (Zhang et al., 2023; Guo et al., 2023; Wang et al., 2024) has fostered copious work to explore the potential of data-driven methods for CO solvers and have shown considerable promise in terms of both effectiveness and efficiency of problem solving.

Learning-based solvers for CO typically employ neural networks to generate neural predictions for either solution construction or search guidance. Given that divide-and-conquer (D&C) frameworks (Ye et al., 2024b; Zheng et al., 2024) and solution optimization approaches (Chen & Tian, 2019; Ma et al., 2023) are theoretically orthogonal to and practically applicable over the main solving stage, in this paper, we focus on the core part of NCO, i.e., the acquisition of solutions for COPs, and categorize the mainstream neural CO solvers into two paradigms[1] based on the way the solutions are constructed. As the first type, **Local Construction (LC)** methods (Kool et al. (2018); Berto et al. (2024); Drakulic et al. (2023); Pan et al. (2025), etc.) cast COPs as Markov Decision Processes (MDPs) and the neural networks are trained to predict the best next-step action given the current state. This process continues until a complete solution is assembled, as depicted in Fig. 4 (a). While LC solvers effectively decompose the solving process into the step-by-step construction to leverage local states for the next decision, they suffer from a lack of global perspective and encounter scalability problems due to the laborious and myopic auto-regression. On the other hand, **Global Prediction (GP)** approaches (Joshi et al. (2019); Qiu et al. (2022); Sun & Yang (2023); Li et al. (2023; 2024; 2025), etc.) use neural networks to globally predict the likelihood of each node or edge being selected, and then decode the probability heatmaps (via heuristics as simple as greedy) to obtain solutions, as shown in Fig. 4 (b). This paradigm provides global guidance mapped from the entire graph structure for constructing the solution. However, as problem scales increase, the predicted heatmap through one-shot inference tends to be overly smooth (with many noisily similar values), hindering its guiding effect for the subsequent decoding procedure. We delineate these observations in Appendix B.2 and B.3.

The issues of these two paradigms can be generalized as

---

[*]Equal contribution  [1]Sch. of Artificial Intelligence & Sch. of Computer Science, Shanghai Jiao Tong University [2]Shanghai Innovation Institute. Correspondence to: Junchi Yan <yanjunchi@sjtu.edu.cn>.
 This work is partly supported by NSFC (62222607, 623B1009). Code available at github repository.

*Proceedings of the $42^{nd}$ International Conference on Machine Learning*, Vancouver, Canada. PMLR 267, 2025. Copyright 2025 by the author(s).

[1]A *paradigm* is detached from any particular *model* or *solver*.

opposing extremes in the decision granularity, i.e., the quantity of variables determined[2] at each step, where GP leans entirely on global prediction while LC depends solely on local information for problem-solving. We note that diffusion- and consistency-based solvers (Sun & Yang, 2023; Li et al., 2023; 2024) constitute a distinct subset of GP solvers, endeavoring to generate globally complete solutions through a sequential denoising process. However, they still fail to ensure the intermediate steps interpretably direct the determination of variables, for the ultimate solution is unrevealed until the final stage. Hence, a promising direction is to explore a principled mechanism to adaptively control the decision granularity in each forwarding step, replacing the current practice of using a fixed step size, e.g., resolving variables one at a time or updating them all at once.

**Definition 1.1** (Variable Determination Process). Given an optimization problem with $N$ decision variables, we define Variable Determination Process $VDP(N, k)$ as an iterative procedure transitioning from complete uncertainty to full determination in $k$ steps. Let $d_0, d_1, \ldots, d_{k-1}$ be the number of determined variables at each step, which satisfies $d_0 < d_1 < \cdots < d_{k-1}$, with $d_0 = 0$ and $d_{k-1} = N$.

To fulfill the above motivation, we first define a new paradigm for NCO solving, namely **Adaptive Expansion (AE)**. Paralleling GP and LC, AE encompasses a model-agnostic training and solving pipeline that orients at fully utilizing the intermediate state information during problem solving while maintaining an adaptively controllable granularity regarding the step-size of the decisive predictions produced by the neural networks. Further, we devise a novel neural solver **COExpander** as a forerunner instantiation of AE, where informative heatmaps are generated by a global predictor learned under the guidance of locally determined partial solutions, and in turn, guide an adaptive process to expand the set of determined decision variables. Specifically, as sketched in Fig. 1, embedded masks representing random intermediate solving states are fed as a prompt input along basic graph features to train COExpander to predict the full distribution under partial guidance. At solving stage, iterative determination processes are performed, and in each round only the most convincing components of the output heatmap is accredited to determine *arbitrary* number of decision variables and update the neural masks denoting current partial solution state that prompts the next iteration.

In Def. 1.2, we provide a formal formulation of AE as well as the comparison with the other two paradigms. Notably, AE combines the merits of both and rectifies their individual shortcomings by incorporating guidance derived from both global and local perspectives. On the one hand, compared to GP solvers, COExpander evolves from the previous "noise-

---

[2]Following Ahn et al. (2020), we regard the solving of a COP as the determination process of decision variables, see Def. 1.1.

**Definition 1.2** (Solving Paradigms). Given $VDP(N, k)$, we defined the following paradigms based on $k$ and $N$:
· When $k = 1$, all decision variables are determined in a single step, this process is defined as *Global Prediction*.
· When $k = N$, each step determines exactly one variable, this process is defined as *Local Construction*.
· When $1 < k < N$, the process is defined as *Expansion*. Further, if the difference between consecutive steps is not constant, the process is referred to as *Adaptive Expansion*.

to-full" to an adaptive "partial-to-full" manner of neural mapping. Instead of predicting full solutions from probabilistic noise, the "partial-to-full" approach starts from a state with several decision variables already determined, effectively narrowing the search space. The reasoning process, thus guided, attentively leverages the most informative portion of the inferential output of the neural model in each determination round, thereby leading to more stable and consistent predictions. On the other hand, in contrast to LC solvers, COExpander evolves from the fixed "next-token" to a flexible "next-few-token" manner of solution construction and incorporates global guidance. This progress chiefly stems from the supervised global predictor in COExpander which furnishes more comprehensive information in a single inference than LC solvers, i.e., providing the probabilistic estimation of all decision variables rather than merely that of the next node to be selected. This augments both efficiency and scalability while sustaining the quality of solutions. We highlight our main contributions as follows:

- We propose the Adaptive Expansion (AE) paradigm and the COExpander solver for COPs. It bridges the global prediction (GP) and local construction (LC) paradigms via utilization of a partial state prompted solution generator, replacing the rigid decision with adaptive step sizes beyond auto-regression and one-shot solving.

- To establish a unified protocol for performance evaluating and reporting for COPs, we re-wrap 5 non-learning baseline solvers (Appendix D) and re-cononicalize 29 standard datasets (Appendix E), providing a just and generalizable benchmark for 6 commonly studied COPs.

- Compared with previous neural state-of-the-arts (SOTA), COExpander has reduced the average optimality drop on 6 COPs from **3.807%** to **0.657%**, with a speedup of **4.0x**. Besides, COExpander shows good cross-scale and cross-distribution generalization ability on real-world and ultra-large instances within each problem type.

## 2. Related Work

**GP Solvers.** Different ways of learning have been applied to generate effective heatmaps globally, including supervised learning (SL) (Joshi et al., 2019; Fu et al., 2021), unsupervised learning (UL) (Wang & Li, 2023; Min et al., 2024), and meta-reinforced learning (RL) like DIMES (Qiu et al.,

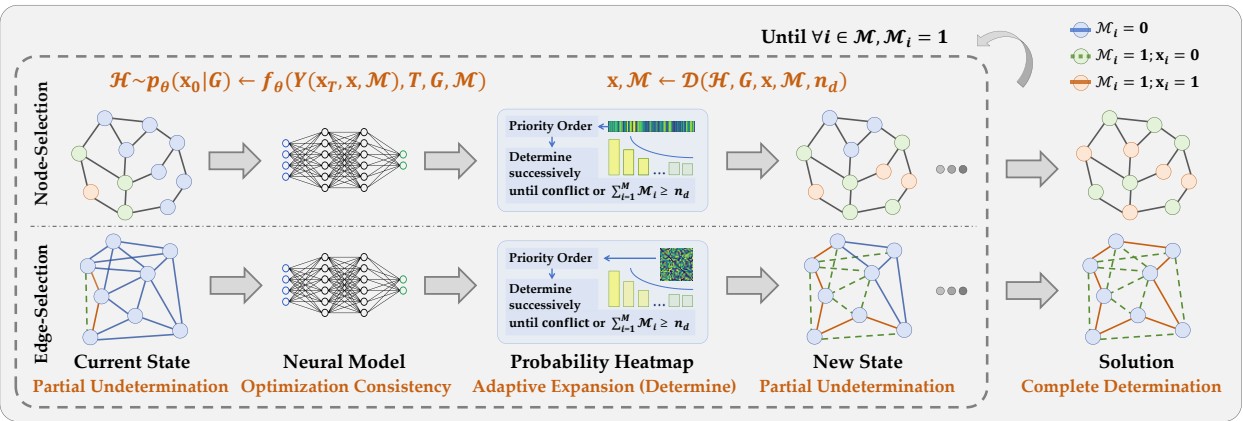

*Figure 1.* General pipeline of COExpander under AE. $\mathbf{x}$: decision variables; $\mathcal{M}$: mask; $\mathbf{y}$: partial solution; $G$: graph data; $\theta$: model parameters; $\mathcal{H}$: probability heatmap; $\mathcal{D}$: determination operator. A comparative illustration of the 3 pipelined paradigms is provided in Appendix B. In this paper, node-selection problems include MCl, MCut, MIS and MVC; edge-selection problems include TSP and ATSP.

2022), etc. Lately promising are the graph diffusion-based generative approaches (DIFUSCO (Sun & Yang, 2023), DIFFUCO (Sanokowski et al., 2024), T2T (Li et al., 2023; 2024), etc.) to learn the instance-level solution distribution.

**LC Solvers.** Most commonly, the policy of the next-step prediction is learned by RL (AM (Kool et al., 2018), POMO (Kwon et al., 2020), MatNet (Kwon et al., 2021), SYM-NCO (Kim et al., 2022), RL4CO (Berto et al., 2024), etc.) with manually imposed masks to hard-guarantee the feasibility. Lately, BQ-NCO (Drakulic et al., 2023) and GOAL (Drakulic et al., 2024) managed to incorporate supervision for sequential decisions to improve the scalability.

We also notice a few works that appear to contain similar determining manners to our AE paradigm (Ahn et al., 2020; Zhou et al., 2025), and defer the detailed discussion, comparison, and empirical studies on them to Appendix B.4. Besides the aforementioned methods to which COExpander is the most comparable, we also discuss in Appendix A 1) traditional solvers from operations research, 2) machine learning-enhanced heuristics, 3) solution optimization methods, and 4) neural divide-and-conquer frameworks, to provide a comprehensive review of existent methods for CO.

## 3. Preliminaries

**Problem Definition.** The COPs studied in this paper can be consistently characterized by the graph structure, i.e., $G = (\mathcal{V}, \mathcal{E})$, where $\mathcal{V} = \{1, \cdots, N\}$ denotes the node set and $\mathcal{E}$ the edge set. Following previous works (Sun & Yang, 2023; Li et al., 2023), we define $\mathbf{x} = \{0, 1\}^M$ as the corresponding decision variables of a given COP. For MCl, MIS, and MVC, $M = N$ and $\mathbf{x}_i$ indicates whether the node $i$ is selected. For MCut, $M = N$ and $\mathbf{x}_i$ indicates which subset the node $i$ belongs to. For TSP and ATSP, $M = N^2$ and $\mathbf{x}_{i \cdot N + j}$ indicates whether the edge $(i, j)$ is selected. Mathematically, the optimization objective is to find optimal solution $\mathbf{x}^* =$

$\underset{\mathbf{x} \in \Omega}{\arg\min}\, c(\mathbf{x}, G)$, where $\Omega$ denotes the feasible set of $\mathbf{x}$ that satisfy the constrains and $c(\cdot, \cdot)$ is the objective cost function of corresponding problems. The formal definitions of the six COPs are presented in Appendix C.

**Global Predictor for COExpander.** Joshi et al. (2019) pioneered with success to adapt Graph Convolutional Network (GCN), the fundamental model for graph data processing, to solving COPs like TSP. Later, Sun & Yang (2023) and Sanokowski et al. (2024) revealed the prospect of generative CO, proposing to integrate the diffusion principles with graph representations and GCN, which entails a serial noising process that converts the initial solution into a noisy vector and a learnable denoising process in reverse. Further, Fast-T2T (Li et al., 2024) devises the scheme of training consistency for vanilla diffusion to ensure that all noised trajectories conditioned on the same graph converge to the same initial solution, which largely reduces the denoising steps needed and puts forward the edge of NCO performance to a great extent. Along this chain, we intuitively aim to improve upon Fast-T2T, thus adapting the consistency model as the global predictor of COExpander in favor of its superior performance and efficiency.

## 4. Methodology

### 4.1. Model Learning: Partial Solution Prompted Generation with Consistent Diffusion

#### 4.1.1. DIFFUSION NOISING PROCESS

Following the definitions in Sec. 3, the distribution of the solution space can be characterized by an $M$-dimensional Bernoulli distribution $p(\mathbf{x}) \in [0, 1]^{M \times 2}$. The objective of applying generative modeling to problem-solving is to learn the distribution of high-quality solutions conditioned on a given $G$, mathematically $p_\theta(\mathbf{x}|G)$. For an initial solution $\mathbf{x}_0$, we adhere to the previous graph-based diffusion and

consistency models (Austin et al., 2021; Sun & Yang, 2023; Li et al., 2024) to define the noise process, i.e.,

$$q(\mathbf{x}_t|\mathbf{x}_0) = p(\mathbf{x}_0)\overline{\mathbf{Q}}_t; \mathbf{Q}_t = \begin{bmatrix} \beta_t & 1-\beta_t \\ 1-\beta_t & \beta_t \end{bmatrix}, \quad (1)$$

where $\overline{\mathbf{Q}}_t = \mathbf{Q}_1\mathbf{Q}_2\cdots\mathbf{Q}_t$ is the cumulative transition probability matrix with $\beta_t \in [0,1]$ to ensure doubly stochasticity.

### 4.1.2. OPTIMIZATION CONSISTENCY

Given a problem instance $G$ and the optimal solution $\mathbf{x}^*$, the noising process for discrete diffusion applied to $\mathbf{x}^*$ is formulated as a trajectory $\mathbf{x}_{0:T} = \mathbf{x}_0, \mathbf{x}_1, \mathbf{x}_2, \cdots, \mathbf{x}_T$, where $\mathbf{x}_0 = \mathbf{x}^*$ and each $\mathbf{x}_i$ is sampled from the distribution $q(\mathbf{x}_t|\mathbf{x}_0)$. While vanilla Diffusion demands multiple inference steps to model $p_\theta(\mathbf{x}_{t-1}|\mathbf{x}_t, G)$ which is time-consuming in practice, to achieve efficient one-step model generation, Consistency Models (Song et al., 2023) have put forward the concept of *self-consistency* that maps any point at any time step to the trajectory's starting point, and Fast-T2T (Li et al., 2024) has further defined the *optimization consistency for COPs* that all points along any trajectory map to their optimal solution conditioned on instance $G$.

The consistency function is denoted as $f_\theta(\mathbf{x}_t, t, G)$ with learnable parameters $\theta$. According to the definition of self-consistency, for two noised points $\mathbf{x}_{t_1}$ and $\mathbf{x}_{t_2}$ on the same trajectory, the original learning objective of the model is to minimize the distance $d(\cdot, \cdot)$ between their mapping points:

$$\mathcal{L}(\theta) = \mathbb{E}\Big[d\Big(f_\theta(\mathbf{x}_{t_1}, t_1, G), f_\theta(\mathbf{x}_{t_2}, t_2, G)\Big)\Big]. \quad (2)$$

To better utilize the supervising information of $\mathbf{x}^*$ during training-stage, the distance between the mapping points $f_\theta(\mathbf{x}_t, t, G)$ and $\mathbf{x}^*$ can be specified as $d(f_\theta(\mathbf{x}_t, t, G), \mathbf{x}^*)$. Further in this work, the triangle inequality is leveraged to bound and reformulate the training objective as

$$\mathcal{L}(\theta) \leq \mathbb{E}\Big[d\big(f_\theta(\mathbf{x}_{t_1}, t_1, G), x^*\big) + d\big(f_\theta(\mathbf{x}_{t_2}, t_2, G), x^*\big)\Big]. \quad (3)$$

This way, the only concentration is as clear as an optimization over the neural estimation $f_\theta$, via supervised learning with $d(\cdot, \cdot) = \text{Binary\_Cross\_Entropy}(\cdot, \cdot)$.

### 4.1.3. THE PROMPT OF PARTIAL SOLUTION

Beyond previous graph-based diffusion and consistency models that directly feed $(\mathbf{x}_t, t, G)$ to the neural network, we introduce an additional prompting component to the model input, i.e., a *masked* array of $(\mathbf{y}_t, t, G, \mathcal{M})$. Here,

$$(\mathbf{y}_t)_i = Y(\mathbf{x}_t, \mathbf{x}^*, \mathcal{M}) = \begin{cases} (\mathbf{x}_t)_i & \mathcal{M}_i = 0 \\ (\mathbf{x}^*)_i & \mathcal{M}_i = 1 \end{cases}, \quad (4)$$

where $\mathcal{M} \in \{0,1\}^M$ has the same shape of the solution $\mathbf{x}$ and is sampled from $M$-dimensional Bernoulli distribution

with a hyper-parameter $\rho$ controlling $\mathbf{Pr}(\mathcal{M}_i = 1)$. This scheme enables the model to learn to make predictions in aware of current solving progress (featured by the partial solution). To guarantee a fundamental performance while promoting data diversity of the model, $\rho$ is also generated conforming to a probability function of

$$P(\rho) = \alpha \cdot \delta(\rho - 0) + (1-\alpha) \cdot U(\rho), \quad (5)$$

where $\delta(\cdot)$ denotes the Dirac delta function and $U(\cdot)$ a uniform distribution over the interval $[0, 1]$. In this paper, we set $\alpha$ as 0.9, and the relevant ablation study is presented in Appendix G.1. Subsequently, the sampled $\mathcal{M}$ is updated according to the constraints of the specific COP type. For instance, in the context of MIS, if a node $j$ that belongs to the maximum independent set (i.e., $\mathbf{x}_j^* = 1$) is in a prompted state (where $\mathcal{M}_j = 1$), then all its neighboring nodes will also be placed in a prompted state.

## 4.2. Problem Solving: Adaptive Expansion via Multi-Step Determination

### 4.2.1. THE MAIN PROCESS OF PROBLEM-SOLVING

In general, with a well-trained model $f_\theta(\cdot, \cdot, \cdot, \cdot)$, COExpander generate solutions $\mathbf{x}$ for a given instance $G$ through multiple determination steps. In each determination process, we first sample $\mathbf{x}_T$ from the uniform distribution and update it with the current partial solution $\mathbf{x}$ and its mask $\mathcal{M}$, then we follow Fast-T2T to use the model to obtain the probability heatmap $\mathcal{H}$ via multi-step inferences. After that, $\mathcal{H}$ is utilized as a guide to expand the number of determined decision variables in solution set and correspondingly update $\mathbf{x}$ and $\mathcal{M}$, as formulated in Algorithm 1.

---

**Algorithm 1** Multi-step solution expanding.

---

**Input:** COExpander model $f_\theta(\cdot, \cdot, \cdot, \cdot)$, problem instance graph $G$, decision variable $\mathbf{x}$ with its mask $\mathcal{M}$, determination operator $\mathcal{D}$, sequence of time points $\tau_1 > \tau_2 > \cdots > \tau_{N_\tau - 1}$, sequence of determination numbers $n_1 < n_2 < \cdots < n_{N_n}$.
Initialize determination step: $d \leftarrow 0$, $\mathcal{M} \leftarrow \mathbf{0}$
**while** $0 \in \mathcal{M}$ **do**
  Sample $\mathbf{x}_T$ from uniform distribution $\mathcal{U}$
  Update with partial solution: $\mathbf{y}_T \leftarrow Y(\mathbf{x}_T, \mathbf{x}, \mathcal{M})$
  Model Inference: $p_\theta(\mathbf{x}_0|G) \leftarrow f_\theta(\mathbf{y}_T, T, G, \mathcal{M})$
  **for** $n = 1$ to $N_\tau - 1$ **do**
    Sample $\mathbf{x}_0$ from distribution $p_\theta(\mathbf{x}_0|G)$
    $\mathbf{x}_{\tau_n} \leftarrow p(\mathbf{x}_0)\overline{\mathbf{Q}}_{\tau_n}$
    $\mathbf{y}_{\tau_n} \leftarrow Y(\mathbf{x}_{\tau_n}, \mathbf{x}, \mathcal{M})$
    $p_\theta(\mathbf{x}_0|G) \leftarrow f_\theta(\mathbf{y}_{\tau_n}, \tau_n, G, \mathcal{M})$
  **end for**
  Predict probability heatmap of $\mathbf{x}$: $\mathcal{H} \sim p_\theta(\mathbf{x}_0|G)$
  $\mathbf{x}, \mathcal{M} \leftarrow \mathcal{D}(\mathcal{H}, G, \mathbf{x}, \mathcal{M}, n_d)$
  $d \leftarrow d + 1$
**end while**
**Output: Solution $\mathbf{x}$**

---

### 4.2.2. DETERMINATION OPERATIONS FOR COPs

**Overview.** A determination operator $\mathcal{D}$ generally consists of two parts. 1) An *order* prioritizing the decision variables $\mathbf{x}_i \in \mathbf{x}$ based on $\mathcal{H}$, and 2) A *rule* to assign values to the decision variables and update the mask accordingly. Note that to avoid extra sensitivity to ad-hoc designs, the operators are implemented upon the simplest greedy heuristics.

During problem solving, given the predicted probability heatmap $\mathcal{H} \in \mathbb{R}^M$, graph instance $G$, current (partial) solution $\mathbf{x} \in \{0,1\}^M$ with the corresponding mask $\mathcal{M} \in \{0,1\}^M$, and the maximum number of determination rounds $n_d$, $\mathcal{D}$ functions iteratively in the following generic manner. It traverses $\mathbf{x}$ in the order of priority, in the case where $\mathcal{M}_i = 0$ (implying an *undetermined* $\mathbf{x}_i$), assign $\mathbf{x}_i = \lambda \in \{0,1\}$ and update its mask via certain rules. If the number of *determined* decision variables (i.e., $\sum_{i=1}^M \mathcal{M}_i$) reaches $n_d$, or if the next assignment of $\mathbf{x}_i$ would violate the constraints of the problem, then terminate this round of determination and return the updated $\mathbf{x}$ and $\mathcal{M}$. Empirically, we define the problem-specific operators as follows.

**MCl.** Order: node $i$ with descending order of $\mathcal{H}_i$. Rule: expand the clique set with node $i$ and assign $\mathbf{x}_i = 1, \mathcal{M}_i = 1$. Further, for $\forall j \in \mathcal{V}, (i,j) \notin \mathcal{E}$, update $\mathbf{x}_j = 0, \mathcal{M}_j = 1$.

**MCut.** Order: node $i$ with descending order of $|\mathcal{H}_i - \frac{1}{2}|$. Rule: expand either partition set with node $i$, i.e., assign $\mathbf{x}_i = \lceil \mathcal{H}_i - \frac{1}{2} \rceil, \mathcal{M}_i = 1$. Note that MCut has no constraints that incur chain reactions for extra variable assignments.

**MIS.** Order: node $i$ with descending order of $\mathcal{H}_i$. Rule: expand the independent set with node $i$, i.e., $\mathbf{x}_i = 1, \mathcal{M}_i = 1$. Further, for $\forall j \in \mathcal{V}, (i,j) \in \mathcal{E}$, update $\mathbf{x}_j = 0, \mathcal{M}_j = 1$.

**MVC.** Order: node $i$ with ascending order of $\mathcal{H}_i$. Rule: select node $i$ that should *not* belong to the vertex cover set, and assign $\mathbf{x}_i = 0, \mathcal{M}_i = 1$. Further, for $\forall j \in \mathcal{V}, (i,j) \in \mathcal{E}$, update $\mathbf{x}_j = 1, \mathcal{M}_j = 1$.

**TSP and ATSP.** Order: edge $(i,j)$ with descending order of $\mathcal{H}_{ij}$. Rule: Select edge $(i,j)$ to prolong the tour and assign $\mathbf{x}_{ij} = 1, \mathcal{M}_{ij} = 1$. Further, if a node has been entered and left, all its neighbors shall also be assigned $\mathcal{M} = 1, \mathbf{x} = 0$.

### 4.3. Model Architecture: A Task-Agnostic Graph Convolutional Network

Sec. 4.1 and Sec. 4.2 have so far depicted a high-level theoretical pipeline of COExpander. Next, we introduce the specific graph convolutional network (GCN) to embody the $\theta$ that previously parameterized the framework. Provided the generality of the six different COPs mentioned earlier, the model is designed with the best cross-problem compatibility so that different COPs can be handled with a single unified model, mainly encompassing tailored embedding layers, graph convolutional layers, and output layers.

**Model Input.** Similar to the training stage, a masked array $(\mathbf{y}_t, t, \mathcal{M}, G)$ is obtained as described in Sec. 4.1.3 and fed to the model. Note the ground truth $x^*$ in Eq. 4 should be replaced by the current (partial) solution $\mathbf{x}$ at solving stage.

**Embedding Layers.** The input items are first projected to h-dimensional embeddings before convolution.

$$(\mathbf{h}_v, \mathbf{h}_e, \mathbf{h}_t) = \mathbf{embed}(\mathbf{y}_t, t, \mathcal{M}, \mathcal{V}, \mathcal{E}), \qquad (6)$$

where $\mathbf{h}_v$, $\mathbf{h}_e$ and $\mathbf{h}_t$ are the embeddings for the node, edge and time step, respectively. In detail, for all 6 COPs, $\mathbf{h}_t = S(t)$. For node-selection problems, $\mathbf{h}_v = S(\mathbf{y}_t)$, $\mathbf{h}_e = S(\mathcal{E})$. For TSP, $\mathbf{h}_v = S(\mathcal{V})$, $\mathbf{h}_e = S(\mathbf{y}_t)$. For ATSP, $\mathbf{h}_v$ is not used, $\mathbf{h}_e = S(\mathbf{y}_t)$. Here, $S(\cdot) = W(S'(\cdot))$, where $S'(\cdot)$ denotes a sinusoidal embedding layer and $W$ denotes a learnable linear layer.

$$S'(t) = \mathrm{cat}\left( \sin \frac{t}{T^{\frac{0}{d}}}, \cos \frac{t}{T^{\frac{0}{d}}}, \sin \frac{t}{T^{\frac{2}{d}}}, \ldots, \cos \frac{t}{T^{\frac{d}{d}}} \right), \quad (7)$$

where $d$ is the embedding dimension, $T$ is a large number (e.g., 10000), and $\mathrm{cat}(\cdot)$ denotes concatenation.

**Convolution Layers.** Developed upon Joshi et al. (2019), this process can be expressed in a compact formulation:

$$(\mathbf{h}_v^{i+1}, \mathbf{h}_e^{i+1}) = \mathrm{Conv}^i(\mathbf{h}_v^i, \mathbf{h}_e^i) + \mathbf{h}_t^i \cdot (\alpha, 1 - \alpha), \quad (8)$$

Specially, for ATSP, we only perform convolution on the edges, using the original edge features $\mathcal{E}$ to update:

$$(\mathbf{h}_e^{i+1}) = \mathrm{Conv}^i(\mathbf{h}_e^i, \mathcal{E}) + \mathbf{h}_t^i \cdot (\alpha, 1 - \alpha), \qquad (9)$$

where $\alpha$ is the indicator of the COP's type (1 for node-selections and 0 for edge-selections). $\mathbf{h}_t^i = W_T^i(\mathrm{ReLU}(\mathbf{h}_t))$ is the time-step feature and $\mathrm{Conv}(\cdot, \cdot)$ denotes the cross-layer convolution operation, i.e., aggregating messages from neighboring nodes/edges and $\mathrm{ReLU}(\cdot)$ activations.

**Output Layers.** To generate heatmaps for both node- and edge-selection tasks, after a group normalization $\mathrm{Norm}(\cdot)$, a 2-D convolution is employed to transform the $L$-th (final) graph convolutional outputs into binary classification logits (corresponding to the Bernoulli distribution), followed by a $\mathrm{Softmax}$ function to form predicted probabilities.

$$\mathcal{H}_{node} = \mathrm{Softmax}(\mathrm{Conv2d}(\mathrm{Norm}(\mathbf{h}_v^L))), \qquad (10)$$

$$\mathcal{H}_{edge} = \mathrm{Softmax}(\mathrm{Conv2d}(\mathrm{Norm}(\mathbf{h}_e^L))). \qquad (11)$$

## 5. Experiments

### 5.1. Datasets and Metrics

**Synthetic Data.** For node-selection problems, we have generated datasets of three types, RB (Xu et al., 2005), ER (Erd6s & Rényi, 1960), and BA (Barabási & Albert, 1999). Following DiffUCO (Sanokowski et al., 2024), DI-FUSCO (Sun & Yang, 2023), we generate *RB-SMALL* and

Table 1. Results on MIS. GP: global prediction; LC: local construction; AE: adaptive expansion, hereinafter.

| METHOD | TYPE | RB-SMALL | | | RB-LARGE | | | ER-700-800 | | | SATLIB | | |
|---|---|---|---|---|---|---|---|---|---|---|---|---|---|
| | | OBJ.↑ | DROP↓ | TIME↓ | OBJ.↑ | DROP↓ | TIME↓ | OBJ.↑ | DROP↓ | TIME↓ | OBJ.↑ | DROP↓ | TIME↓ |
| KaMIS | Heuristics | 20.09* | 0.00% | 45.81s | 43.00* | 0.00% | 56.97s | 44.97* | 0.00% | 60.75s | 425.95* | 0.00% | 24.37s |
| Gurobi | OR | 20.09 | 0.00% | 0.54s | 42.19 | 1.83% | 33.84s | 38.78 | 13.75% | 60.49s | 425.92 | 0.01% | 3.95s |
| DIFUSCO (S=1,$I_s$=50) ‡ | GP | – | – | – | – | – | – | 38.64 | 14.08% | 2.80s | 424.74 | 0.29% | 2.74s |
| Fast-T2T (S=1,$G_s$=5,$I_s$=5) ‡ | GP | 19.50 | 2.89% | 0.41s | – | – | – | 40.69 | 9.51% | 1.03s | 424.44 | 0.36% | 1.70s |
| DiffUCO: CE (S=1, F=1) ‡ | GP | 19.20 | 4.37% | 0.47s | 38.49 | 10.43% | 4.71s | – | – | – | – | – | – |
| COExpander (S=1,$D_s$=20,$I_s$=1) | AE | 19.66 | 2.09% | **0.11s** | 41.23 | 4.06% | 0.62s | 42.38 | 5.75% | 0.47s | 425.05 | 0.22% | **0.23s** |
| DIFUSCO (S=4,$I_s$=50) ‡ | GP | – | – | – | – | – | – | 40.97 | 8.89% | 5.45s | 425.11 | 0.20% | 2.96s |
| Fast-T2T (S=4,$G_s$=5,$I_s$=5) ‡ | GP | 19.74 | 1.70% | 0.86s | – | – | – | 41.74 | 7.17% | 1.96s | 425.00 | 0.25% | 2.67s |
| DiffUCO: CE (S=4, F=1) ‡ | GP | 19.38 | 3.46% | 1.59s | 39.55 | 7.94% | 25.48s | – | – | – | – | – | – |
| COExpander (S=4,$D_s$=20,$I_s$=1) | AE | 19.71 | 1.88% | **0.12s** | 41.44 | 3.58% | **1.30s** | **42.56** | **5.34%** | **0.70s** | 424.78 | 0.28% | **0.27s** |
| COExpander (S=4,$D_s$=5,$I_s$=20) | AE | **19.79** | **1.48%** | 0.65s | 40.69 | 5.31% | 5.48s | 42.13 | 6.32% | 4.30s | **425.28** | **0.16%** | 1.06s |

*RB-LARGE* for MCl, MIS, MVC, *ER-700-800* for MIS, and *BA-SMALL* and *BA-LARGE* for MCut. For TSP, we follow DIFUSCO to conduct experiments on *TSP-50*, *TSP-100*, *TSP-500*, *TSP-1K*, and for ATSP, we conduct experiments on *ATSP-50* and *ATSP-100*, *ATSP-200* and *ATSP-500*.

**Real-World Data.** Following Meta-EGN (Wang & Li, 2023), we conduct experiments for MCl, MVC on two real datasets Twitter (Jure, 2014) and COLLAB (Yanardag & Vishwanathan, 2015). For MIS and TSP, we follow DI-FUSCO (Sun & Yang, 2023) to use SATLIB and TSPLIB.

**Ultra-Large Data.** To further study the generalization of the model, we generate the ultra-large scale datasets. *RB-GIANT* for MCl, MIS, MVC; *ER-1400-1600* for MIS; *BA-GIANT* for MCut and *TSP-10K*. Detailed information regarding the datasets are introduced in Appendix E.

**Metrics. 1) Objective.** The average objective of the solutions w.r.t. the corresponding instances. For MCl, MIS, MVC, the objective is the vertices number of the selected subset. For MCut, the objective is the sum of weights of edges between the two partitioned subsets. For TSP and ATSP, it is the total distance of the solved tour. **2) Drop.** The relative performance drop w.r.t. the objective compared to the solutions obtained by the baseline exact solver or heuristic solver. Mathematically, $\text{drop} = \left| \frac{c(\mathbf{x};G) - c(\mathbf{x}^*;G)}{c(\mathbf{x}^*;G)} \right| \cdot 100\%$. **3) Time.** The average computational time per instance. Unless otherwise specified, all experiments were conducted with a batch size set to 1 or in single-thread mode.

**5.2. Evaluated Methods and Model Settings**

**Baseline Solvers.** We take Gurobi (Gurobi Optimization, 2023) for MCl, MCut, MIS, MVC; KaMIS (Lamm et al., 2016) for MIS; LKH (Helsgaun, 2017), GA-EAX (Nagata & Kobayashi, 2013) and Concorde (Applegate et al., 2006) for TSP, as baseline solvers to obtain reference solutions for computing the optimality drops of the other solvers.

**Neural Solvers.** We compared the following recent neural methods. GCN (Joshi et al., 2019), MatNet (Kwon et al., 2021), GNNGLS (Hudson et al., 2021), DIMES (Qiu et al., 2022), Meta-EGN (Wang & Li, 2023), BQ-NCO (Drakulic et al., 2023), DIFUSCO (Sun & Yang, 2023), RL4CO (Berto et al., 2024), DiffUCO (Sanokowski et al., 2024), Fast-T2T (Li et al., 2024), GOAL (Drakulic et al., 2024).

**Training Settings.** During the training phase, we set the number of convolution layers to 12, the learning rate to 0.0002, and the number of training epochs to 50. During the fine-tuning phase, we set the learning rate to 0.00005 and the number of training epochs to 10. Detailed parameters are provided in Appendix F.

**Basic Solving Settings.** 1) Inference Step $\mathbf{I_s}$: the number of time-steps the model denoises in the inference phase. 2) Sampling Number $\mathbf{S}$: the number of heatmaps sampled from $p_\theta(x_0|s)$ with different random seeds (Sun & Yang, 2023).

**Special Solving Settings.** 1) Determination Step $\mathbf{D_s}$: applied by our COExpander, the maximum number of determination; 2) Gradient-search Step $\mathbf{G_s}$: applied by Fast-T2T, the number of gradient search steps. 3) Evaluation Factor $\mathbf{F}$: applied by DiffUCO, the times the number of diffusion steps is increased compared to that used during training.

**Note on the Implementation.** First, all the aforementioned traditional solvers have been re-implemented and, in the process, we find several insights into their solving performance worthy of further discussion (see Appendix D). Consequently, this paper refrains from citing several obsolete data from previous research. Second, for neural solvers, we employ pre-trained checkpoints to replicate their performance whenever available (in the table: ‡). Otherwise, we retrain the models from scratch (in the table: #). Generally, we present the re-evaluated results of all the compared methods obtained on our standardized datasets and a unified hardware configuration, thus ensuring an impartial and consistent comparison.

**5.3. Problem-Specific Improving Techniques**

By convention, typical per-instance searching techniques are used to further enhance the quality of solutions.

**Beam Search.** For MIS and MCl, when $\mathbf{D_s}$ is set as 1, the

*Table 2.* Results on MCl, MVC and MCut. BS: beam search. FT: finetuned model.

| MAXIMUM CLIQUE | TYPE | RB-SMALL | | | RB-LARGE | | |
|---|---|---|---|---|---|---|---|
| | | OBJ.↑ | DROP↓ | TIME↓ | OBJ.↑ | DROP↓ | TIME↓ |
| Gurobi | OR | 19.08* | 0.00% | 0.90s | 40.18* | 0.00% | 276.66s |
| Meta-EGN ‡ | GP | 17.51 | 8.30% | 0.27s | 33.79 | 15.49% | 0.54s |
| DiffUCO: CE (S=1, F=1) ‡ | GP | 15.14 | 18.25% | 0.56s | – | – | – |
| COExpander (S=1,$D_s$=20,$I_s$=1) | AE | **18.77** | **1.89%** | 0.05s | 36.75 | 8.82% | 0.15s |
| COExpander (S=1,$D_s$=1,$I_s$=1) + BS | GP | 18.66 | 2.51% | **0.02s** | **38.75** | **3.80%** | **0.10s** |
| DiffUCO: CE (S=4, F=1) ‡ | GP | 16.21 | 12.53% | 1.41s | – | – | – |
| COExpander (S=4,$D_s$=5,$I_s$=20) | AE | **19.00** | **0.50%** | 0.65s | 39.06 | 2.99% | 6.36s |
| COExpander (S=4,$D_s$=1,$I_s$=1) + BS | GP | 18.98 | 0.68% | 0.06s | **39.88** | **0.90%** | **0.37s** |

| MINIMUM VERTEX COVER | TYPE | RB-SMALL | | | RB-LARGE | | |
|---|---|---|---|---|---|---|---|
| | | OBJ.↓ | DROP↓ | TIME↓ | OBJ.↓ | DROP↓ | TIME↓ |
| Gurobi | OR | 205.76* | 0.00% | 3.34s | 968.23* | 0.00% | 290.23s |
| Meta-EGN ‡ | GP | 208.97 | 1.56% | 0.30s | 1010.69 | 4.40% | 1.03s |
| COExpander (S=1,$D_s$=20,$I_s$=1) | AE | **206.75** | **0.48%** | **0.21s** | **969.92** | **0.18%** | **0.52s** |
| COExpander (S=4,$D_s$=20,$I_s$=1) | AE | **206.51** | **0.37%** | 0.39s | **969.81** | **0.16%** | 0.63s |

| MAXIMUM CUT | TYPE | BA-SMALL | | | BA-LARGE | | |
|---|---|---|---|---|---|---|---|
| | | OBJ.↑ | DROP↓ | TIME↓ | OBJ.↑ | DROP↓ | TIME↓ |
| Gurobi | OR | 727.84* | 0.00% | 60.61s | 2936.89* | 0.00% | 300.21s |
| DiffUCO: CE (S=1, F=1) ‡ | GP | 726.90 | 0.15% | 0.20s | **2986.93** | **-1.69%** | 0.65s |
| COExpander (S=1,$D_s$=20,$I_s$=1) | AE | **727.53** | **0.06%** | 0.18s | 2978.20 | -1.39% | 0.20s |
| COExpander-FT (S=1,$D_s$=20,$I_s$=1) | AE | 726.54 | 0.20% | **0.17s** | 2980.51 | -1.47% | **0.20s** |
| DiffUCO: CE (S=4, F=1) ‡ | GP | 727.53 | 0.06% | 0.61s | **2989.46** | **-1.77%** | 2.70s |
| COExpander (S=4,$D_s$=1,$I_s$=20) | GP | **728.32** | **-0.05%** | 0.20s | 2960.66 | -0.80% | **0.36s** |
| COExpander-FT (S=4,$D_s$=1,$I_s$=20) | GP | 728.27 | -0.04% | **0.20s** | 2987.14 | -1.69% | 0.36s |

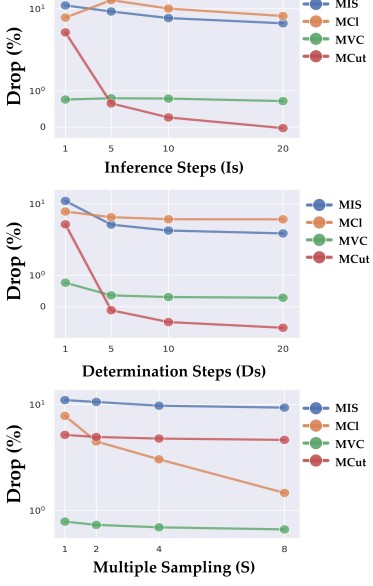

*Figure 2.* Ablation study.

determination process can be enhanced by a beam search strategy where $k$ (default: 16) candidate sets are maintained simultaneously during the search. When a node is being determined, each candidate set is checked, and once the constraints met, the node is added to the corresponding set.

**Objective Guided Fine-Tuning.** Inspired by Sanokowski et al. (2023; 2024), we adopt the energy function of MCut as an unsupervised loss function to fine-tune the MCut model, where $C$ is a constant (1000) to limit the loss function value. Note that we used the same training set as initial training.

$$L = \sum_{(i,j)\in\mathcal{E}} \frac{1}{C} \cdot (2\mathbf{x}_i - 1) \cdot (2\mathbf{x}_j - 1) \qquad (12)$$

**Two-Opt.** Two-Opt is widely employed to improve the solutions for routing problems (Deudon et al., 2018; da Costa et al., 2020; Li et al., 2023). For TSP, given a distance matrix D and an initial tour $\mathbf{x}_1, ..., \mathbf{x}_p, \mathbf{x}_{p+1}, ..., \mathbf{x}_{q-1}, \mathbf{x}_q, \mathbf{x}_{q+1}, ...,$ two nodes $\mathbf{x}_p$ and $\mathbf{x}_q$ are selected to perform the Two-Opt operation, i.e., connecting the two nodes and swapping the subsequence between them and resulting a new path $\mathbf{x}_1, ..., \mathbf{x}_p, \mathbf{x}_q, \mathbf{x}_{q-1}..., \mathbf{x}_{p+1}, \mathbf{x}_{q+1}$. The reward $r = \mathrm{D}_{p,p+1} + \mathrm{D}_{q,q+1} - \mathrm{D}_{p,q} - \mathrm{D}_{p+1,q+1}$ denotes the improvement after the swap. For ATSP, given an initial tour $\mathbf{x}_1, ..., \mathbf{x}_p, \mathbf{x}_{p+1}, ..., \mathbf{x}_{q-1}, \mathbf{x}_q, \mathbf{x}_{q+1}, ...,$ Two-Opt performing on nodes $\mathbf{x}_p$ and $\mathbf{x}_q$ results in a new path $\mathbf{x}_1, ..., \mathbf{x}_p, \mathbf{x}_q, \mathbf{x}_{p+1}..., \mathbf{x}_{q-1}, \mathbf{x}_{q+1}$ with reward $r = \mathrm{D}_{p,p+1} + \mathrm{D}_{q-1,q} + \mathrm{D}_{q,q+1} - \mathrm{D}_{p,q} - \mathrm{D}_{q,p+1} - \mathrm{D}_{q-1,q+1}$. In each iteration, the new tour that yields the maximum reward is fixed for the next round. The iteration terminates when either no further improvements can be made or the maximum number of iterations (default: 5000) is reached.

### 5.4. Main Results

Main experimental results are presented in Table 1 (MIS), Table 2 (MCl, MVC and MCut), and Table 3 (TSP & ATSP). The complete results with more model settings evaluated and standard deviations of the performance drop reported are provided in Table 18 through Table 23 in Appendix G.

**Node-selection Problems.** Compared to previous SOTA, COExpander reduces the drop by **37.9%**, **94.4%**, **92.4%** and **1.4%** on average, with a speedup of **8.4x**, **1.1x**, **2.2x** and **5.9x**, on MIS, MCl, MVC, and MCut, respectively.

**Edge-selection Problems.** Compared to previous SOTA, our COExpander achieves an average performance improvement of **27.8%**, **75.4%** on TSP and ATSP respectively.

### 5.5. Ablation Study

**On Inference Steps.** As shown in Fig. 2 (top), with $\mathbf{I_s}$ increasing (1, 5, 10, 20), the optimality drop generally varies in a downward trend, decreasing from an average of about 7.0% to 4.1%. This naturally aligns with what has been observed in previous diffusion-based models.

**On Determination Steps.** As $\mathbf{D_s}$ increases from 1 to 20, the drop decreases significantly from an average of about 7.0% to 2.2% (Fig. 2 (middle)). This also reasonably conforms to the motivation of AE that more determination rounds enables a more precise utilization of each prediction.

**On Multiple Sampling.** As shown in Fig. 2 (bottom), the drop decreased intuitively as $\mathbf{S}$ grows, reaching 4.8% as $\mathbf{S}$ set to 8 to allow richer diversity of solution sampling.

*Table 3.* Results on TSP and ATSP across problem sizes.

| METHOD | TYPE | TSP-50 | | | TSP-100 | | | TSP-500 | | | TSP-1K | | |
|---|---|---|---|---|---|---|---|---|---|---|---|---|---|
| | | OBJ.↓ | DROP↓ | TIME↓ | OBJ.↓ | DROP↓ | TIME↓ | OBJ.↓ | DROP↓ | TIME↓ | OBJ.↓ | DROP↓ | TIME↓ |
| Concorde | Exact | 5.69* | 0.00% | 0.06s | 7.76* | 0.00% | 0.24s | 16.55* | 0.00% | 18.67s | 23.12* | 0.00% | 84.41s |
| LKH (500) | Heuristics | 5.69 | 0.00% | 0.06s | 7.76 | 0.00% | 0.18s | 16.55 | 0.00% | 1.85s | 23.12 | 0.00% | 4.64s |
| GA-EAX | Heuristics | 5.69 | 0.00% | 0.10s | 7.76 | 0.00% | 1.86s | 16.55 | 0.00% | 1.86s | 23.12 | 0.00% | 17.54s |
| GCN + 2OPT [#] | GP | 5.69 | 0.07% | **0.01s** | 7.78 | 0.24% | **0.01s** | 16.77 | 1.35% | **0.06s** | 23.53 | 1.77% | **0.23s** |
| GNNGLS + 2OPT [#] | GP | 5.71 | 0.33% | 0.02s | 7.86 | 1.30% | 0.13s | – | – | – | – | – | – |
| DIMES + 2OPT [#] | GP | 5.89 | 3.58% | 0.01s | 8.11 | 4.54% | 0.01s | 17.66 | 6.71% | 0.31s | 24.91 | 7.74% | 0.66s |
| DIFUSCO (S=1,$I_s$=50) + 2OPT [‡] | GP | 5.69 | 0.10% | 2.59s | 7.78 | 0.27% | 2.69s | 16.81 | 1.59% | 2.78s | 23.54 | 1.83% | 3.42s |
| Fast-T2T (S=1,$I_s$=5) + 2OPT [‡] | GP | 5.69 | 0.02% | 0.25s | 7.76 | 0.07% | 0.25s | 16.70 | 0.91% | 0.33s | 23.39 | 1.16% | 0.95s |
| RL4CO (SymNCO) + 2OPT [‡] | LC | 5.73 | 0.68% | 0.17s | 7.89 | 1.75% | 0.32s | – | – | – | – | – | – |
| BQ-NCO + 2OPT [‡] | LC | 5.80 | 1.89% | 0.21s | 7.89 | 1.77% | 0.39s | 16.84 | 1.77% | 2.45s | 23.65 | 2.29% | 5.72s |
| COExpander (S=1,$D_s$=3,$I_s$=5) + 2OPT | AE | **5.69** | **0.02%** | 0.08s | **7.76** | **0.04%** | 0.09s | **16.63** | **0.49%** | 0.24s | **23.34** | **0.95%** | 0.70s |
| DIFUSCO (S=4,$I_s$=50) + 2OPT [‡] | GP | 5.69 | 0.02% | 2.69s | 7.76 | 0.07% | 2.69s | 16.70 | 0.92% | 3.03s | 23.42 | 1.31% | 7.89s |
| Fast-T2T (S=4,$I_s$=5) + 2OPT [‡] | GP | 5.69 | 0.01% | 0.26s | 7.76 | 0.02% | 0.27s | 16.63 | 0.50% | **0.47s** | 23.29 | 0.74% | **1.93s** |
| BQ-NCO + Beam-16 + 2OPT [‡] | LC [‡] | 5.79 | 1.72% | 0.24s | 7.87 | 1.48% | 0.90s | 16.77 | 1.33% | 4.02s | 23.51 | 1.71% | 10.34s |
| COExpander (S=4,$D_s$=3,$I_s$=5) + 2OPT | AE | **5.69** | **0.01%** | 0.09s | **7.76** | **0.01%** | 0.18s | **16.59** | **0.25%** | 0.66s | **23.27** | **0.64%** | 2.43s |

| METHOD | TYPE | ATSP-50 | | | ATSP-100 | | | ATSP-200 | | | ATSP-500 | | |
|---|---|---|---|---|---|---|---|---|---|---|---|---|---|
| | | OBJ.↓ | DROP↓ | TIME↓ | OBJ.↓ | DROP↓ | TIME↓ | OBJ.↓ | DROP↓ | TIME↓ | OBJ.↓ | DROP↓ | TIME↓ |
| LKH (1000) | Heuristics | 1.55* | 0.00% | 0.10s | 1.57* | 0.00% | 0.24s | 1.56* | 0.00% | 0.72s | 1.57* | 0.00% | 4.38s |
| MatNet [‡] | LC | 1.58 | 1.32% | **0.04s** | 1.62 | 3.26% | **0.06s** | 3.83 | 145.08% | **0.11s** | – | – | – |
| GOAL [#] | LC | 1.65 | 5.85% | 0.31s | 1.64 | 4.61% | 0.64s | 1.62 | 3.35% | 1.4s | 1.72 | 9.02% | 3.26s |
| COExpander (S=1,$D_s$=3,$I_s$=5) + 2OPT | AE | **1.57** | **1.13%** | 0.09s | **1.60** | **2.26%** | 0.10s | **1.60** | **2.56%** | 0.52s | **1.61** | **2.41%** | **1.77s** |
| MatNet (×16) [‡] | LC | 1.56 | 0.30% | **0.04s** | 1.59 | 1.58% | **0.07s** | 3.73 | 138.40% | **0.16s** | – | – | – |
| GOAL + Beam-16 [#] | LC | 1.63 | 4.74% | 0.35s | 1.62 | 3.53% | 0.91s | 1.61 | 2.86% | 4.69s | 1.70 | 8.13% | 32.24s |
| COExpander (S=4,$D_s$=3,$I_s$=5) + 2OPT | AE | **1.56** | **0.17%** | 0.10s | **1.58** | **0.95%** | 0.52s | **1.59** | **1.50%** | 1.10s | **1.60** | **1.57%** | **6.60s** |

*Table 4.* Results for generalization experiments on ultra-large scale node-selection problems.

| METHOD | TYPE | RB-GIANT (MCl) | | BA-GIANT (MCut) | | RB-GIANT (MIS) | | ER-1400-1600 (MIS) | | RB-GIANT (MVC) | |
|---|---|---|---|---|---|---|---|---|---|---|---|
| | | OBJ.↑ | TIME↓ | OBJ.↑ | TIME↓ | OBJ.↑ | TIME↓ | OBJ.↑ | TIME↓ | OBJ.↓ | TIME↓ |
| Gurobi (60s) | OR | 50.98 | 60.25s | 7216.96 | 61.23s | 44.76 | 56.01s | 40.82 | 62.14s | 2400.80 | 60.01s |
| Gurobi (300s) | OR | 51.92 | 302.64s | 7217.86 | 300.50s | 46.72 | 302.73s | 41.20 | 304.49s | 2398.48 | 211.42s |
| Gurobi (3600s) | OR | 81.52 | 3606.20s | 7217.90 | 3601.34s | **48.56** | 3426.21s | 44.81 | 3602.52s | **2396.78** | 1813.79s |
| COExpander | AE | **84.12** | **1.52s** | **7381.92** | **1.76s** | 46.64 | **14.66s** | 48.43 | **4.52s** | 2400.36 | **8.20s** |

Additionally, under the same number of model calls (e.g., with $\mathbf{D_s} \cdot \mathbf{I_s}$ fixed), the "partial-to-full" multi-step determinations (AE) demonstrate better stability and effectiveness than the "noise-to-full" multi-step inferences (GP). Complete results are provided in Table 17 (Appendix G).

**On Backbone Models.** Despite our design choice of the advanced generative model to embody COExpander, we also test vanilla GCN to empower AE. Results in Table 16 show superior performance of COExpander-GCN over GCN, validating our stance that any backbone model (readily tailored for COP solving) can be adapted to instantiate the AE paradigm to enjoy the model-agnostic performance gain.

### 5.6. Generalization Study

**Cross-Distribution Generalization: On Real-World Data.** For MCl and MVC, we train COExpander on *RB-SMALL* and test it on *Twitter* and *COLLAB*. As shown in Table 19 and Table 20, COExpander achieves a performance improvement of **19.7%** on MCl and **98.5%** on MVC compared with the previous SOTA, i.e., Meta-EGN. For TSP, we test our model on 2D Euclidean instances selected from TSPLIB with 51 to 1002 nodes. A competitive drop of only **0.367%** is achieved. Per-instance results are provided in Table 24.

**Cross-Scale Generalization: On Ultra-Large Scale Data.** For node-selection problems, COExpander is trained on *RB-LARGE* and *ER-700-800*, and tested on *RB-GIANT* and *ER-1400-1600*. Results in Table 4 demonstrate COExpander as the first neural solver to outperform Gurobi(3600s) by **1.9%** with a significant speedup of **1134.1x**. For TSP, we fine-tune the *TSP-1K* model to solve *TSP-10K*, reducing the drop from **1.595%** to a new SOTA of **1.450%** with a speedup of **1.39x**, as shown in Table 22.

**A summary of results on all COPs comparing COExpander with previous SOTA is listed in a single Table 25.**

## 6. Extended Discussions and Experiments

### 6.1. Sampling Methods for the Node-Selection Problems

In addition to learning-based methods, sampling methods like iSCO (Sun et al., 2023) and RLSA (Feng & Yang, 2025) have shown strong capabilities in solving COPs with simple constraints, such as node-selection problems in this paper. These methods transform a problem into an energy function and perform sampling guided by gradient information.

Given that these sampling methods do not depend on specific distributions and do not require training, we take the

*Table 5.* Using the RLSA as a post-processor for COExpander on MIS. $\tau_0$: initial temperature; $d$: regularization factor; $k$: parallel numbers; $t$: iterations; $\beta$: penalty factor of the energy function; $\alpha_1 \& \alpha_2$: initialization factors; FAIL: no feasible solution generated.

| DATASET | RLSA SETTINGS | | | | | | | RLSA (using $2t$) | | | COEXPANDER + RLSA | | |
|---|---|---|---|---|---|---|---|---|---|---|---|---|---|
| | $\tau_0$ | $d$ | $k$ | $t$ | $\beta$ | $\alpha_1$ | $\alpha_2$ | OBJ. | DROP↓ | TIME↓ | OBJ. | DROP↓ | TIME↓ |
| RB-SMALL | 0.01 | 5 | 1000 | 500 | 1.02 | 0.16 | 0.3 | **20.084** | **0.028±0.712%** | 0.534s | 20.070 | 0.093±0.655% | 0.449s |
| RB-LARGE | 0.01 | 5 | 1000 | 1000 | 1.02 | 0.09 | 0.3 | 41.870 | 2.569±1.745% | 2.166s | **42.400** | **1.366±1.493%** | 1.738s |
| ER-700-800 | 0.2 | 10 | 1000 | 1000 | 1.001 | 0.12 | 0.3 | 44.906 | 0.130±1.488% | 1.657s | **44.984** | **-0.041±1.389%** | 1.352s |
| SATLIB | 0.01 | 5 | 1000 | 1000 | 1.02 | 0.67 | 0.3 | FAIL | – | – | **425.316** | **0.151±0.173%** | 1.778s |
| ER-1400-1600 | 0.2 | 10 | 1000 | 1000 | 1.001 | 0.07 | 0.3 | 50.414 | 1.019±1.335% | 3.797s | **50.719** | **0.418±1.489%** | 4.416s |
| RB-GIANT | 0.01 | 5 | 1000 | 1000 | 1.02 | 0.04 | 0.3 | FAIL | – | – | **47.880** | **2.741±2.001%** | 9.526s |

*Table 6.* Results on CVRP across problem sizes.

| METHOD | TYPE | CVRP-50 | | | CVRP-100 | | | CVRP-200 | | | CVRP-500 | | |
|---|---|---|---|---|---|---|---|---|---|---|---|---|---|
| | | OBJ.↓ | DROP↓ | TIME↓ | OBJ.↓ | DROP↓ | TIME↓ | OBJ.↓ | DROP↓ | TIME↓ | OBJ.↓ | DROP↓ | TIME↓ |
| HGS (Vidal et al., 2012) | Heuristics | 10.37* | 0.00% | 1.01s | 15.56* | 0.00% | 20.03s | 19.63* | 0.00% | 60.02s | 37.15* | 0.00% | 360.38s |
| RL4CO (Sym-NCO) + Classic-LS # | LC | **10.56** | **1.91%** | 0.09s | **15.93** | **2.38%** | 0.17s | **20.19** | **2.88%** | 0.34s | **38.70** | **4.17%** | 0.88s |
| COExpander + Classic-LS | AE | 10.77 | 3.90% | **0.04s** | 16.22 | 4.25% | **0.06s** | 20.59 | 4.89% | **0.15s** | 39.12 | 5.34% | **0.61s** |

RLSA as an example and orthogonally apply it as a post-processing plugin for the vertex selection problem on top of our method. Algorithm 2 shows how we integrate RLSA as a post-processor, where $\tau_0$ denotes the initial temperature; $d$ represents the regularization term, where the $d$-th largest gradient value is used to control the overall flipping probability; $k$ indicates the number of parallel runs, which must be greater than $d$; $t$ represents the total number of iterations; $\alpha_2$ denotes the noise factor. To diversify the initialization and facilitate escaping local optima, we introduce some noise to the initial solution $\mathbf{x}_0$. Besides, we have adjusted the initialization part of original RLSA by incorporating prior knowledge: the initial probability of each node is changed from $p_0 = 0.5$ to $p_0 = 0.5\alpha_1$, where $\alpha_1 = \frac{2}{N} \sum_{i=1}^{N}(x_i)$.

---

**Algorithm 2** RLSA as Post-Processor.

**Input:** Graph $G$, Energy Function $H(\cdot)$, hyperparameter ($\tau_0$, $d$, $k$, $t$, $\alpha_2$), Initial Solution $\mathbf{x}_0$.
Parallel $k$:
$\mathbf{x} \leftarrow \text{Bernoulli}((1 - \alpha_2) \cdot \mathbf{x}_0 + \alpha_2 \cdot \text{Bernoulli}(p = 0.5))$
**for** $s = 1$ to $t$ **do**
    $\tau \leftarrow \tau_0(1 - \frac{s-1}{t}), \quad \Delta \leftarrow (2\mathbf{x} - 1) \odot \nabla H(\mathbf{x})$
    **for** $i = 1$ to $N$ **do**
        $p \leftarrow \text{Sigmoid}\left((\Delta_i - \Delta_{(d)})/(2\tau)\right)$
        $c \sim \text{Bernoulli}(p)$
        $\mathbf{x}_i \leftarrow \mathbf{x}_i \cdot (1 - c) + (1 - \mathbf{x}_i) \cdot c$
    **end for**
    **if** $H(\mathbf{x}) < H(\mathbf{x}^*)$ **then**
        $\mathbf{x}^* \leftarrow \mathbf{x}$
    **end if**
**end for**
**Output:** Solution $\mathbf{x}$

---

We conduct experiments on the MIS using RLSA as a solver and as a post-processor for COExpander, as shown in Table 5. It should be noted that, to ensure fairness in the time dimension, we used twice the number of iterations $t$ when employing RLSA as a solver. The experimental results demonstrate the strong solving capability of the sampling

method, but there are also cases where no feasible solution could be found. However, when served as a post-processor, since the initial state starts from a feasible solution, the phenomenon of not finding a feasible solution can be avoided.

### 6.2. Applied to Capacitated Vehicle Routing Problem

We have attempted to apply COExpander to *CVRP-50*, *CVRP-100*, *CVRP-200* and *CVRP-500*, as shown in Table 6, where *Classic-LS* is the post-processing method for CVRP that we have designed based on previous works (Prins, 2004; Vidal et al., 2012). Details of the experimental setup and corresponding discussions are provided in the Appendix H.

The results demonstrate that COExpander performs less favorably on CVRP. We believe this is due to the limitations of the global predictor, i.e. consistency model, of COExpander. To our best knowledge, before our work, consistency model (Li et al., 2024) has only been applied to two combinatorial optimization problems: TSP and MIS. As a direction for future work, we elaborate in Appendix H on the potential of extending AE to more complex constrained problems such as CVRP, via integration with Markov Decision Processes (MDP) and Proximal Policy Optimization (PPO, Schulman et al. (2017)).

## 7. Conclusion and Future Work

We propose the AE paradigm and COExpander model for NCO, combining the advantages of global prediction and local construction methods to achieve adaptive granularity of neural decisions. Our method reaches SOTA performance on 6 COPs compared to 11 neural baselines. Future work includes 1) coupling outer mask updates with inner diffusion steps; 2) solving more COPs like the vehicle routing problems; and 3) fine-tuning the iterative determination with RL for a more principled expansion. Details of future work and limitations of our paper are discussed in Appendix H.

## Impact Statement

This work aims to improve the field of neural combinatorial optimization with a streamlined generative method of stronger performance, higher efficiency and better interpretability. To the best of our knowledge, no aspect of our research raises ethical issues or harmful insights.

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

# A. Additional Related Work

Prior to the prevalence of end-to-end neural combinatorial optimization as we introduced in Sec. 1 and Sec. 2, traditional solvers from operations research and some neural-heuristic attempts were widely applied, demonstrating the progress of the manner and level of neural involvement for solving COPs. Additionally, a series of solution optimizing approaches as well as divide-and-conquer frameworks have also been devised as post- or prior-inference techniques at a higher and orthogonal level for solving COPs. We hereby include them as related work to form a comprehensive review.

**Solvers from Operations Research.** These methods depend on linear (integer) programming algorithms to discover exact or near-optimal solutions to COPs, commonly resorting to branch and bound, cutting planes, meta-heuristics, etc. Representatively, CPLEX (Studio, 2020), Gurobi (Gurobi Optimization, 2023), etc., are globally leading large-scale optimizers. For certain tasks, Concorde (Applegate et al., 2006), LKH (Helsgaun, 2017), GA-EAX (Nagata & Kobayashi, 2013) for TSP, and KaMIS (Lamm et al., 2016) for MIS, etc., are designed with reputed efficiency and quality. These traditional solvers have exhibited strong performance on their respective problems, yet at the cost of a great deal of expert knowledge and solving time, especially when the instances scale up large.

**Machine Learning Enhanced Heuristics.** These works have focused on devising or improving heuristics incorporating machine learning components. They either adjust partial parameters of traditional solvers or design heuristics under neural guidance. VSRLKH (Zheng et al., 2021) and NeuroLKH (Xin et al., 2021) combine LKH with reinforcement learning (RL) and supervised learning (SL) respectively. GNNGLS (Hudson et al., 2021) and NeuralGLS (Sui et al., 2023) hybridize GNNs and Guided Local Search (GLS) for TSP solving. Similar works (da Costa et al., 2020; Sui et al., 2021; Ma et al., 2023) are there to solve COPs by neurally defining 2-, 3-, and k-opt heuristics in a data-driven manner. Kool et al. (2022); Ye et al. (2024a); Kim et al. (2025) also explore to combine ML techniques with heuristics upon dynamic programming, ant colony, and probabilistic search algorithms. Note that a common drawback is their heavy dependence on established solvers and lack of universality, as they require tailoring specialized neural tricks for particular base solvers to achieve satisfactory synergy.

**Solution Optimization Approaches.** These works primarily focus on how to optimize starting from an initial solution, which is typically obtained using a simple heuristic, such as a greedy algorithm. NeuRewriter (Chen & Tian, 2019) uses neural networks to learn how to select heuristics (e.g. $k$-opt) and iteratively rewrite local parts of the current solution to optimize it. NeuOpt (Ma et al., 2023) learns to perform flexible $k$-opt exchanges based on a tailored action factorization method and a customized recurrent dual-stream decoder, and proposed the Guided Infeasible Region Exploration (GIRE) scheme, applying the idea of taboo search to combinatorial optimization. Att-GCRN (Fu et al., 2021) use the heatmap obtained from the global predictor to guide the Monte Carlo search (MCTS), thereby optimizing the initial TSP tour.

**Divide-and-Conquer (D&C) Frameworks.** In terms of solving large scaled CO problems, resorting to divide-and-conquer paradigm is popular and proved feasible and performant, as shown in Fig. 3. Fu et al. (2021) trains a lightweight model to predict sub-heatmaps which are later combined for large TSP instances. Luo et al. (2024) proposes a self-improved learning method for better scalability. Most recently, GLOP (Ye et al., 2024b) and UDC (Zheng et al., 2024) learn to partition large routing problems into sub-TSPs or sub-VRPs and conquer them via local revisers. Kim et al. (2021) introduces a collaborative policy framework that explicitly decouples exploration (via diversified solution generation) and exploitation (via parallelized local revision). In this paper, we primarily focus on the solution construction process, yet D&C methods can always be further employed on top of these paradigms.

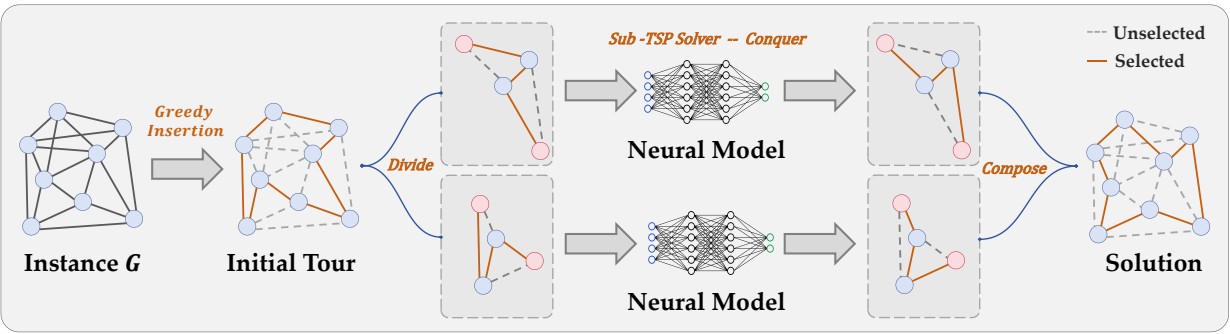

*Figure 3.* The solving pipeline of the D&C frameworks (GLOP (Ye et al., 2024b)). They are mainly applied for edge-selection problems.

# B. Discussion on the Paradigms for Neural Combinatorial Optimization

## B.1. Comparison of Solving Pipelines

**Local Construction (LC).** As shown in Fig. 4 (a), LC use the well-trained neural networks to predict the best next-step action (e.g., selecting the next node/edge) given the current state. This process is executed continuously until a complete solution is constructed.

**Global Prediction (GP).** As shown in Fig. 4 (b), given graph data $G$, GP use the well-trained neural networks (i.e. $f_\theta(\cdot)$) to globally predict the probability $\mathcal{H}$ that each node or edge is selected, and then decode $\mathcal{H}$ to obtain solutions $\mathbf{x}$.

**Adaptive Expansion (AE).** As shown in Fig. 4 (c), given graph data $G$, AE iteratively perform adaptive expansion processes. For each process, AE use the global predictor (i.e. $f_\theta(\cdot, \cdot, \cdot)$) to globally predict the probability $\mathcal{H}$ that each node or edge is selected, and then utilizes the most convincing components of the $\mathcal{H}$ to determine (i.e. $\mathcal{D}(\cdot, \cdot, \cdot, \cdot, \cdot)$) the corresponding decision variables $\mathbf{x}$. We further discuss existing methods that can be categorized in a generalized AE family in Sec. B.4.

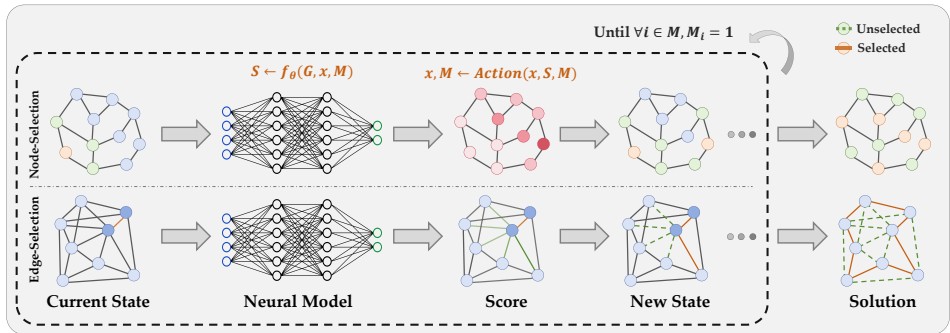

(a) Local Construction.

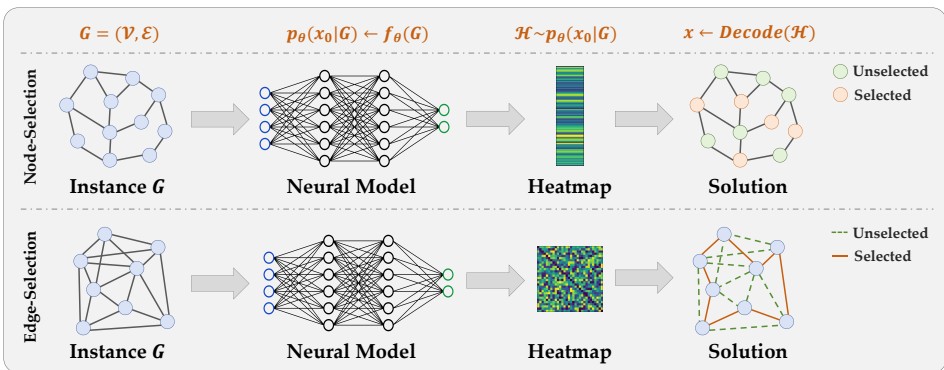

(b) Global Prediction.

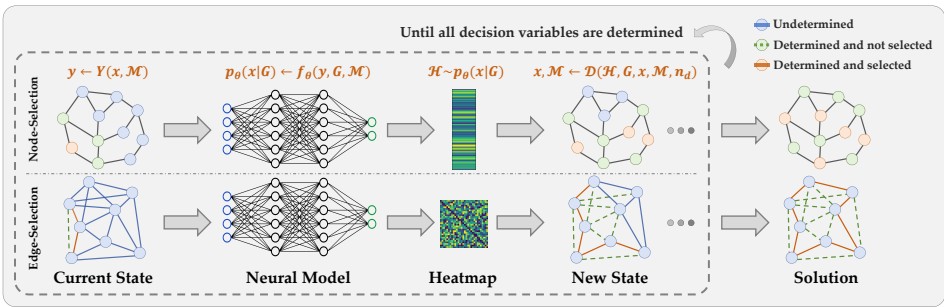

(c) Adaptive Expansion (ours).

*Figure 4.* The solving pipeline of the three paradigms. $G$: graph data; $x$: decision variables; $\mathcal{M}$ and $M$: mask; $y$: partial solution; $\theta$: model parameters; $\mathcal{H}$: probability heatmap; $\mathcal{D}$: determination operator.

## B.2. Limitations of Existing LC Solvers

**Modeling Limitation.** The past LC solvers (Kwon et al., 2021; Drakulic et al., 2023; Berto et al., 2024; Drakulic et al., 2024) generally involves primarily trained to learn how to predict the next node to select/visit the next node based on the current state rather than determining whether the next nodes belong to the solution set. Therefore, they are difficult to directly apply to the four node-selection problems discussed in this paper.

**Subpar Performance and Inefficiency.** We take TSP as an example to compare the differences between Fast-T2T (GP solver) and SymNCO (LC solver) in terms of performance and time. Both adopt a greedy decoding method, and the results are presented in Table 7. Despite the consistent total amount of reasoning information (For TSP-$K$, GP reasons $K^2$ edges in one inference, and LC also reasons a total of $K^2$ edges in $K$ inferences), the actual reasoning time of LCis obviously higher than that of GP due to its sequential decoding method. Additionally, when the scale increases, the solving performance of Sym-NCO drops sharply, and it is even worse than the greedy insertion algorithm when it comes to *TSP-500*.

*Table 7.* Comparison of performance and time of LC and GP paradigms on TSP.

| METHOD | TYPE | TSP-50 | | | TSP-100 | | | TSP-500 | | |
|---|---|---|---|---|---|---|---|---|---|---|
| | | OBJ.↓ | DROP↓ | TIME↓ | OBJ.↓ | DROP↓ | TIME↓ | OBJ.↓ | DROP↓ | TIME↓ |
| Concorde (Applegate et al., 2006) | Exact | 5.688* | 0.000% | 0.074s | 7.756* | 0.000% | 0.404s | 16.546* | 0.000% | 18.672s |
| Insertion (Ye et al., 2024b) | Heuristic | 6.126 | 7.708% | 0.002s | 8.508 | 9.699% | 0.003s | 18.608 | 12.461% | 0.006s |
| Fast-T2T (S=1, I$_s$=1) (Li et al., 2024) | GP | 5.706 | 0.297% | 0.058s | 7.858 | 1.269% | 0.068s | 17.828 | 7.500% | 0.164s |
| Sym-NCO (RL4CO) (Berto et al., 2024) | LC | 5.738 | 0.877% | 0.169s | 7.927 | 2.209% | 0.304s | 21.105 | 27.561% | 0.460s |

## B.3. Limitations of Existing GP Solvers

**Modeling Limitation.** We find that the graph neural networks (GNNs) used by previous GP solvers (Joshi et al., 2019; Sun & Yang, 2023; Li et al., 2024) all performed convolutions on nodes, which required initial node features, such as the node coordinates in TSP. Therefore, these GNNs were difficult to apply to the ATSP, which only has a distance matrix. The MatNet (Kwon et al., 2021) of LC paradigm provides a method for obtaining node features from an asymmetric matrix, but we fail to apply it to our convolutional network. To address this issue, we abandoned node features and adopted convolutions only on edges. To our best knowledge, this is the first global predictor with good performance for ATSP.

**Prediction Conflicts.** Although GP solvers can obtain a large amount of information, i.e., probability heatmap $\mathcal{H}$, in a single inference process, they still require specific decoding methods to extract solutions from $\mathcal{H}$. Take the most common greedy decoding algorithm applied to MIS as an example. When selecting nodes from high to low according to $\mathcal{H}$, the most tricky thing is the addition of the current node causing constraint conflicts. Since the acquisition and decoding of $\mathcal{H}$ is a two-stage process, the only way for GP solvers to deal with such conflicts in the decoding stage is to skip the conflict node and continue searching, which leads to poor final solving results. We refer to this kind of conflict as ***prediction conflict***, and use $\mathcal{N}_c$ to denote the total number of prediction conflict, use $\mathcal{C}_1$ to denote the number of determined nodes/edges when the first prediction conflicts occurs. We take Fast-T2T (Li et al., 2024) as the representative of GP solvers to study and the results are shown in Table 8. We provide the following explanation for the experimental results: 1) An increase in the number of inferences can reduce the frequency of prediction conflicts in the heatmap decoding and delay the first occurrence of prediction conflicts; 2) The reduction of prediction conflicts generally improves the decoding results. However, when the first prediction conflict occurs relatively early (i.e., $\mathcal{C}_1 \ll Obj.$) the interpretability of subsequent predictions is lower, thereby leading the possibility that a single step of inference may perform worse than multi-steps.

*Table 8.* Study of prediction conflicts on MIS.

| METHOD | RB-SMALL | | | | | SATLIB | | | | |
|---|---|---|---|---|---|---|---|---|---|---|
| | Obj.*↑ | Obj.↑ | Drop↓ | $\mathcal{N}_c \downarrow$ | $\mathcal{C}_1 \uparrow$ | Obj.*↑ | Obj.↑ | Drop↓ | $\mathcal{N}_c \downarrow$ | $\mathcal{C}_1 \uparrow$ |
| Fast-T2T (S=1, I$_s$=1) (Li et al., 2024) | 20.090 | 18.552 | 7.553% | 39.334 | 9.604 | 425.954 | 421.734 | 0.995% | 187.240 | 291.448 |
| Fast-T2T (S=1, I$_s$=3) (Li et al., 2024) | 20.090 | 18.412 | 8.239% | 9.532 | 15.034 | 425.954 | 423.266 | 0.634% | 97.326 | 343.450 |
| Fast-T2T (S=1, I$_s$=5) (Li et al., 2024) | 20.090 | 18.818 | 6.265% | 3.092 | 17.272 | 425.954 | 424.112 | 0.434% | 27.862 | 407.824 |

| METHOD | ER-700-800 | | | | | ER-1400-1600 | | | | |
|---|---|---|---|---|---|---|---|---|---|---|
| | Obj.*↑ | Obj.↑ | Drop↓ | $\mathcal{N}_c \downarrow$ | $\mathcal{C}_1 \uparrow$ | Obj.*↑ | Obj.↑ | Drop↓ | $\mathcal{N}_c \downarrow$ | $\mathcal{C}_1 \uparrow$ |
| Fast-T2T (S=1, I$_s$=1) (Li et al., 2024) | 44.969 | 36.922 | 17.888% | 152.070 | 17.414 | 50.938 | 36.414 | 28.499% | 1116.219 | 4.531 |
| Fast-T2T (S=1, I$_s$=3) (Li et al., 2024) | 44.969 | 36.117 | 19.680% | 73.539 | 21.359 | 50.938 | 34.305 | 32.649% | 1010.984 | 4.039 |
| Fast-T2T (S=1, I$_s$=5) (Li et al., 2024) | 44.969 | 37.375 | 16.888% | 22.289 | 29.516 | 50.938 | 33.203 | 34.806% | 959.797 | 5.016 |

## B.4. Comparison of Existing AE Solvers

Similar to the mature paradigms GP and LC, we are dedicated to fostering our newly proposed AE paradigm so that more innovative works can be subsumed within this category. To our knowledge, similar concepts have been devised in a few previous works, while systematic proposal and analyses of a new solving paradigm have yet to emerge till our work. LwD (Ahn et al., 2020) introduces a deferred Markov Decision Process (MDP) that dynamically adjusts the number of decision stages by allowing the agent to either determine or defer vertex inclusion at each step. Extended upon LCP (Kim et al., 2021) and GLOP (Ye et al., 2024b), DualOPT (Zhou et al., 2025) attempts to address TSP through a dual divide-and-conquer strategy. It integrates a grid-based partitioning phase, which decomposes the problem into subgrids solved in parallel using LKH3, and a path-based optimization phase, where neural solvers refine subpaths iteratively. The nodes and edges within each round of conquering process is determined by the rough solution from LKH (Helsgaun, 2017), and only interior determinations are fixed towards subsequent rounds.

Table 9. Comparison of methods with similar design as our AE paradigm.

| METHOD | PARADIGM | TECHNIQUE | APPLICABLE PROBLEMS |
|---|---|---|---|
| LwD (Ahn et al., 2020) | AE | GCN+RL | MIS (locally decomposable) |
| DualOPT (Zhou et al., 2025) | AE | LKH+GAT+RL | TSP |
| COExpander (Ours) | AE | Diffusion+SL | MIS, MVC, MCl, MCut, TSP, ATSP, CVRP |

Table 10. Comparative results on MIS and TSP of AE Solvers. † denotes results reported from the original paper.

| MIS | TYPE | ER-700-800 | | | ER-1400-1600 | | |
|---|---|---|---|---|---|---|---|
| | | OBJ.↑ | DROP↓ | TIME↓ | OBJ.↑ | DROP↓ | TIME↓ |
| KaMIS | Heur. | 44.97* | 0.00% | 60.75s | 50.94* | 0.00% | 60.82s |
| LwD (Ahn et al., 2020) (S=1) | AE | 37.29 | 17.07% | 0.59s | 39.52 | 22.40% | 1.47s |
| COExpander (S=1,$D_s$=20,$I_s$=1) | AE | **42.38** | **5.75%** | **0.47s** | **48.16** | **5.43%** | **2.21s** |
| LwD (Ahn et al., 2020) (S=4) | AE | 39.21 | 12.80% | 0.61s | 42.48 | 16.60% | 1.48s |
| COExpander (S=4,$D_s$=20,$I_s$=1) | AE | **42.56** | **5.34%** | **0.46s** | **48.43** | **4.91%** | 4.52s |

| TSP | TYPE | TSP-1K | | | TSP-10K | | |
|---|---|---|---|---|---|---|---|
| | | OBJ.↓ | DROP↓ | TIME↓ | OBJ.↓ | DROP↓ | TIME↓ |
| Concorde / LKH(500) | Exact / Heur. | 23.12* | 0.00% | 84.41s | 71.76* | 0.00% | 332.76s |
| DualOPT (Zhou et al., 2025)† | AE | 23.31 | 0.83% | 5.60s | **72.62** | **1.21%** | 33.90s |
| COExpander | AE | **23.27** | **0.64%** | **2.43s** | 72.80 | 1.45% | **29.50s** |

**Reproduction of LwD**. We follow the training code provided in the github repository[3] of the Lwd (Ahn et al., 2020) and have made some fixes. We train the model for 20,000 iterations on the unlabeled *ER-700-800* dataset (generated randomly) and test it on our *ER-700-800* and *ER-1400-1600* datasets. The results are shown in Table 10.

**Remark.** Lwd is trained using reinforcement learning (Proximal Policy Optimization), which inevitably encounters challenges such as sparse rewards and scalability constraints. Moreover, its application scope is relatively narrow, solely applicable to the "locally decomposable" problems (i.e., node-selection tasks in our discourse). DualOPT, on the other hand, places greater emphasis on the divide-and-conquer principle. Its core competence stems from the robust initial solutions computed by LKH, even requiring up to 10 runs. However, it is exclusively tailored for TSP. Notably, COExpander represents a pioneering effort in synergizing the latest generative architectures with supervised learning, exhibiting extensive applicability across node-selection (isolated constraints like MIS), edge-selection tasks (simple global constraints like TSP), and even more complex-constrained VRPs (discussed thoroughly in Appendix **??**). On basic COPs on graphs, COExpander achieves state-of-the-art performance without relying on any external pre-solving techniques. Despite the limitations inherent in these methods, we retain an open-minded stance to view them as the preliminary practices within the paradigm of *adaptive solution expansion* (AE) for combinatorial optimization problems on graphs. It is our hope that this novel NCO pipeline will foster further innovations.

---

[3] https://github.com/sungsoo-ahn/learning_what_to_defer

## C. Formal Problem Definitions

In this paper, our proposed method is evaluated on six representative COPs on graphs. Given a graph $G = (\mathcal{V}, \mathcal{E})$, where $\mathcal{V} = \{1, 2, \cdots, N\}$ denotes the node set and $\mathcal{E}$ the edge set, the tasks can be defined as follows.

**Maximum Clique (MCl).** Given $G$, a *clique* $K \subseteq \mathcal{V}$ is a subset of nodes such that every pair of nodes in $K$ is adjacent. Mathematically, it aims to find $K$ that maximizes $|K|$ s.t., $\forall i, j \in K, (i, j) \in \mathcal{E}$.

**Maximum Cut (MCut).** Given $G$, a *cut* $C = (S, \overline{S})$ is a partition of the node set $\mathcal{V}$ into two disjoint sets $S$ and $\overline{S}$. Mathematically, the problem is to find a cut $C$ that maximizes $\sum_{i \in S, j \in \overline{S}} \mathbf{C}_{ij}$, where $\mathbf{C}$ is the adjacency matrix of $G$.

**Maximum Independent Set (MIS).** Given $G$, an *independent set* $S \subseteq \mathcal{V}$ is a subset of nodes such that no two nodes in $S$ are adjacent. Mathematically, it aims to find $S$ that maximizes $|S|$ s.t., $\forall i, j \in S, (i, j) \notin \mathcal{E}$.

**Minimum Vertex Cover (MVC).** Given $G$, a *vertex cover* $C \subseteq \mathcal{V}$ is a subset of nodes such that for every edge $(i, j) \in \mathcal{E}$, at least one of $i$ or $j$ is in $C$. Mathematically, it aims to find $C$ that minimizes $|C|$ s.t., $\forall (i, j) \in \mathcal{E}, i \in C$ or $j \in C$.

**Traveling Salesman Problem (TSP).** Given $G$ along with a cost matrix $\mathbf{C}$ of shape $N \times N$ where the entry $\mathbf{C}_{ij}$ is the cost for edge $(i, j) \in \mathcal{E}$, it aims to find the tour $\tau = (i_1, \cdots, i_N)$ to minimize the cost $\sum_{k=1}^{N-1} \mathbf{C}_{i_k i_{k+1}} + \mathbf{C}_{i_N i_1}$.

**Asymmetric Traveling Salesman Problem (ATSP).** ATSP is a special case of TSP where the cost matrix is not necessarily symmetric, i.e., $\mathbf{C}_{ij} = \mathbf{C}_{ji}$ dose not always hold for all $i, j \in \mathcal{V}$. In particular, we follow (Kwon et al., 2021; Drakulic et al., 2023; Ye et al., 2024b) to study the metric ATSP where the triangle inequality holds, i.e., $\mathbf{C}_{ij} + \mathbf{C}_{jk} \geq \mathbf{C}_{ik}$ for any different three nodes $i$, $j$, and $k$.

## D. Discussion of Non-Learning Baseline Solvers

During our experimental implementation, a number of issues have been uncovered regarding the baseline solvers and benchmark data referenced in numerous prior articles. 1) there is a continuous citation of articles from several years past, yet the cited solvers are either outdated or have incorrect parameter settings. 2) the performance of the devices (dating back several years) utilized is subpar, which presents an obvious unfairness when compared to neural combinatorial optimization (NCO) approaches employing advanced CPU and GPU equipment. Therefore, we have used the more recent versions of all the non-learning traditional solvers employed in this paper, analyzed and discussed the solving parameters of some solvers, and conducted experiments on the dataset in a fair and advanced CPU environment. Additionally, to facilitate the use by future researchers, we have encapsulated all solvers and wrote convenient Python versions.

### D.1. Gurobi

**Introduction.** Gurobi (Gurobi Optimization, 2023) is a versatile optimization solver used for solving various types of mathematical programming problems, including linear programming (LP), mixed-integer programming (MIP), quadratic programming (QP), and more. In this paper, we use Gurobi to solve problems such as MCl, MCut, MIS and MVC. Although the field researching these problems is currently making extensive use of the Gurobi, we notice that there is no work that systematically introduces how Gurobi solves these problems. Therefore, we will provide a detailed introduction to the three stages of Gurobi's solution process for these COPs, thereby providing a basis for the setting of Gurobi's time limits. Note that we use the version 11.0.3[4], released on July 11, 2024.

**Solving Stages. 1) Pre-solving.** The pre-solving stage is the first step in Gurobi's optimization process. During this stage, Gurobi performs several operations to simplify the problem. The objective of pre-solving is to remove unnecessary variables, constraints, and simplify the structure of the problem, making it easier to solve. Additionally, Gurobi will also use built-in heuristic algorithms to obtain an initial solution during this stage. **2) Root Relaxation.** After pre-solving, Gurobi will solve the root relaxation, which refers to solving a relaxed version of the MIP (e.g. treating integer variables as continuous). Root relaxation is a key part of Gurobi's branch-and-bound algorithm, providing an initial solution and helping guide the subsequent search process. **3) Optimization.** The optimization stage follows the root relaxation and begins the main branch-and-bound process. During this stage, Gurobi explores the search tree by branching on decision variables and solving subproblems recursively. The optimization process continues until an optimal solution is found or the time limit is reached.

**An Example.** We take an instance each from the *RB-LARGE* and *RB-GIANT* test datasets of MCl as examples. As shown in

---

[4]https://www.gurobi.com/downloads/gurobi-optimizer-readme-v11-0-3/

Fig. 5, the time for Pre-Solve is very short, which means the solver obtained a heuristic initial solution at a very fast speed and then began the Root Relaxation. During the Root Relaxation stage, the solver is dealing with the relaxed problem, so it does not modify the solution of the original problem. Therefore, if the preset time limit is reached during this stage, it is equivalent to directly returning a relatively poor heuristic solution. This is also why the results are so poor when we set a 300-second time limit for Gurobi to solve *RB-GIANT*. Therefore, when setting the time limit for Gurboi, it is necessary to first obtain the approximate time for Root Relaxation of the problem to be solved, and the time limit must exceed this time.

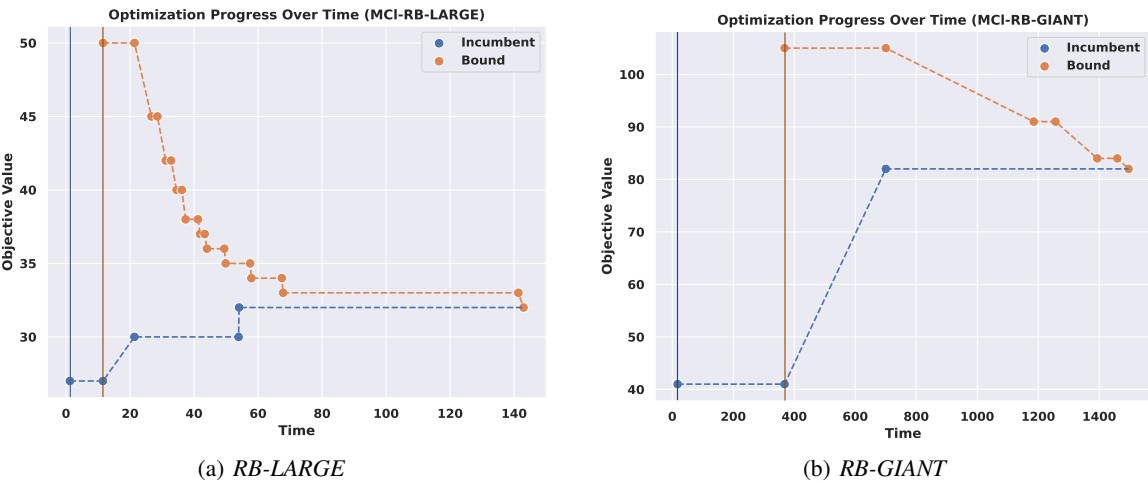

(a) *RB-LARGE*                    (b) *RB-GIANT*

*Figure 5.* The curve of objective during Gurobi solving MCl. Incumbent: the current searched best solution.

## D.2. LKH

LKH (Helsgaun, 2017), Lin-Kernighan-Helsgaun, is a heuristic algorithm for solving the TSP and its variants, such as the ATSP. LKH uses local search and neighborhood exchange techniques (like 2-opt, 3-opt, etc.) to iteratively improve the current solution, generating shorter paths. Known for its strong performance, LKH can find near-optimal solutions and excel in real-world applications, especially for large-scale problems. The version of LKH we used is 3.0.7[5].

We greatly appreciate the work done by PyLKH[6], which provides a Python interface for LKH and has been widely used in various previous studies on ML4TSP, such as DIFUSCO (Sun & Yang, 2023), T2TCO (Li et al., 2023), FastT2T (Li et al., 2024), etc. However, during our re-implementation process, we discovered that the lkh package[7] utilizes the **SPECIAL** parameter when calling LKH. This setting affects several parameters, including Gain23Used. When SPECIAL is enabled, Gain23Used is set to 0 in LKH, which disables the use of both 2-opt and 3-opt. Therefore, we removed this setting when solving the TSP, resulting in a significant improvement. As shown in Table 11, for TSP, we achieved an average performance improvement of **97.7%** in terms of drop when trials set to 100 and **95.8%** when trials set to 500; for ATSP, the performance improvement is **99.7%** and **99.9%** respectively.

## D.3. Others

**KaMIS Solver** (Lamm et al., 2016) is a powerful tool designed for solving the MIS problem. KaMIS employs a combination of greedy algorithms, local search techniques, and meta-heuristics like Simulated Annealing and Genetic Algorithms to explore the solution space. These methods are designed to quickly converge to good solutions, even for large-scale graphs where exact methods would be too slow. We use the version 0.3.0[8], released on Oct 11, 2024.

**Concorde Solver** (Applegate et al., 2006) is an exact solver for the TSP. It combines advanced mathematical optimization techniques such as cutting-plane and branch-and-bound to achieve exact solutions efficiently. We use the version 03.12.19[9], released on Dec 19, 2003.

---

[5] http://akira.ruc.dk/~keld/research/LKH-3/LKH-3.0.7.tgz
[6] https://github.com/ben-hudson/pylkh
[7] https://pypi.org/project/lkh/2.0.0/
[8] https://github.com/KarlsruheMIS/KaMIS/releases/tag/v3.0
[9] https://www.math.uwaterloo.ca/tsp/concorde/downloads/codes/src/co031219.tgz

**GA-EAX Solver** (Nagata & Kobayashi, 2013) is a genetic algorithm-based method for solving TSP. It leverages the Edge Assembly Crossover (EAX) operator, a well-known crossover technique specifically designed for TSP, to generate high-quality solutions. We use the version 1.0[10], released on Dec 29, 2021.

*Table 11.* Comparative results regarding our re-implemented LKH and the widely used original package. "Improv." denotes the relative improvement in terms of the optimality drop.

| DATASET | TRIALS | LKH Package | | | Re-Implementation | | | IMPROV. |
|---------|--------|------|------|------|------|------|------|---------|
|         |        | OBJ.↓ | DROP↓ | TIME↓ | OBJ.↓ | DROP↓ | TIME↓ |         |
| TSP-50 | 100 | 5.68941 | 0.03120±0.15186% | 0.03379s | 5.68770 | 0.00180±0.02233% | 0.03667s | **94.231%** |
| TSP-50 | 500 | 5.68787 | 0.00478±0.05548% | 0.04831s | 5.68763 | 0.00071±0.01361% | 0.05833s | **85.146%** |
| TSP-100 | 100 | 7.77324 | 0.22374±0.37015% | 0.05615s | 7.75615 | 0.00394±0.03000% | 0.07887s | **98.239%** |
| TSP-100 | 500 | 7.76210 | 0.08038±0.19958% | 0.07386s | 7.75593 | 0.00099±0.01413% | 0.17751s | **98.768%** |
| TSP-500 | 100 | 16.79727 | 1.51973±0.55313% | 0.37958s | 16.54771 | 0.01153±0.02733% | 0.72745s | **99.241%** |
| TSP-500 | 500 | 16.69069 | 0.87548±0.37302% | 0.41296s | 16.54616 | 0.00217±0.01042% | 1.84759s | **99.752%** |
| TSP-1K | 100 | 23.60515 | 2.10707±0.53664% | 1.31383s | 23.12207 | 0.01709±0.02402% | 2.11221s | **99.189%** |
| TSP-1K | 500 | 23.43974 | 1.39137±0.36672% | 1.36344s | 23.11936 | 0.00536±0.01069% | 4.64110s | **99.615%** |
| ATSP-50 | 100 | 1.55778 | 0.20841±0.41944% | 0.05988s | 1.55449 | 0.00068±0.01395% | 0.05986s | **99.674%** |
| ATSP-50 | 500 | 1.55572 | 0.07764±0.23443% | 0.08643s | 1.55449 | 0.00014±0.00496% | 0.08226s | **99.820%** |
| ATSP-100 | 100 | 1.57642 | 0.66077±0.64928% | 0.11557s | 1.56606 | 0.00131±0.01127% | 0.12431s | **99.802%** |
| ATSP-100 | 500 | 1.57037 | 0.27685±0.36311% | 0.16125s | 1.56604 | 0.00007±0.00236% | 0.21403s | **99.975%** |

# E. Re-Standardization of Benchmark Datasets

Upon reviewing the recent work relevant to the COPs investigated in this paper, we have identified a substantial disparity in the benchmark datasets employed for evaluation among the works that examine the same problems, including, but not limited to, unfixed data generation process, varied quantity of testing samples, different normalization tricks, etc. This significant variation considerably reduces the informativeness and fairness of the evaluation process. In response to this issue, our work represents the first and vastest attempt to standardize the training and testing datasets within the NCO community for the six COPs included in this paper. Through this effort, we have established a total of 29 normative datasets that serve as benchmarks for different scales of the corresponding problems, as summarized in Table 12. The specific parameters for data synthesis of each type are introduced separately below.

## E.1. Datasets for Node-Selection Problems

**RB Graph.** Following previous works (Li et al., 2024; Sanokowski et al., 2024), we use the so-called RB-Model (Xu et al., 2005) to generate RB Graph. Three main generation parameters: 1) $n$: the number of cliques, which are groups of nodes that are fully interconnected; 2) $k$: the number of nodes within the clique are specified. 3) $p$: the parameter that controls the level of interconnectivity between different cliques. We have generated three scales of RB datasets, each corresponding to a distinct number of nodes. **RB-SMALL**: $v \in (200, 300)$, $n \in (20, 25)$, $k \in (5, 12)$, $p \in (0.3, 1.0)$. **RB-LARGE**: $v \in (800, 1200)$, $n \in (40, 55)$, $k \in (20, 25)$, $p \in (0.3, 1.0)$. **RB-GIANT**: $v \in (2000, 3000)$, $n \in (50, 60)$, $k \in (40, 50)$, $p \in (0.3, 1.0)$. $v$ denotes the number of nodes in a graph.

**ER Graph.** ER graph (Erd6s & Rényi, 1960) is randomly generated with each edge maintaining a fixed probability of being present or absent, independently of the other edges. We follow DIFUSCO (Sun & Yang, 2023), Fast-T2T (Li et al., 2024) to set the probability as 0.15 and generate two datasets of different scales: **ER-700-800** and **ER-1400-1600**.

**BA Graph.** BA graph (Barabási & Albert, 1999) is a scale-free graph where nodes are added incrementally, linking preferentially to highly connected nodes. At each step, a new node is added to the graph, which connects to at most $n$ existing nodes. We follow DiffUCO (Sanokowski et al., 2024) to set $n$ as 4 and generate three datasets of different scales: **BA-SMALL**, **BA-LARGE**, and **BA-GIANT**. The number of nodes are consistent with the RB graph.

**Twitter Graph and COLLAB Graph.** Twitter Graph dataset (Jure, 2014) and COLLAB dataset (Yanardag & Vishwanathan, 2015) are both part of the Stanford Network Analysis Project collection. Twitter Graph represents a snapshot of the social network formed by Twitter users and their interactions, and COLLAB Graph represents collaboration networks among scientists in various fields (nodes are scientists, and edges denote co-authorships). We download the original datasets from

---

[10]https://github.com/nagata-yuichi/GA-EAX

TUDataset[11] and follow Meta-EGN (Wang & Li, 2023) to select 20% of the data as the generalization test sets.

### E.2. Datasets for Edge-Selection Problems

**Uniform Graph.** For TSP, we follow Sun & Yang (2023) to generate TSP instances by randomly sampling nodes from a uniform distribution over the unit square and the data scale ranges from 50 to 10K. For ATSP, we follow the pioneering work (Kwon et al., 2021) to randomly sample the distance matrix from a uniform distribution over the unit square and ensure the satisfaction of triangle inequality. Note that unlike recent works (Drakulic et al., 2023; Ye et al., 2024b; Drakulic et al., 2024; Lischka et al., 2024) that re-permute or re-normalize the distance matrices differently, we resist performing any additional tricks on the graph data to ensure fair comparison. The data scale of ATSP includes 50, 100, 200, 500.

**SATLIB.** Follow previous works (Qiu et al., 2022; Sun & Yang, 2023; Li et al., 2024), we divide the SATLIB[12] dataset, and then transform the SAT problems into MIS problems.

**TSPLIB.** We select 2D-Euclidean distance data with the number of nodes ranging from 51 to 1002 from TSPLIB[13].

*Table 12.* List of our re-collated datasets benchmarking six COPs for training and testing COExpander and any related neural solvers. The "MODEL" column denotes the dataset on which the model is trained. I.e., datasets with the same "TYPE" and "MODEL" is suggested for i.i.d. training-testing, whereas datasets with different "TYPE" and "MODEL" (with "testing" part only) are designed for evaluating the o.o.d. generalization performance of models trained on the corresponding (smaller-scaled, differently distributed, etc.) data. The parentheses indicate the solver parameters used: the maximum solution time for Gurobi and KaMIS; the maximum trials for LKH.

| ID | PROBLEM | TYPE | MODEL | Training | | | Testing | | |
|---|---|---|---|---|---|---|---|---|---|
| | | | | DATA SIZE | SOLVER | STORAGE | DATA SIZE | SOLVER | OBJ. |
| 1 | MCl | RB-SMALL | RB-SMALL | 64,000 | Gurobi(60s) | 3.42GB | 500 | Gurobi(60s) | 19.082 |
| 2 | MCl | Twitter | RB-SMALL | – | – | – | 195 | Gurobi(60s) | 14.210 |
| 3 | MCl | COLLAB | RB-SMALL | – | – | – | 1000 | Gurobi(60s) | 42.113 |
| 4 | MCl | RB-LARGE | RB-LARGE | 6,400 | Gurobi(300s) | 4.74GB | 500 | Gurobi(300s) | 40.182 |
| 5 | MCl | RB-GIANT | RB-LARGE | – | – | – | 50 | Gurobi(3600s) | 81.520 |
| 6 | MCut | BA-SMALL | BA-SMALL | 128,000 | Gurobi(60s) | 1.78GB | 500 | Gurobi(60s) | 727.844 |
| 7 | MCut | BA-LARGE | BA-LARGE | 128,000 | Gurobi(300s) | 8.08GB | 500 | Gurobi(300s) | 2936.886 |
| 8 | MCut | BA-GIANT | BA-LARGE | – | – | – | 50 | Gurobi(300s) | 7217.900 |
| 9 | MIS | RB-SMALL | RB-SMALL | 64,000 | KaMIS(10s) | 3.52GB | 500 | KaMIS(60s) | 20.090 |
| 10 | MIS | RB-LARGE | RB-LARGE | 6,400 | KaMIS(60s) | 4.74GB | 500 | KaMIS(60s) | 43.004 |
| 11 | MIS | RB-GIANT | RB-LARGE | – | – | – | 50 | KaMIS(60s) | 49.260 |
| 12 | MIS | ER-700-800 | ER-700-800 | 12,800 | KaMIS(60s) | 7.83GB | 128 | KaMIS(60s) | 44.969 |
| 13 | MIS | ER-1400-1600 | ER-700-800 | – | – | – | 128 | KaMIS(60s) | 50.938 |
| 14 | MIS | SATLIB | SATLIB | 39,500 | KaMIS(60s) | 3.75GB | 500 | KaMIS(60s) | 425.954 |
| 15 | MVC | RB-SMALL | RB-SMALL | 128,000 | Gurobi(60s) | 7.01GB | 500 | Gurobi(60s) | 205.764 |
| 16 | MVC | Twitter | RB-SMALL | – | – | – | 195 | Gurobi(60s) | 85.251 |
| 17 | MVC | COLLAB | RB-SMALL | – | – | – | 1000 | Gurobi(60s) | 65.086 |
| 18 | MVC | RB-LARGE | RB-LARGE | 6,400 | Gurobi(300s) | 4.74GB | 500 | Gurobi(300s) | 968.228 |
| 19 | MVC | RB-GIANT | RB-LARGE | – | – | – | 50 | Gurobi(300s) | 2398.480 |
| 20 | TSP | Uniform-50 | Uniform-50 | 1,280,000 | Concorde | 2.48GB | 1280 | Concorde | 5.688 |
| 21 | TSP | Uniform-100 | Uniform-100 | 1,280,000 | Concorde | 4.95GB | 1280 | Concorde | 7.756 |
| 22 | TSP | Uniform-500 | Uniform-500 | 64,000 | Concorde | 1.26GB | 128 | Concorde | 16.546 |
| 23 | TSP | Uniform-1K | Uniform-1K | 64,000 | LKH(1000) | 2.53GB | 128 | Concorde | 23.118 |
| 24 | TSP | Uniform-10K | Uniform-10K | 6,400 | LKH(500) | 2.59GB | 16 | LKH(500) | 71.755 |
| 25 | TSP | TSPLIB | Mixed | – | – | – | 49 | Concorde | 8.062 |
| 26 | ATSP | Uniform-50 | Uniform-50 | 640,000 | LKH(500) | 14.72GB | 2500 | LKH(1000) | 1.5545 |
| 27 | ATSP | Uniform-100 | Uniform-100 | 128,000 | LKH(500) | 11.78GB | 2500 | LKH(1000) | 1.5660 |
| 28 | ATSP | Uniform-200 | Uniform-200 | 32,000 | LKH(1000) | 11.76GB | 100 | LKH(1000) | 1.5647 |
| 29 | ATSP | Uniform-500 | Uniform-500 | 6,400 | LKH(1000) | 14.70GB | 100 | LKH(1000) | 1.5734 |

**Broader Impact.** Future research on NCO solvers can be conveniently and consistently evaluated within the unified protocol of our re-standardized benchmarks and re-implemented baseline solvers.

---

[11] https://chrsmrrs.github.io/datasets/docs/datasets/
[12] https://www.cs.ubc.ca/~hoos/SATLIB/Benchmarks/SAT/CBS/descr_CBS.html
[13] http://comopt.ifi.uni-heidelberg.de/software/TSPLIB95/

# F. Details of Model Setting

**Hardware.** All models are trained and tested using NVIDIA H800 (80G) GPUs and an Intel(R) Xeon(R) Platinum 8558 96-Core Processor CPU. For all test evaluations, a single GPU is utilized, and the batch size is set to 1 to ensure a fair comparison of the solving time across different models.

**Training Settings and Hyper-parameters.** We have organized the training settings and model parameters of COExpander in Table 13. In the table, we adopt the name of the targeted training data to denote each individually trained model (21 in total) throughout this paper. In a curriculum-learning manner, models trained on larger-scale data are initialized from those trained on smaller-scale data. For instance, we first train the model for MCl problem on RB-SMALL dataset (line 1) for at least 50 epochs from scratch, and subsequently use the exact model weights to initialize the training on RB-LARGE (line 2) for at least another 10 epochs. Note that AdamW (Loshchilov & Hutter, 2018) optimizer is adopted with a cosine-decayed LR scheduler and a weight decay of 0.0001 to train all the models.

*Table 13.* Hyper-parameters and datasets for training COExpander on different COPs. Pre-train: the checkpoint to initialize corresponding model. Epoch: the minimum number of training epochs required (over corresponding data size) to guarantee reliable reproduction of our reported results. LR: learning rate. LR and batch size are tuned to align each other to suit our computational resources. Sparse: sparsification factor by K-Nearest Neighbors, with -1 meaning the model is trained on dense graphs. Layers: the number of GCN layers.

| ID | PROBLEM | MODEL | PRE-TRAIN | Training Parameters | | | | Model Parameters | | |
|----|---------|-------|-----------|-----------|-------|------------|------|--------|--------|-------------|
| | | | | DATA SIZE | EPOCH | BATCH SIZE | LR | SPARSE | LAYERS | HIDDEN DIM. |
| 1 | MCl | RB-SMALL | – | 64,000 | 50 | 8 | 2e-4 | N/A | 12 | 256 |
| 2 | MCl | RB-LARGE | RB-SMALL | 6,400 | 10 | 4 | 5e-5 | N/A | 12 | 256 |
| 3 | MCut | BA-SMALL | – | 128,000 | 50 | 8 | 2e-4 | N/A | 12 | 256 |
| 4 | MCut | BA-SMALL-FT | BA-SMALL | 128,000 | 10 | 4 | 5e-5 | N/A | 12 | 256 |
| 5 | MCut | BA-LARGE | BA-SMALL | 128,000 | 10 | 4 | 5e-5 | N/A | 12 | 256 |
| 6 | MCut | BA-LARGE-FT | BA-LARGE | 128,000 | 10 | 4 | 5e-5 | N/A | 12 | 256 |
| 7 | MIS | RB-SMALL | – | 64,000 | 50 | 8 | 2e-4 | N/A | 12 | 256 |
| 8 | MIS | RB-LARGE | RB-SMALL | 6,400 | 10 | 4 | 5e-5 | N/A | 12 | 256 |
| 9 | MIS | ER-700-800 | – | 12,800 | 50 | 4 | 2e-4 | N/A | 12 | 256 |
| 10 | MIS | SATLIB | – | 39,500 | 50 | 4 | 2e-4 | N/A | 12 | 256 |
| 11 | MVC | RB-SMALL | – | 128,000 | 50 | 8 | 2e-4 | N/A | 12 | 256 |
| 12 | MVC | RB-LARGE | RB-SMALL | 6,400 | 10 | 4 | 5e-5 | N/A | 12 | 256 |
| 13 | TSP | Uniform-50 | – | 1,280,000 | 50 | 32 | 2e-4 | -1 | 12 | 256 |
| 14 | TSP | Uniform-100 | Uniform-50 | 1,280,000 | 50 | 16 | 2e-4 | -1 | 12 | 256 |
| 15 | TSP | Uniform-500 | Uniform-100 | 64,000 | 50 | 4 | 2e-4 | 50 | 12 | 256 |
| 16 | TSP | Uniform-1K | Uniform-500 | 64,000 | 50 | 2 | 2e-4 | 100 | 12 | 256 |
| 17 | TSP | Uniform-10K | Uniform-1K | 6,400 | 10 | 1 | 5e-5 | 50 | 12 | 256 |
| 18 | ATSP | Uniform-50 | – | 640,000 | 50 | 32 | 2e-4 | -1 | 12 | 256 |
| 19 | ATSP | Uniform-100 | Uniform-50 | 128,000 | 50 | 16 | 2e-4 | -1 | 12 | 256 |
| 20 | ATSP | Uniform-200 | Uniform-100 | 32,000 | 50 | 4 | 2e-4 | -1 | 12 | 256 |
| 21 | ATSP | Uniform-500 | Uniform-200 | 6,400 | 10 | 1 | 5e-5 | -1 | 12 | 256 |

**Discussion on Graph Sparsification.** For node-selection problems, the original data is sparse by nature, eliminating the need for additional processing. In contrast, for edge-selection problems, e.g., TSP and ATSP, in order to cut down on training memory and shrink the search space, the prevalent approach is to employ K-Nearest Neighbors (KNN) to sparsify the distance matrix. However, the application of KNN inevitably results in the loss of partial information from the original problem, which consequently causes deviations in problem-solving.

For instance, given graph data $G$, consider an edge $e_{i,j}$ belonging to the optimal solution set $\mathbf{x}^*$, if $e_{i,j}$ is the $s$-th nearest edge from point $i$, and if $K$ is less than $s$, then the optimal solution of the sparse problem $G_K^s$, i.e., $\mathbf{x}_K^s$, will no longer be equivalent to that on the original dense graph, i.e., $\mathbf{x}^*$. We define $\mathcal{R}_K^s$ as the proportion of edges in $\mathbf{x}^*$ that remain in $G_K^s$, which reflects the degree of influence introduced by sparsification over the quality of the training data with supervision. Additionally, we define the K-nearest sparse loss as $\mathcal{L}_K^s = \left| \frac{c(\mathbf{x}_K^s; G) - c(\mathbf{x}^*; G)}{c(\mathbf{x}^*; G)} \right| \cdot 100\%$, which represents the relative performance drop of $\mathbf{x}_K^s$ with respect to $\mathbf{x}^*$. Obviously, the choice of $K$ involves a trade-off among $\mathcal{R}_K^s$, $\mathcal{L}_K^s$ and the benefits of sparsity, such as reduced memory usage and accelerated search. Therefore, we conduct experiments on the sparse factor $K$ for each node-selection problem, and the results are demonstrated in Table 14. The experimental results indicate that graph sparsification has little adverse impact on problem-solving for TSP ($\mathcal{R}_K^s = 100\%$ with $K = 50$), thus justifying the

presumption that $\mathbf{x}_K^s = \mathbf{x}^*$ when $K$ is set as 50. However, for ATSP, such impact is significant: even with a large value assigned to $K$ (e.g., $K > 100$), a considerable number of optimal edges would be excluded from the supervision label after sparsification. Consequently, in this paper, we apply graph sparsification with $K = 50$ to TSP, but refrain from using graph sparsification for ATSP.

*Table 14.* Results on edge-selection tasks with different sparsification factors.

| PROBLEM | OBJ.*↓ | $K = 20$ | | | $K = 50$ | | | $K = 100$ | | |
|---|---|---|---|---|---|---|---|---|---|---|
| | | $\mathcal{R}_s^K$↑ | OBJ.↓ | $\mathcal{L}_s^K$↓ | $\mathcal{R}_s^K$↑ | OBJ.↓ | $\mathcal{L}_s^K$↓ | $\mathcal{R}_s^K$↑ | OBJ.↓ | $\mathcal{L}_s^K$↓ |
| TSP-500 | 16.546 | 99.990% | – | – | 100.000% | – | – | – | – | – |
| TSP-1K | 23.118 | 99.993% | – | – | 100.000% | – | – | – | – | – |
| ATSP-200 | 1.5647 | 95.120% | 1.6949 | 8.2896% | 98.400% | 1.6082 | 2.7630% | 99.455% | 1.5800 | 0.9551% |
| ATSP-500 | 1.5734 | 94.802% | 1.7391 | 10.5228% | 98.054% | 1.6365 | 4.0116% | 99.124% | 1.6000 | 1.6945% |

| PROBLEM | OBJ.*↓ | $K = 150$ | | | $K = 200$ | | | $K = 250$ | | |
|---|---|---|---|---|---|---|---|---|---|---|
| | | $\mathcal{R}_s^K$↓ | OBJ.↓ | $\mathcal{L}_s^K$↓ | $\mathcal{R}_s^K$↓ | OBJ.↓ | $\mathcal{L}_s^K$↓ | $\mathcal{R}_s^K$↓ | OBJ.↓ | $\mathcal{L}_s^K$↓ |
| ATSP-200 | 1.5647 | 99.860% | 1.5688 | 0.2566% | – | – | – | – | – | – |
| ATSP-500 | 1.5734 | 99.488% | 1.5884 | 0.9526% | 99.668% | 1.5830 | 0.6064% | 99.794% | 1.5802 | 0.4267% |

# G. Supplementary Experiments and Discussions

## G.1. Supplementary Results of Ablation Study

**Ablation on $\alpha$ in Eq. 5**. To determine the value of $\alpha$ in Eq. 5, we conduct ablation experiments on MIS(*RB-SMALL*) as an example. We train the same model under six different settings ($\alpha \in [0.2, 0.5, 0.8, 0.9, 0.95, 0.99]$), focusing primarily on the results of the first seven epochs, as shown in Table 15 and Fig. 6. Since $\alpha$ is merely a probability value for prompting and does not directly affect the loss function, the training results tend to converge when the number of epoch is large. The experimental results show that, when the value of $\alpha$ is set relatively low (e.g., 0.2, 0.5), the model has poor effect on one-shot reasoning. And when $\alpha$ is set relatively high (e.g., 0.95, 0.99), due to the smaller proportion of prompted samples, there may be a certain delay in the model's learning of the mask $\mathcal{M}$. Therefore, we believe that the value of alpha should be set around 0.8 to 0.9. Furthermore, we observed that the first determination round often dominates the final performance, we thus ultimately chose $\alpha = 0.9$ in this paper, with the expectation that the model will be sufficiently trained on cases to solve from scratch while balancing the training diversity for solving from arbitrary intermediate states.

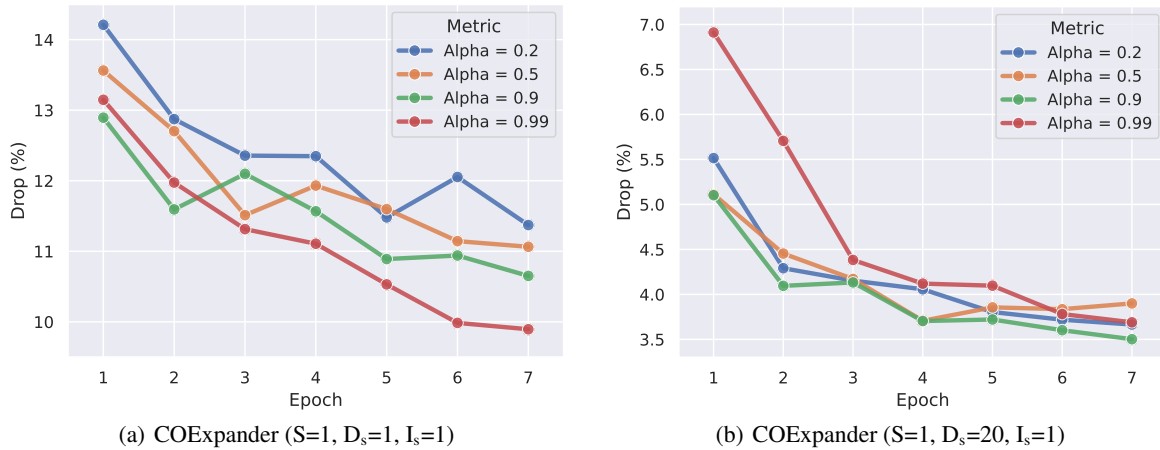

(a) COExpander (S=1, D$_s$=1, I$_s$=1)    (b) COExpander (S=1, D$_s$=20, I$_s$=1)

*Figure 6.* Ablation study on $\alpha$ in Eq. 5 using MIS (*RB-SMALL*) as an example

*Table 15.* Ablation study on $\alpha$ in Eq. 5 using MIS (*RB-SMALL*) as an example.

| | | | COExpander (S=1, $D_s$=1, $I_s$=1) | | | | |
|---|---|---|---|---|---|---|---|
| $\alpha$ | Epoch-1 | Epoch-2 | Epoch-3 | Epoch-4 | Epoch-5 | Epoch-6 | Epoch-7 |
| 0.2 | 17.200, 14.210% | 17.478, 12.873% | 17.584, 12.356% | 17.584, 12.348% | 17.762, 11.480% | 17.642, 12.053% | 17.782, 11.372% |
| 0.5 | 17.338, 13.562% | 17.508, 12.703% | 17.756, 11.512% | 17.668, 11.931% | 17.732, 11.598% | 17.828, 11.144% | 17.842, 11.064% |
| 0.8 | 17.608, 12.222% | 17.644, 12.044% | 17.824, 11.146% | 17.878, 10.881% | 18.012, 10.229% | 17.968, 10.229% | 17.950, 10.522% |
| 0.9 | 17.470, 12.894% | 17.730, 11.594% | 17.634, 12.098% | 17.740, 11.567% | 17.878, 10.890% | 17.868, 10.940% | 17.924, 10.651% |
| 0.95 | 17.548, 12.509% | 17.634, 12.073% | 17.644, 12.046% | 17.890, 10.817% | 17.922, 10.662% | 17.924, 10.662% | 17.948, 10.546% |
| 0.99 | 17.418, 13.147% | 17.658, 11.974% | 17.792, 11.314% | 17.836, 11.107% | 17.950, 10.531% | 18.064, 9.985% | 18.080, 9.895% |
| | | | COExpander (S=1, $D_s$=20, $I_s$=1) | | | | |
| $\alpha$ | Epoch-1 | Epoch-2 | Epoch-3 | Epoch-4 | Epoch-5 | Epoch-6 | Epoch-7 |
| 0.2 | 18.970, 5.515% | 19.214, 4.291% | 19.238, 4.153% | 19.268, 4.057% | 19.312, 3.804% | 19.334, 3.719% | 19.344, 3.665% |
| 0.5 | 19.054, 5.113% | 19.184, 4.454% | 19.240, 4.172% | 19.334, 3.707% | 19.304, 3.856% | 19.312, 3.836% | 19.296, 3.900% |
| 0.8 | 19.114, 4.792% | 19.166, 4.540% | 19.306, 3.839% | 19.354, 3.598% | 19.348, 3.636% | 19.364, 3.550% | 19.356, 3.600% |
| 0.9 | 19.052, 5.103% | 19.252, 4.095% | 19.246, 4.133% | 19.334, 3.704% | 19.334, 3.721% | 19.356, 3.602% | 19.378, 3.502% |
| 0.95 | 18.906, 5.810% | 19.146, 4.632% | 19.184, 4.437% | 19.312, 3.817% | 19.350, 3.632% | 19.340, 3.688% | 19.366, 3.540% |
| 0.99 | 18.684, 6.911% | 18.928, 5.706% | 19.198, 4.383% | 19.246, 4.120% | 19.256, 4.097% | 19.320, 3.782% | 19.340, 3.690% |

**Ablation on Backbone model.** In this paper, we primarily adopt the consistency model (Li et al., 2024) as the global predictor for its powerful expressiveness and performance. However, this does not mean that AE is dependent on the specific consistency model. It should be noted that the AE we propose is a solving paradigm and does not rely on any specific model or solver. To verify this point, we have supplemented experiments using GCN (Joshi et al., 2019) as the global predictor on the MIS problem, as shown in Table 16. The experimental results indicate that the GCN model applying the AE paradigm achieved an average performance improvement of **68.4%**. Additionally, it can also be seen that the consistency model (single inference) has an approximate **6.6%** performance enhancement compared with the GCN model.

*Table 16.* Ablation results of different backbone models for COExpander on the MIS problem. Compare the line of method "X" and the line of method "COExpander-X" to learn the model-agnostic enhancing effect of our AE paradigm. COExpander-CM is the same model as referred to as "COExpander" elsewhere in this paper.

| Method | Type | RB-SMALL | | | RB-LARGE | | |
|---|---|---|---|---|---|---|---|
| | | Obj.↑ | Drop↓ | Time↓ | Obj.↑ | Drop↓ | Time↓ |
| KaMIS (Lamm et al., 2016) | Heuristics | 20.090* | 0.000±0.000% | 45.809s | 43.004* | 0.000±0.000% | 56.974s |
| Gurobi (Gurobi Optimization, 2023) | OR | 20.090 | 0.000±0.000% | 0.538s | 42.192 | 1.829±2.942% | 33.843s |
| GCN (Joshi et al., 2019) | GP | 18.056 | 9.997±6.852% | 0.014s | 36.040 | 16.029±6.449% | 0.044s |
| COExpander-GCN ($D_s$=5) | AE | 19.292 | 3.905±3.936% | 0.050s | 40.092 | 6.683±3.705% | 0.188s |
| COExpander-GCN ($D_s$=10) | AE | 19.376 | 3.480±3.573% | 0.084s | 40.674 | 5.361±3.216% | 0.361s |
| COExpander-GCN ($D_s$=20) | AE | **19.464** | **3.055±3.296%** | 0.121s | **40.892** | **4.862±3.042%** | 0.650s |
| CM (Li et al., 2024) (S=1, $I_s$=1) | GP | 18.400 | 8.305±6.381% | 0.014s | 36.394 | 15.219±6.259% | 0.044s |
| COExpander-CM (S=1, $D_s$=5, $I_s$=1) | AE | 19.500 | 2.876±3.486% | 0.052s | 40.674 | 5.359±3.202% | 0.188s |
| COExpander-CM (S=1, $D_s$=10, $I_s$=1) | AE | 19.596 | 2.410±3.038% | 0.092s | 41.056 | 4.477±2.935% | 0.364s |
| COExpander-CM (S=1, $D_s$=20, $I_s$=1) | AE | **19.662** | **2.088±2.891%** | 0.112s | **41.234** | **4.060±2.822%** | 0.624s |
| Method | Type | ER-700-800 | | | ER-1400-1600 | | |
| | | Obj.↑ | Drop↓ | Time↓ | Obj.↑ | Drop↓ | Time↓ |
| KaMIS (Lamm et al., 2016) | Heuristics | 44.969* | 0.000±0.000% | 60.753s | 50.938* | 0.000±0.000% | 60.824s |
| Gurobi (Gurobi Optimization, 2023) | OR | 38.781 | 13.749±3.017% | 60.489s | 44.813 | 12.015±2.736% | 3602.519s |
| GCN (Joshi et al., 2019) | GP | 35.359 | 21.360±4.087% | 0.039s | 37.742 | 25.895±3.484% | 0.118s |
| COExpander-GCN ($D_s$=5) | AE | 40.477 | 9.984±3.340% | 0.172s | 43.695 | 14.204±3.358% | 0.531s |
| COExpander-GCN ($D_s$=10) | AE | 41.656 | 7.356±2.909% | 0.313s | 45.453 | 10.755±2.702% | 1.016s |
| COExpander-GCN ($D_s$=20) | AE | **42.117** | **6.331±2.712%** | 0.477s | **46.203** | **9.284±2.494%** | 1.555s |
| CM (Li et al., 2024) (S=1, $I_s$=1) | GP | 35.711 | 20.580±4.169% | 0.039s | 37.859 | 25.670±3.368% | 0.117s |
| COExpander-CM (S=1, $D_s$=5, $I_s$=1) | AE | 41.055 | 8.686±3.184% | 0.172s | 45.625 | 10.417±3.282% | 0.531s |
| COExpander-CM (S=1, $D_s$=10, $I_s$=1) | AE | 41.961 | 6.683±2.866% | 0.305s | 47.094 | 7.538±2.866% | 1.001s |
| COExpander-CM (S=1, $D_s$=20, $I_s$=1) | AE | **42.383** | **5.746±2.639%** | 0.469s | **47.523** | **6.691±2.566%** | 1.508s |

**Ablation on S, $D_s$ and $I_s$.** The complete table of the ablation study (corresponding to Fig. 2) is presented in Table 17.

*Table 17.* Ablation results on the number of sampling (**S**), determination steps (**D$_s$**), and inference steps (**I$_s$**). S: SMALL; L: LARGE; ER: ER-700-800; ST: SATLIB.

| Settings | MIS-RB-S | MIS-RB-L | MIS-ER | MIS-ST | MCl-RB-S | MCl-RB-L | MVC-R-S | MVC-R-L | MCut-BA-S | MCut-BA-L |
|---|---|---|---|---|---|---|---|---|---|---|
| S=1,D$_s$=1,I$_s$=1 | 8.305% | 15.219% | 20.580% | 1.031% | 4.870% | 9.809% | 0.827% | 0.696% | 3.504% | 5.231% |
| S=1,D$_s$=1,I$_s$=5 | 6.420% | 14.208% | 15.231% | 0.539% | 4.748% | 22.722% | 0.972% | 0.621% | 0.789% | 0.523% |
| S=1,D$_s$=1,I$_s$=10 | 4.646% | 11.485% | 12.380% | 0.413% | 3.220% | 16.967% | 1.032% | 0.543% | 0.548% | -0.011% |
| S=1,D$_s$=1,I$_s$=20 | 3.710% | 9.388% | 10.618% | 0.329% | 2.850% | 12.672% | 0.984% | 0.453% | 0.319% | -0.369% |
| S=1,D$_s$=5,I$_s$=1 | 2.876% | 5.359% | 8.686% | 0.356% | 2.801% | 8.870% | 0.466% | 0.256% | 0.379% | -0.599% |
| S=1,D$_s$=10,I$_s$=1 | 2.410% | 4.464% | 6.683% | 0.269% | 2.406% | 8.355% | 0.410% | 0.205% | 0.157% | -1.124% |
| S=1,D$_s$=20,I$_s$=1 | 2.088% | 4.060% | 5.746% | 0.215% | 1.892% | 8.817% | 0.398% | 0.176% | 0.058% | -1.387% |
| S=2,D$_s$=1,I$_s$=1 | 7.502% | 15.228% | 19.226% | 0.964% | 2.292% | 5.018% | 0.732% | 0.651% | 3.243% | 5.009% |
| S=4,D$_s$=1,I$_s$=1 | 6.877% | 12.829% | 18.095% | 0.914% | 1.510% | 2.977% | 0.665% | 0.614% | 3.080% | 4.821% |
| S=8,D$_s$=1,I$_s$=1 | 6.409% | 12.083% | 17.382% | 0.878% | 1.027% | 1.720% | 0.611% | 0.581% | 2.949% | 4.670% |

## G.2. Supplementary Results of Generalization Study

**TSPLIB.** For TSP, we conduct generalization experiments on the well-known TSPLIB. The results are shown in Table 24. Our average gap on the number of nodes ranging from 51 to 1002 is only **0.367%**.

## G.3. Complete Results on the Six COPs

**Results for MIS.** The complete results are presented in Table 18. Compare with pervious SOTA NCO solvers, COExpander has significant advantages in both performance and time. Without using sampling, COExpander achieves a performance improvement of **27.7%**, **61.1%**, **39.6%**, **24.6%**, **80.4%**, **61.3%** and a speedup of **3.7x**, **7.6x**, **2.2x**, **12.0x**, **1.6x**, **8.6x** on *RB-SMALL*, *RB-LARGE*, *ER-700-800*, *SATLIB*, *ER-1400-1600*, *RB-GIANT*, respectively. When all methods employ sampling (×4), COExpander realizes a performance improvements of **14.7%**, **54.9%**, **25.5%**, **20.0%**, **20.0%**, **20.0%**, and a speedup of **1.5x**, **19.6x**, **2.8x**, **2.8x**, **1.5x**, **23.1x**. Compare with traditional solvers, COExpander is superior in solving speed, and as the scale increases, such as *ER-1400-1600*, COExpander even excels Gurobi in solving performance.

**Results for MCl.** The complete results are presented in Table 19. Compared with previous SOTA NCO solvers, COExpander has significantly enhanced the solution performance, with a **94.8%** improvement for *RB-SMALL*; and a **94.2%** improvement for *RB-LARGE*. Besides, we have studied how the time limit affect the solution quality of Gurobi on *RB-GIANT* and have made a detailed analysis in Appendix D. However, even when Gurobi is set with a maximum time limit of 3600 seconds, COExpander can still achieve an improvement of **6.4%** with a speedup of **2372.5x**, which fully demonstrates the significant advantage of COExpander in large-scale MCl problems.

**Results for MVC.** The complete results are presented in Table 20. Compared with previous SOTA NCO solvers, COExpander has significantly enhanced the solution performance, with a **81.4%** improvement for *RB-SMALL*; and a **96.3%** improvement for *RB-LARGE*. Besides, we have extended the problem scale to *RB-GIANT* for the first time and have surpassed Gurobi-60s.

**Results for MCut.** The complete results are presented in Table 21. On *BA-SMALL* dataset, COExpander have achieved a breakthrough surpassing the previous SOTA as well as Gurobi-60s; on *BA-LARGE* and *BA-GIANT* dataset, although there was no improvement in performance, COExpander achieves a speed increase of **7.5x**.

**Results for TSP.** The complete results are presented in Table 22. The TSP is one of the most extensively studied problems in NCO, which is also the only one among the six problems discussed in this paper that covers three paradigms as well as the divide-and-conquer (D&C) method. From the experimental results, we can observe that the GP solvers and our COExpander have significant advantages in both performance and speed compared to the LC solver and the D&C method. The previous SOTA is Fast-T2T, which is also the global predictor adopted in this paper, and our COExpander has achieved an improvement of about **17.5%** on this basis.

**Results for ATSP.** The complete results are presented in Table 23. Compare with pervious SOTA NCO solvers, COExpander has achieved a performance improvement of **50.8%**, **36.6%**, **45.5%**, **70.7%** on *ATSP-50*, *ATSP-100*, *ATSP-200*, *ATSP500* respectively. Besides, it is worth noting that when **D$_s$** = 1, COExpander can be regarded as a type of GP solver. To our best knowledge, this is the first effective global predictor for ATSP.

*Table 18.* Complete results on MIS.

| METHOD | TYPE | RB-SMALL | | | RB-LARGE | | |
|---|---|---|---|---|---|---|---|
| | | OBJ.↑ | DROP↓ | TIME↓ | OBJ.↑ | DROP↓ | TIME↓ |
| KaMIS (Lamm et al., 2016) | Heuristics | 20.090* | 0.000±0.000% | 45.809s | 43.004* | 0.000±0.000% | 56.974s |
| Gurobi (Gurobi Optimization, 2023) | OR | 20.090 | 0.000±0.000% | 0.538s | 42.192 | 1.829±2.942% | 33.843s |
| Fast-T2T (S=1,G$_s$=5,I$_s$=5) (Li et al., 2024) | GP | 19.498 | 2.887±3.325% | 0.414s | – | – | – |
| DiffUCO: CE (S=1,F=1) (Sanokowski et al., 2024) | GP | 19.200 | 4.369±3.965% | 0.470s | 38.490 | 10.428±3.552% | 4.712s |
| COExpander (S=1,D$_s$=1,I$_s$=1) | GP | 18.400 | 8.305±6.381% | 0.014s | 36.394 | 15.219±6.259% | 0.044s |
| COExpander (S=1,D$_s$=20,I$_s$=1) | AE | **19.662** | **2.088±2.891%** | 0.112s | **41.234** | **4.060±2.822%** | 0.624s |
| COExpander (S=1,D$_s$=1,I$_s$=20) | GP | 19.330 | 3.710±3.952% | 0.192s | 38.936 | 9.388±4.740% | 0.714s |
| COExpander (S=1,D$_s$=5,I$_s$=20) | AE | 19.604 | 2.375±3.003% | 0.802s | 40.590 | 5.559±3.334% | 3.562s |
| COExpander (S=1,D$_s$=1,I$_s$=1) + Beam-16 | GP | 18.466 | 7.973±6.078% | 0.020s | 36.530 | 14.902±5.974% | 0.104s |
| Fast-T2T (S=4,G$_s$=5,I$_s$=5) (Li et al., 2024) | GP | 19.738 | 1.700±3.321% | 0.860s | – | – | – |
| DiffUCO: CE (S=4,F=1) (Sanokowski et al., 2024) | GP | 19.380 | 3.464±3.388% | 1.587s | 39.546 | 7.944±3.171% | 25.479s |
| COExpander (S=4,D$_s$=1,I$_s$=1) | GP | 18.690 | 6.877±5.675% | 0.018s | 37.424 | 12.829±5.764% | 0.138s |
| COExpander (S=4,D$_s$=20,I$_s$=1) | AE | 19.706 | 1.880±2.691% | 0.120s | **41.438** | **3.582±2.379%** | 1.298s |
| COExpander (S=4,D$_s$=1,I$_s$=20) | GP | 19.742 | 1.690±2.467% | 0.332s | 40.066 | 6.742±3.971% | 2.398s |
| COExpander (S=4,D$_s$=5,I$_s$=20) | AE | **19.786** | **1.482±2.417%** | 0.652s | 40.686 | 5.306±3.290% | 5.484s |
| COExpander (S=4,D$_s$=1,I$_s$=1) + Beam-16 | GP | 18.748 | 6.584±5.404% | 0.052s | 37.498 | 12.658±5.619% | 0.368s |

| METHOD | TYPE | ER-700-800 | | | SATLIB | | |
|---|---|---|---|---|---|---|---|
| | | OBJ.↑ | DROP↓ | TIME↓ | OBJ.↑ | DROP↓ | TIME↓ |
| KaMIS (Lamm et al., 2016) | Heuristics | 44.969* | 0.000±0.000% | 60.753s | 425.954* | 0.000±0.000% | 24.368s |
| Gurobi (Gurobi Optimization, 2023) | OR | 38.781 | 13.749±3.017% | 60.489s | 425.924 | 0.007±0.074% | 3.953s |
| DIFUSCO (S=1,I$_s$=50) (Sun & Yang, 2023) | GP | 38.641 | 14.079±6.564% | 2.797s | 424.744 | 0.285±0.252% | 2.744s |
| Fast-T2T (S=1,G$_s$=5,I$_s$=5) (Li et al., 2024) | GP | 40.688 | 9.513±3.153% | 1.031s | 424.438 | 0.358±0.256% | 1.704s |
| COExpander (S=1,D$_s$=1,I$_s$=1) | GP | 35.711 | 20.580±4.169% | 0.039s | 421.578 | 1.031±0.508% | 0.016s |
| COExpander (S=1,D$_s$=20,I$_s$=1) | AE | **42.383** | **5.746±2.639%** | 0.469s | **425.046** | **0.215±0.235%** | 0.228s |
| COExpander (S=1,D$_s$=1,I$_s$=20) | GP | 40.195 | 10.618±3.117% | 0.641s | 424.556 | 0.329±0.247% | 0.190s |
| COExpander (S=1,D$_s$=5,I$_s$=20) | AE | 41.992 | 6.610±2.577% | 3.188s | 424.734 | 0.287±0.228% | 1.034s |
| DIFUSCO (S=4,I$_s$=50) (Sun & Yang, 2023) | GP | 40.969 | 8.889±3.122% | 5.445s | 425.108 | 0.200±0.201% | 2.962s |
| Fast-T2T (S=4,G$_s$=5,I$_s$=5) (Li et al., 2024) | GP | 41.742 | 7.169±2.837% | 1.961s | 425.002 | 0.252±0.197% | 2.670s |
| COExpander (S=4,D$_s$=1,I$_s$=1) | GP | 36.828 | 18.095±3.726% | 0.109s | 422.078 | 0.914±0.480% | 0.036s |
| COExpander (S=4,D$_s$=20,I$_s$=1) | AE | **42.563** | **5.343±2.229%** | 0.703s | 424.776 | 0.278±0.245% | 0.268s |
| COExpander (S=4,D$_s$=1,I$_s$=20) | GP | 41.430 | 7.867±2.460% | 2.094s | 425.276 | 0.160±0.161% | 0.428s |
| COExpander (S=4,D$_s$=5,I$_s$=20) | AE | 42.125 | 6.321±2.323% | 4.297s | **425.278** | **0.160±0.168%** | 1.056s |

| METHOD | TYPE | ER-1400-1600 | | | RB-GIANT | | |
|---|---|---|---|---|---|---|---|
| | | OBJ.↑ | DROP↓ | TIME↓ | OBJ.↑ | DROP↓ | TIME↓ |
| KaMIS (Lamm et al., 2016) | Heuristics | 50.938* | 0.000±0.000% | 60.824s | 49.260* | 0.000±0.000% | 60.960s |
| Gurobi (3600s) (Gurobi Optimization, 2023) | OR | 44.813 | 12.015±2.736% | 3602.519s | 48.560 | 1.377±2.452% | 3426.207s |
| Gurobi (300s) (Gurobi Optimization, 2023) | OR | 41.203 | 19.098±2.795% | 304.485s | 46.720 | 4.900±7.737% | 302.726s |
| Gurobi (60s) (Gurobi Optimization, 2023) | OR | 40.820 | 19.856±2.575% | 62.141s | 44.760 | 8.806±10.968% | 56.010s |
| Fast-T2T (S=1,G$_s$=5,I$_s$=5) (Li et al., 2024) | GP | 36.797 | 27.749±2.602% | 3.570s | – | – | – |
| DiffUCO: CE (S=1,F=1) (Sanokowski et al., 2024) | GP | – | – | – | 40.480 | 17.722±5.765% | 45.680s |
| COExpander (S=1,D$_s$=1,I$_s$=1) | GP | 37.859 | 25.670±3.368% | 0.117s | 38.200 | 22.316±6.721% | 0.220s |
| COExpander (S=1,D$_s$=50,I$_s$=1) | AE | **48.164** | **5.432±2.188%** | 2.211s | **45.840** | **6.856±3.531%** | 5.340s |
| COExpander (S=1,D$_s$=1,I$_s$=50) | GP | 44.094 | 13.430±3.135% | 5.117s | 38.380 | 21.995±6.499% | 8.980s |
| Fast-T2T (S=4,G$_s$=5,I$_s$=5) (Li et al., 2024) | GP | 37.984 | 25.417±2.197% | 6.961s | – | – | – |
| DiffUCO: CE (S=4,F=1) (Sanokowski et al., 2024) | GP | – | – | – | 42.120 | 14.352±4.582% | 339.010s |
| COExpander (S=4,D$_s$=50,I$_s$=1) | AE | **48.429** | **4.911±2.107%** | 4.516s | **46.640** | **5.221±2.970%** | 14.660s |
| COExpander (S=4,D$_s$=1,I$_s$=50) | GP | 46.117 | 9.446±2.654% | 23.781s | 40.260 | 18.120±6.109% | 43.760s |

*Table 19.* Complete results on MCl.

| METHOD | TYPE | RB-SMALL | | | RB-LARGE | | |
|---|---|---|---|---|---|---|---|
| | | OBJ.↑ | DROP↓ | TIME↓ | OBJ.↑ | DROP↓ | TIME↓ |
| Gurobi (Gurobi Optimization, 2023) | OR | 19.082* | 0.000±0.000% | 0.900s | 40.182* | 0.000±0.000% | 276.657s |
| DiffUCO: CE (S=1,F=1) (Sanokowski et al., 2024) | GP | 15.142 | 18.245±18.185% | 0.559s | – | – | – |
| Meta-EGN (Wang & Li, 2023) | GP | 17.512 | 8.300±12.719% | 0.270s | 33.792 | 15.487±18.576% | 0.542s |
| COExpander (S=1,$D_s$=1,$I_s$=1) | GP | 18.258 | 4.870±10.775% | 0.016s | 36.402 | 9.809±16.700% | 0.042s |
| COExpander (S=1,$D_s$=20,$I_s$=1) | AE | **18.766** | **1.892±4.479%** | 0.046s | **36.752** | **8.817±15.326%** | 0.150s |
| COExpander (S=1,$D_s$=1,$I_s$=20) | GP | 18.608 | 2.850±6.945% | 0.200s | 35.384 | 12.672±19.874% | 0.708s |
| COExpander (S=1,$D_s$=5,$I_s$=20) | AE | 18.686 | 2.374±5.831% | 0.516s | 36.176 | 10.159±16.850% | 1.660s |
| COExpander (S=1,$D_s$=1,$I_s$=1) + Beam-16 | GP | 18.664 | 2.510±6.024% | 0.022s | **38.754** | **3.798±9.039%** | 0.104s |
| COExpander (S=1,$D_s$=1,$I_s$=20) + Beam-16 | GP | **18.768** | **1.914±4.745%** | 0.200s | – | – | – |
| DiffUCO: CE (S=4,F=1) (Sanokowski et al., 2024) | GP | 16.206 | 12.527±18.723% | 1.412s | – | – | – |
| COExpander (S=4,$D_s$=1,$I_s$=1) | GP | 18.846 | 1.510±4.082% | 0.018s | 39.074 | 2.977±8.296% | 0.130s |
| COExpander (S=4,$D_s$=20,$I_s$=1) | AE | 18.922 | 1.018±3.009% | 0.056s | **39.170** | **2.722±7.875%** | 0.414s |
| COExpander (S=4,$D_s$=1,$I_s$=20) | GP | 18.982 | 0.607±2.521% | 0.328s | 38.706 | 4.197±11.586% | 2.386s |
| COExpander (S=4,$D_s$=5,$I_s$=20) | AE | **19.002** | **0.503±2.077%** | 0.646s | 39.062 | 2.993±8.985% | 6.358s |
| COExpander (S=4,$D_s$=1,$I_s$=1) + Beam-16 | GP | 18.976 | 0.675±2.156% | 0.058s | **39.880** | **0.898±3.150%** | 0.368s |
| COExpander (S=4,$D_s$=1,$I_s$=20) + Beam-16 | GP | **19.014** | **0.434±2.061%** | 0.362s | – | – | – |

| METHOD | TYPE | Twitter | | | COLLAB | | |
|---|---|---|---|---|---|---|---|
| | | OBJ.↑ | DROP↓ | TIME↓ | OBJ.↑ | DROP↓ | TIME↓ |
| Gurobi (Gurobi Optimization, 2023) | OR | 14.210* | 0.000±0.000% | 0.276s | 42.113* | 0.000±0.000% | 0.063s |
| Meta-EGN (Wang & Li, 2023) | GP | 13.677 | 3.812±6.352% | 0.134s | 41.993 | 0.662±3.392% | 0.129s |
| COExpander (S=1,$D_s$=1,$I_s$=1) | GP | 13.051 | 9.536±13.448% | 0.015s | 41.593 | 2.496±9.210% | 0.013s |
| COExpander (S=1,$D_s$=20,$I_s$=1) | AE | **13.528** | **6.154±11.888%** | 0.067s | **41.874** | **1.808±8.092%** | 0.139s |
| COExpander (S=1,$D_s$=1,$I_s$=20) | GP | 12.236 | 16.961±24.014% | 0.195s | 39.501 | 14.496±26.836% | 0.222s |
| COExpander (S=1,$D_s$=1,$I_s$=1) + Beam-16 | GP | **13.303** | **6.504±8.451%** | 0.021s | **41.772** | **1.695±5.767%** | 0.014s |
| COExpander (S=1,$D_s$=1,$I_s$=20) + Beam-16 | GP | 12.979 | 10.067±13.192% | 0.200s | 41.084 | 5.815±11.750% | 0.220s |
| COExpander (S=4,$D_s$=1,$I_s$=1) | GP | 13.687 | 4.672±9.833% | 0.021s | 42.000 | 1.089±7.022% | 0.022s |
| COExpander (S=4,$D_s$=20,$I_s$=1) | AE | **13.805** | **3.997±10.243%** | 0.082s | **42.040** | **0.841±6.363%** | 0.092s |
| COExpander (S=4,$D_s$=1,$I_s$=20) | GP | 13.349 | 8.596±18.106% | 0.297s | 41.051 | 5.974±17.730% | 0.314s |
| COExpander (S=4,$D_s$=1,$I_s$=1) + Beam-16 | GP | **13.810** | **2.766±4.777%** | 0.041s | **42.034** | **0.583±3.183%** | 0.025s |
| COExpander (S=4,$D_s$=1,$I_s$=20) + Beam-16 | GP | 13.610 | 5.157±9.572% | 0.318s | 41.620 | 3.023±9.230% | 0.437s |

| METHOD | TYPE | RB-GIANT | | |
|---|---|---|---|---|
| | | OBJ.↑ | DROP↓ | TIME↓ |
| Gurobi (3600s) (Gurobi Optimization, 2023) | OR | 81.520* | 0.000±0.000% | 3606.201s |
| Gurobi (300s) (Gurobi Optimization, 2023) | OR | 51.920 | 32.774±23.131% | 302.640s |
| Gurobi (60s) (Gurobi Optimization, 2023) | OR | 50.980 | 33.917±22.693% | 60.248s |
| COExpander (S=1,$D_s$=1,$I_s$=1) | GP | 60.860 | 22.902±29.840% | 0.240s |
| COExpander (S=1,$D_s$=1,$I_s$=1) + Beam-16 | GP | **77.340** | **2.732±24.924%** | 0.420s |
| COExpander (S=4,$D_s$=1,$I_s$=1) | GP | 77.860 | 2.145±25.410% | 0.820s |
| COExpander (S=4,$D_s$=1,$I_s$=1) + Beam-16 | GP | **84.120** | **-6.424±26.720%** | 1.520s |

*Table 20.* Complete results on MVC.

| METHOD | TYPE | RB-SMALL | | | RB-LARGE | | |
|---|---|---|---|---|---|---|---|
| | | OBJ.↓ | DROP↓ | TIME↓ | OBJ.↓ | DROP↓ | TIME↓ |
| Gurobi (Gurobi Optimization, 2023) | OR | 205.764* | 0.000±0.000% | 3.341s | 968.228* | 0.000±0.000% | 290.227s |
| Meta-EGN (Wang & Li, 2023) | GP | 208.974 | 1.563±0.579% | 0.297s | 1010.692 | 4.400±0.337% | 1.025s |
| COExpander (S=1,D$_s$=1,I$_s$=1) | GP | 207.460 | 0.827±0.575% | 0.016s | 974.950 | 0.696±0.276% | 0.044s |
| COExpander (S=1,D$_s$=20,I$_s$=1) | AE | **206.576** | **0.398±0.381%** | 0.108s | **969.922** | **0.176±0.128%** | 0.522s |
| COExpander (S=1,D$_s$=1,I$_s$=20) | GP | 207.778 | 0.984±0.577% | 0.186s | 972.608 | 0.453±0.195% | 0.718s |
| COExpander (S=1,D$_s$=5,I$_s$=20) | AE | 207.374 | 0.787±0.535% | 0.870s | 970.734 | 0.259±0.151% | 3.520s |
| COExpander (S=4,D$_s$=1,I$_s$=1) | GP | 207.126 | 0.665±0.500% | 0.018s | 974.162 | 0.614±0.244% | 0.132s |
| COExpander (S=4,D$_s$=20,I$_s$=1) | AE | **206.360** | **0.290±0.332%** | 0.086s | **969.806** | **0.163±0.118%** | 0.626s |
| COExpander (S=4,D$_s$=1,I$_s$=20) | GP | 207.060 | 0.633±0.462% | 0.322s | 971.420 | 0.330±0.163% | 2.412s |
| COExpander (S=4,D$_s$=5,I$_s$=20) | AE | 206.906 | 0.557±0.430% | 0.646s | 970.416 | 0.227±0.124% | 4.778s |

| METHOD | TYPE | Twitter | | | COLLAB | | |
|---|---|---|---|---|---|---|---|
| | | OBJ.↓ | DROP↓ | TIME↓ | OBJ.↓ | DROP↓ | TIME↓ |
| Gurobi (Gurobi Optimization, 2023) | OR | 85.251* | 0.000±0.000% | 0.133s | 65.086* | 0.000±0.000% | 0.058s |
| Meta-EGN (Wang & Li, 2023) | GP | 92.728 | 8.670±4.598% | 0.192s | 65.694 | 1.248±2.457% | 0.148s |
| COExpander (S=1,D$_s$=20,I$_s$=1) | AE | **85.574** | **0.326±0.574%** | 0.138s | 65.121 | **0.047±0.303%** | 0.056s |
| COExpander (S=1,D$_s$=5,I$_s$=20) | AE | 86.031 | 0.857±1.066% | 0.862s | **65.114** | 0.057±0.391% | 0.503s |
| COExpander (S=4,D$_s$=20,I$_s$=1) | AE | **85.446** | **0.204±0.481%** | 0.062s | 65.107 | 0.028±0.227% | 0.038s |
| COExpander (S=4,D$_s$=5,I$_s$=20) | AE | 85.574 | 0.318±0.578% | 0.446s | **65.092** | **0.008±0.110%** | 0.354s |

| METHOD | TYPE | RB-GIANT | | |
|---|---|---|---|---|
| | | OBJ.↓ | DROP↓ | TIME↓ |
| Gurobi (3600s) (Gurobi Optimization, 2023) | OR | 2396.780* | 0.000±0.000% | 1813.786s |
| Gurobi (300s) (Gurobi Optimization, 2023) | OR | 2398.480 | 0.067±0.113% | 211.428s |
| Gurobi (60s) (Gurobi Optimization, 2023) | OR | 2400.800 | 0.162±0.172% | 58.121s |
| COExpander (S=1,D$_s$=1,I$_s$=1) | GP | 2407.540 | 0.450±0.110% | 0.240s |
| COExpander (S=1,D$_s$=50,I$_s$=1) | AE | **2400.600** | **0.160±0.061%** | 4.360s |
| COExpander (S=4,D$_s$=1,I$_s$=1) | GP | 2406.060 | 0.388±0.104% | 0.820s |
| COExpander (S=4,D$_s$=50,I$_s$=1) | AE | **2400.360** | **0.149±0.052%** | 8.200s |

*Table 21.* Complete results on MCut.

| METHOD | TYPE | BA-SMALL | | | BA-LARGE | | |
|---|---|---|---|---|---|---|---|
| | | OBJ.↑ | DROP↓ | TIME↓ | OBJ.↑ | DROP↓ | TIME↓ |
| Gurobi (Gurobi Optimization, 2023) | OR | 727.844* | 0.000±0.000% | 60.612s | 2936.886* | 0.000±0.000% | 300.214s |
| DiffUCO: CE (S=1,F=1) (Sanokowski et al., 2024) | GP | 726.900 | 0.146±0.483% | 0.197s | **2986.932** | **-1.688±0.480%** | 0.654s |
| COExpander (S=1,D$_s$=1,I$_s$=1) | GP | 702.376 | 3.504±1.841% | 0.014s | 2783.834 | 5.231±1.156% | 0.016s |
| COExpander (S=1,D$_s$=20,I$_s$=1) | AE | **727.526** | **0.058±0.419%** | 0.182s | **2978.200** | **-1.387±0.517%** | 0.204s |
| COExpander (S=1,D$_s$=1,I$_s$=20) | GP | 725.624 | 0.319±0.557% | 0.172s | 2948.112 | -0.369±0.580% | 0.194s |
| COExpander (S=1,D$_s$=5,I$_s$=20) | AE | 725.888 | 0.284±0.508% | 0.888s | 2952.520 | -0.518±0.554% | 0.992s |
| COExpander-FT(S=1,D$_s$=1,I$_s$=1) | GP | 718.592 | 1.281±0.765% | 0.014s | 2941.740 | -0.153±0.583% | 0.016s |
| COExpander-FT(S=1,D$_s$=20,I$_s$=1) | AE | 726.382 | 0.219±0.455% | 0.184s | 2975.712 | -1.304±0.511% | 0.202s |
| COExpander-FT(S=1,D$_s$=1,I$_s$=20) | GP | **726.538** | **0.195±0.476%** | 0.172s | **2980.508** | **-1.467±0.513%** | 0.196s |
| COExpander-FT(S=1,D$_s$=5,I$_s$=20) | AE | 726.524 | 0.196±0.495% | 0.884s | 2978.486 | -1.399±0.509% | 0.972s |
| DiffUCO: CE (S=4,F=1) (Sanokowski et al., 2024) | GP | 727.534 | 0.061±0.462% | 0.610s | **2989.458** | **-1.773±0.466%** | 2.701s |
| COExpander (S=4,D$_s$=1,I$_s$=1) | GP | 705.446 | 3.080±1.503% | 0.016s | 2795.804 | 4.821±1.051% | 0.024s |
| COExpander (S=4,D$_s$=20,I$_s$=1) | AE | 726.798 | 0.159±0.442% | 0.064s | **2961.500** | **-0.821±0.544%** | 0.114s |
| COExpander (S=4,D$_s$=1,I$_s$=20) | GP | **728.316** | **-0.049±0.371%** | 0.202s | 2960.664 | -0.797±0.494% | 0.362s |
| COExpander-FT(S=4,D$_s$=1,I$_s$=1) | GP | 720.240 | 1.056±0.666% | 0.016s | 2954.102 | -0.574±0.534% | 0.024s |
| COExpander-FT(S=4,D$_s$=20,I$_s$=1) | AE | 726.232 | 0.238±0.467% | 0.064s | 2980.652 | -1.475±0.477% | 0.114s |
| COExpander-FT(S=4,D$_s$=1,I$_s$=20) | GP | **728.272** | **-0.043±0.386%** | 0.196s | **2987.142** | **-1.693±0.474%** | 0.364s |

| METHOD | TYPE | BA-GIANT | | |
|---|---|---|---|---|
| | | OBJ.↑ | DROP↓ | TIME↓ |
| Gurobi (3600s) (Gurobi Optimization, 2023) | OR | 7217.900* | 0.000±0.000% | 3601.342s |
| Gurobi (300s) (Gurobi Optimization, 2023) | OR | 7217.860 | 0.001±0.003% | 300.504s |
| Gurobi (60s) (Gurobi Optimization, 2023) | OR | 7216.960 | 0.011±0.075% | 61.228s |
| DiffUCO: CE (S=1,F=1) (Sanokowski et al., 2024) | GP | 7384.020 | -2.306±0.326% | 2.480s |
| DiffUCO: CE (S=4,F=1) (Sanokowski et al., 2024) | GP | **7387.760** | **-2.358±0.325%** | 10.760s |
| COExpander (S=1,D$_s$=1,I$_s$=1) | GP | 6860.220 | 4.935±0.841% | 0.060s |
| COExpander (S=1,D$_s$=50,I$_s$=1) | AE | **7369.980** | **-2.111±0.330%** | 0.720s |
| COExpander (S=1,D$_s$=1,I$_s$=50) | GP | 7308.260 | -1.258±0.403% | 0.700s |
| COExpander (S=4,D$_s$=1,I$_s$=50) | GP | 7329.420 | -1.553±0.339% | 1.760s |
| COExpander-FT(S=1,D$_s$=1,I$_s$=1) | GP | 7264.660 | -0.656±0.448% | 0.060s |
| COExpander-FT(S=1,D$_s$=50,I$_s$=1) | AE | 7361.800 | -1.997±0.318% | 0.740s |
| COExpander-FT(S=1,D$_s$=1,I$_s$=50) | GP | 7372.100 | -2.134±0.330% | 0.720s |
| COExpander-FT(S=4,D$_s$=1,I$_s$=50) | GP | **7381.920** | **-2.276±0.327%** | 1.760s |

*Table 22.* Complete results on TSP.

| METHOD | TYPE | TSP-50 | | | TSP-100 | | |
|---|---|---|---|---|---|---|---|
| | | OBJ.↓ | DROP↓ | TIME↓ | OBJ.↓ | DROP↓ | TIME↓ |
| Concorde (Applegate et al., 2006) | Exact | 5.688* | 0.000±0.000% | 0.059s | 7.756* | 0.000±0.000% | 0.238s |
| LKH (500) (Helsgaun, 2017) | Heuristics | 5.688 | 0.001±0.014% | 0.058s | 7.756 | 0.001±0.014% | 0.176s |
| GA-EAX (Nagata & Kobayashi, 2013) | Heuristics | 5.688 | 0.000±0.001% | 0.101s | 7.756 | 0.000±0.004% | 1.862s |
| GCN + 2OPT (Joshi et al., 2019) | GP | 5.691 | 0.066±0.316% | 0.007s | 7.775 | 0.243±0.546% | 0.009s |
| GNNGLS + GLS (Hudson et al., 2021) | GP | 5.707 | 0.333±0.809% | 0.020s | 7.857 | 1.295±1.026% | 0.129s |
| DIMES + 2OPT (Qiu et al., 2022) | GP | 5.891 | 3.578±2.361% | 0.007s | 8.108 | 4.543±1.838% | 0.012s |
| DIFUSCO (S=1,$I_s$=50) + 2OPT (Sun & Yang, 2023) | GP | 5.693 | 0.099±0.307% | 2.587s | 7.776 | 0.265±0.463% | 2.691s |
| Fast-T2T (S=1,$I_s$=5) + 2OPT (Li et al., 2024) | GP | **5.689** | **0.022±0.083%** | 0.254s | **7.762** | **0.074±0.226%** | 0.252s |
| RL4CO (AM) + 2OPT (Berto et al., 2024) | LC | 5.733 | 0.799±0.932% | 0.171s | 7.902 | 1.887±1.117% | 0.247s |
| RL4CO (POMO) + 2OPT (Berto et al., 2024) | LC | 5.727 | 0.686±0.857% | 0.221s | 7.918 | 2.097±1.189% | 0.319s |
| RL4CO (SymNCO) + 2OPT (Berto et al., 2024) | LC | 5.726 | 0.680±0.807% | 0.169s | 7.891 | 1.746±1.081% | 0.315s |
| BQ-NCO + 2OPT (Drakulic et al., 2023) | LC | 5.795 | 1.894±1.902% | 0.205s | 7.893 | 1.772±1.281% | 0.387s |
| GLOP (W=35,$I_{10}$=5,$I_{20}$=10,$I_{50}$=30) (Ye et al., 2024b) | D&C | **5.704** | **0.287±0.495%** | 2.719s | **7.831** | **0.973±0.843%** | 2.777s |
| COEXpander (S=1,$D_s$=1,$I_s$=1) + 2OPT | GP | 5.691 | 0.065±0.344% | 0.007s | 7.773 | 0.223±0.529% | 0.009s |
| COEXpander (S=1,$D_s$=3,$I_s$=1) + 2OPT | AE | 5.689 | 0.029±0.145% | 0.018s | 7.765 | 0.124±0.361% | 0.021s |
| COEXpander (S=1,$D_s$=1,$I_s$=5) + 2OPT | GP | 5.689 | 0.029±0.119% | 0.030s | 7.762 | 0.078±0.197% | 0.031s |
| COEXpander (S=1,$D_s$=3,$I_s$=5) + 2OPT | AE | **5.689** | **0.018±0.059%** | 0.084s | **7.759** | **0.043±0.107%** | 0.091s |
| DIFUSCO (S=4,$I_s$=50) + 2OPT (Sun & Yang, 2023) | GP | 5.689 | 0.016±0.063% | 2.688s | 7.761 | 0.067±0.154% | 2.692s |
| Fast-T2T (S=4,$I_s$=5) + 2OPT (Li et al., 2024) | GP | **5.688** | **0.008±0.037%** | 0.256s | **7.757** | **0.017±0.058%** | 0.265s |
| BQ-NCO + Beam-16 + 2OPT (Drakulic et al., 2023) | LC | 5.785 | 1.723±1.837% | 0.239s | 7.871 | 1.484±1.229% | 0.901s |
| GLOP (W=140,$I_{10}$=5,$I_{20}$=10,$I_{50}$=30) (Ye et al., 2024b) | D&C | 5.697 | 0.169±0.394% | 2.722s | 7.817 | 0.783±0.719% | 2.969s |
| COEXpander (S=4,$D_s$=1,$I_s$=1) + 2OPT | GP | 5.691 | 0.051±0.275% | 0.009s | 7.768 | 0.151±0.377% | 0.021s |
| COEXpander (S=4,$D_s$=3,$I_s$=1) + 2OPT | AE | 5.689 | 0.020±0.136% | 0.023s | 7.762 | 0.076±0.244% | 0.046s |
| COEXpander (S=4,$D_s$=1,$I_s$=5) + 2OPT | GP | 5.688 | 0.006±0.038% | 0.031s | 7.757 | 0.017±0.053% | 0.065s |
| COEXpander (S=4,$D_s$=3,$I_s$=5) + 2OPT | AE | **5.688** | **0.005±0.030%** | 0.088s | **7.757** | **0.014±0.048%** | 0.180s |

| METHOD | TYPE | TSP-500 | | | TSP-1K | | |
|---|---|---|---|---|---|---|---|
| | | OBJ.↓ | DROP↓ | TIME↓ | OBJ.↓ | DROP↓ | TIME↓ |
| Concorde (Applegate et al., 2006) | Exact | 16.546* | 0.000±0.000% | 18.672s | 23.118* | 0.000±0.000% | 84.413s |
| LKH (500) (Helsgaun, 2017) | Heuristics | 16.546 | 0.002±0.010% | 1.848s | 23.119 | 0.005±0.011% | 4.641s |
| GA-EAX (Nagata & Kobayashi, 2013) | Heuristics | 16.546 | 0.001±0.004% | 1.857s | 23.118 | 0.000±0.001% | 17.544s |
| GCN + 2OPT (Joshi et al., 2019) | GP | 16.769 | 1.348±0.589% | 0.063s | 23.527 | 1.769±0.434% | 0.227s |
| DIMES + 2OPT (Qiu et al., 2022) | GP | 17.655 | 6.707±1.121% | 0.314s | 24.906 | 7.735±0.823% | 0.662s |
| DIFUSCO (S=1,$I_s$=50) + 2OPT (Sun & Yang, 2023) | GP | 16.810 | 1.593±0.728% | 2.781s | 23.540 | 1.826±0.540% | 3.422s |
| Fast-T2T (S=1,$I_s$=5) + 2OPT (Li et al., 2024) | GP | **16.696** | **0.907±0.573%** | 0.328s | **23.387** | **1.164±0.421%** | 0.953s |
| BQ-NCO + 2OPT (Drakulic et al., 2023) | LC | 16.838 | 1.766±0.625% | 2.454s | 23.647 | 2.287±0.486% | 5.722s |
| GLOP (W=1,$I_{20}$=5,$I_{50}$=25,$I_{100}$=20) (Ye et al., 2024b) | D&C | 17.196 | 3.932±0.707% | 5.796s | 24.200 | 4.682±0.603% | 5.967s |
| COEXpander (S=1,$D_s$=1,$I_s$=1) + 2OPT | GP | 16.750 | 1.233±0.687% | 0.055s | 23.481 | 1.571±0.412% | 0.211s |
| COEXpander (S=1,$D_s$=3,$I_s$=1) + 2OPT | AE | 16.684 | 0.837±0.512% | 0.070s | 23.421 | 1.309±0.387% | 0.273s |
| COEXpander (S=1,$D_s$=1,$I_s$=5) + 2OPT | GP | 16.690 | 0.869±0.528% | 0.102s | 23.423 | 1.319±0.387% | 0.328s |
| COEXpander (S=1,$D_s$=3,$I_s$=5) + 2OPT | AE | **16.626** | **0.487±0.388%** | 0.242s | **23.337** | **0.946±0.409%** | 0.703s |
| DIFUSCO (S=4,$I_s$=50) + 2OPT (Sun & Yang, 2023) | GP | 16.697 | 0.917±0.375% | 3.031s | 23.421 | 1.308±0.396% | 7.891s |
| Fast-T2T (S=4,$I_s$=5) + 2OPT (Li et al., 2024) | GP | **16.629** | **0.500±0.283%** | 0.469s | **23.289** | **0.740±0.257%** | 1.930s |
| BQ-NCO + Beam-16 + 2OPT (Drakulic et al., 2023) | LC | 16.765 | 1.326±0.497% | 4.016s | 23.513 | 1.707±0.360% | 10.336s |
| GLOP (W=10,$I_{20}$=5,$I_{50}$=25,$I_{100}$=20) (Ye et al., 2024b) | D&C | 17.094 | 3.313±0.608% | 5.836s | 24.085 | 4.184±0.453% | 6.056s |
| COEXpander (S=4,$D_s$=1,$I_s$=1) + 2OPT | GP | 16.698 | 0.920±0.506% | 0.180s | 23.428 | 1.343±0.318% | 0.719s |
| COEXpander (S=4,$D_s$=3,$I_s$=1) + 2OPT | AE | 16.644 | 0.594±0.341% | 0.219s | 23.359 | 1.042±0.318% | 0.945s |
| COEXpander (S=4,$D_s$=1,$I_s$=5) + 2OPT | GP | 16.627 | 0.490±0.294% | 0.320s | 23.339 | 0.954±0.214% | 1.141s |
| COEXpander (S=4,$D_s$=3,$I_s$=5) + 2OPT | AE | **16.587** | **0.251±0.193%** | 0.656s | **23.266** | **0.640±0.212%** | 2.430s |

| METHOD | TYPE | TSP-10K | | |
|---|---|---|---|---|
| | | OBJ.↓ | DROP↓ | TIME↓ |
| LKH (500) (Helsgaun, 2017) | Heuristics | 71.755* | 0.000±0.000% | 332.758s |
| DIFUSCO (S=1,$I_s$=50) + 2OPT (Sun & Yang, 2023) | GP | 73.955 | 3.066±0.197% | 55.438s |
| Fast-T2T (S=1,$I_s$=5) + 2OPT (Li et al., 2024) | GP | **72.900** | **1.595±0.086%** | 40.938s |
| GLOP (W=1,$I_{20}$=5,$I_{50}$=25,$I_{100}$=50) (Ye et al., 2024b) | D&C | 75.902 | 5.780±0.269% | 11.776s |
| COEXpander (S=1,$D_s$=1,$I_s$=5) + 2OPT | GP | 72.982 | 1.710±0.120% | 28.313s |
| COEXpander (S=1,$D_s$=3,$I_s$=5) + 2OPT | AE | **72.796** | **1.450±0.096%** | 29.495s |

*Table 23.* Complete results on ATSP.

| METHOD | TYPE | ATSP-50 | | | ATSP-100 | | |
|---|---|---|---|---|---|---|---|
| | | OBJ.↓ | DROP↓ | TIME↓ | OBJ.↓ | DROP↓ | TIME↓ |
| LKH (1000) (Helsgaun, 2017) | Heuristics | 1.5545* | 0.0000±0.0000% | 0.097s | 1.5660* | 0.0000±0.0000% | 0.238s |
| LKH (100) (Helsgaun, 2017) | Heuristics | 1.5545 | 0.0007±0.0139% | 0.065s | 1.5661 | 0.0013±0.0113% | 0.119s |
| MatNet (Kwon et al., 2021) | LC | **1.5752** | **1.3232±1.0733%** | 0.035s | **1.6171** | **3.2551±1.2724%** | 0.061s |
| GOAL (Drakulic et al., 2024) | LC | 1.6450 | 5.8473±3.0859% | 0.313s | 1.6382 | 4.6066±2.1008% | 0.638s |
| COExpander (S=1,$D_s$=1,$I_s$=1) | GP | 1.6623 | 6.9484±5.1644% | 0.006s | 1.6651 | 6.3286±3.3771% | 0.008s |
| COExpander (S=1,$D_s$=3,$I_s$=1) | AE | 1.6367 | 5.3044±4.2257% | 0.019s | 1.6455 | 5.0855±3.1454% | 0.023s |
| COExpander (S=1,$D_s$=1,$I_s$=5) | GP | 1.6019 | 3.0433±4.0207% | 0.029s | 1.6275 | 3.9351±2.3738% | 0.033s |
| COExpander (S=1,$D_s$=3,$I_s$=5) | AE | **1.5823** | **1.7901±2.9643%** | 0.086s | **1.6166** | **3.2326±2.1912%** | 0.099s |
| COExpander (S=1,$D_s$=1,$I_s$=1) + 2OPT | GP | 1.6135 | 3.8045±2.9147% | 0.007s | 1.6303 | 4.1041±2.1430% | 0.009s |
| COExpander (S=1,$D_s$=3,$I_s$=1) + 2OPT | AE | 1.6014 | 3.0289±2.5430% | 0.019s | 1.6186 | 3.3611±1.9963% | 0.024s |
| COExpander (S=1,$D_s$=1,$I_s$=5) + 2OPT | GP | 1.5797 | 1.6160±2.2942% | 0.029s | 1.6076 | 2.6648±1.6867% | 0.034s |
| COExpander (S=1,$D_s$=3,$I_s$=5) + 2OPT | AE | **1.5720** | **1.1253±1.8664%** | 0.086s | **1.6013** | **2.2576±1.5627%** | 0.099s |
| MatNet (×16) (Kwon et al., 2021) | LC | **1.5592** | **0.3001±0.4365%** | 0.037s | **1.5909** | **1.5823±0.7496%** | 0.067s |
| GOAL + Beam-16 (Drakulic et al., 2024) | LC | 1.6277 | 4.7351±2.7136% | 0.354s | 1.6212 | 3.5325±1.6100% | 0.913s |
| COExpander (S=4,$D_s$=1,$I_s$=1) | GP | 1.6541 | 6.4197±4.9830% | 0.008s | 1.6479 | 5.2313±2.9126% | 0.029s |
| COExpander (S=4,$D_s$=3,$I_s$=1) | AE | 1.6152 | 3.9176±3.4767% | 0.024s | 1.6207 | 3.4889±2.2488% | 0.103s |
| COExpander (S=4,$D_s$=1,$I_s$=5) | GP | 1.5620 | 0.4850±1.4123% | 0.033s | 1.5946 | 1.8267±1.6501% | 0.161s |
| COExpander (S=4,$D_s$=3,$I_s$=5) | AE | **1.5587** | **0.2688±1.0225%** | 0.101s | **1.5878** | **1.3929±1.4938%** | 0.511s |
| COExpander (S=4,$D_s$=1,$I_s$=1) + 2OPT | GP | 1.6088 | 3.5017±2.8542% | 0.009s | 1.6192 | 3.3956±1.9178% | 0.032s |
| COExpander (S=4,$D_s$=3,$I_s$=1) + 2OPT | AE | 1.5890 | 2.2313±2.1856% | 0.025s | 1.6026 | 2.3398±1.5522% | 0.101s |
| COExpander (S=4,$D_s$=1,$I_s$=5) + 2OPT | GP | 1.5583 | 0.2414±0.7138% | 0.034s | 1.5842 | 1.1596±1.1187% | 0.167s |
| COExpander (S=4,$D_s$=3,$I_s$=5) + 2OPT | AE | **1.5572** | **0.1709±0.6111%** | 0.101s | **1.5808** | **0.9464±1.0335%** | 0.516s |

| METHOD | TYPE | ATSP-200 | | | ATSP-500 | | |
|---|---|---|---|---|---|---|---|
| | | OBJ.↓ | DROP↓ | TIME↓ | OBJ.↓ | DROP↓ | TIME↓ |
| LKH (1000) (Helsgaun, 2017) | Heuristics | 1.5647* | 0.0000±0.0000% | 0.724s | 1.5734* | 0.0000±0.0000% | 4.376s |
| LKH (100) (Helsgaun, 2017) | Heuristics | 1.5647 | 0.0011±0.0056% | 0.244s | 1.5735 | 0.0010±0.0039% | 1.022s |
| MatNet (Kwon et al., 2021) | LC | 3.8307 | 145.0832±8.6424% | 0.114s | – | – | – |
| GOAL (Drakulic et al., 2024) | LC | **1.6170** | **3.3481±1.2360%** | 1.412s | **1.7152** | **9.0157±1.3033%** | 4.056s |
| GLOP + MatNet (Ye et al., 2024b) | D&C | 2.0222 | 29.3824±21.4434% | 0.143s | 2.3353 | 48.4783±14.5705% | 0.352s |
| COExpander (S=1,$D_s$=1,$I_s$=1) | GP | 1.6785 | 7.2801±2.4748% | 0.037s | 1.6747 | 6.4434±1.4536% | 0.127s |
| COExpander (S=1,$D_s$=3,$I_s$=1) | AE | 1.6284 | 4.0780±2.0543% | 0.099s | 1.6463 | 4.6344±1.7081% | 0.403s |
| COExpander (S=1,$D_s$=1,$I_s$=5) | GP | 1.6381 | 4.6773±2.0655% | 0.162s | 1.6410 | 4.2941±1.6633% | 0.558s |
| COExpander (S=1,$D_s$=3,$I_s$=5) | AE | **1.6146** | **3.2001±1.4612%** | 0.547s | **1.6228** | **3.1280±1.4024%** | 1.669s |
| COExpander (S=1,$D_s$=1,$I_s$=1) + 2OPT | GP | 1.6524 | 5.6083±1.9575% | 0.046s | 1.6534 | 5.0880±1.0975% | 0.231s |
| COExpander (S=1,$D_s$=3,$I_s$=1) + 2OPT | AE | 1.6161 | 3.2926±1.5259% | 0.111s | 1.6299 | 3.5915±1.1390% | 0.478s |
| COExpander (S=1,$D_s$=1,$I_s$=5) + 2OPT | GP | 1.6154 | 3.2517±1.4556% | 0.168s | 1.6218 | 3.0776±1.0087% | 0.648s |
| COExpander (S=1,$D_s$=3,$I_s$=5) + 2OPT | AE | **1.6047** | **2.5634±1.1579%** | 0.517s | **1.6114** | **2.4080±0.8883%** | 1.766s |
| MatNet (×16) (Kwon et al., 2021) | LC | 3.7263 | 138.4024±7.6532% | 0.156s | – | – | – |
| GOAL + Beam-16 (Drakulic et al., 2024) | LC | **1.6093** | **2.8627±1.0017%** | 4.690s | **1.7012** | **8.1275±1.1215%** | 32.239s |
| COExpander (S=4,$D_s$=1,$I_s$=1) | GP | 1.6632 | 6.2877±2.0898% | 0.087s | 1.6520 | 4.9994±1.1873% | 0.497s |
| COExpander (S=4,$D_s$=3,$I_s$=1) | AE | 1.6085 | 2.7981±1.1689% | 0.278s | 1.6233 | 3.1669±0.8271% | 1.492s |
| COExpander (S=4,$D_s$=1,$I_s$=5) | GP | 1.6083 | 2.7717±1.0595% | 0.360s | 1.6135 | 2.5429±0.8289% | 2.144s |
| COExpander (S=4,$D_s$=3,$I_s$=5) | AE | 1.5928 | 1.7904±0.9533% | 1.073s | **1.6044** | **1.9648±0.6598%** | 6.267s |
| COExpander (S=4,$D_s$=1,$I_s$=1) + 2OPT | GP | 1.6407 | 4.8634±1.7464% | 0.124s | 1.6364 | 4.0057±0.9356% | 0.922s |
| COExpander (S=4,$D_s$=3,$I_s$=1) + 2OPT | AE | 1.6015 | 2.3567±0.9981% | 0.288s | 1.6148 | 2.6292±0.7023% | 1.807s |
| COExpander (S=4,$D_s$=1,$I_s$=5) + 2OPT | GP | 1.5974 | 2.0943±0.8417% | 0.391s | 1.6046 | 1.9800±0.6106% | 2.484s |
| COExpander (S=4,$D_s$=3,$I_s$=5) + 2OPT | AE | **1.5882** | **1.5011±0.8053%** | 1.097s | **1.5981** | **1.5681±0.4867%** | 6.597s |

*Table 24.* Complete results on individual instances from TSPLIB. "Orig.": solved tour length w.r.t the original distance of each instance. "Norm.": tour length calculated on normalized distances for clearer comparison.

| INSTANCE | CONCORDE SOLVER | | COEXPANDER | | | |
|---|---|---|---|---|---|---|
| | Length(Orig.) | Length(Norm.) | Used Model | Length(Orig.) | Length(Norm.) | Drop |
| eil51 | 428.872 | 6.701 | Uniform-100 | 428.872 | 6.701 | 0.000% |
| berlin52 | 7544.366 | 4.348 | Uniform-100 | 7544.366 | 4.348 | 0.000% |
| st70 | 677.110 | 6.839 | Uniform-100 | 677.110 | 6.839 | 0.000% |
| eil76 | 544.369 | 7.561 | Uniform-100 | 544.369 | 7.561 | 0.000% |
| pr76 | 108159.438 | 5.518 | Uniform-100 | 108159.438 | 5.518 | 0.000% |
| rat99 | 1219.244 | 5.671 | Uniform-500 | 1219.244 | 5.671 | 0.000% |
| kroA100 | 21285.443 | 5.408 | Uniform-500 | 21285.443 | 5.408 | 0.000% |
| kroB100 | 22139.075 | 5.626 | Uniform-500 | 22303.429 | 5.668 | 0.742% |
| kroC100 | 20750.763 | 5.276 | Uniform-500 | 20820.370 | 5.294 | 0.335% |
| kroD100 | 21294.291 | 5.431 | Uniform-500 | 21294.291 | 5.431 | 0.000% |
| kroE100 | 22068.759 | 5.557 | Uniform-500 | 22068.759 | 5.557 | 0.000% |
| rd100 | 7910.396 | 8.065 | Uniform-500 | 7910.396 | 8.065 | 0.000% |
| eil101 | 640.212 | 8.536 | Uniform-500 | 640.212 | 8.536 | 0.000% |
| lin105 | 14382.996 | 4.756 | Uniform-500 | 14382.996 | 4.756 | 0.000% |
| pr107 | 44301.684 | 3.778 | Uniform-500 | 44346.187 | 3.782 | 0.100% |
| pr124 | 59030.736 | 5.009 | Uniform-500 | 59492.914 | 5.048 | 0.783% |
| bier127 | 118293.524 | 6.106 | Uniform-500 | 118293.524 | 6.106 | 0.000% |
| ch130 | 6110.722 | 8.751 | Uniform-500 | 6110.722 | 8.751 | 0.000% |
| pr136 | 96770.924 | 7.767 | Uniform-500 | 97348.263 | 7.813 | 0.597% |
| pr144 | 58535.222 | 4.538 | Uniform-500 | 58620.693 | 4.544 | 0.146% |
| ch150 | 6530.903 | 9.342 | Uniform-500 | 6530.903 | 9.342 | 0.000% |
| kroA150 | 26524.863 | 6.693 | Uniform-500 | 26761.764 | 6.753 | 0.893% |
| kroB150 | 26127.358 | 6.635 | Uniform-500 | 26153.602 | 6.641 | 0.100% |
| pr152 | 73683.641 | 5.234 | Uniform-500 | 73824.720 | 5.244 | 0.191% |
| u159 | 42075.670 | 6.574 | Uniform-500 | 42075.670 | 6.574 | 0.000% |
| rat195 | 2333.873 | 7.993 | Uniform-500 | 2343.642 | 8.026 | 0.419% |
| d198 | 15808.652 | 3.924 | Uniform-500 | 15885.773 | 3.944 | 0.488% |
| kroA200 | 29369.407 | 7.437 | Uniform-500 | 29385.727 | 7.441 | 0.056% |
| kroB200 | 29440.412 | 7.467 | Uniform-500 | 29691.071 | 7.530 | 0.851% |
| ts225 | 126645.934 | 10.554 | Uniform-500 | 128058.222 | 10.672 | 1.115% |
| tsp225 | 3859.000 | 7.895 | Uniform-500 | 3866.710 | 7.911 | 0.200% |
| pr226 | 80370.257 | 5.279 | Uniform-500 | 80813.061 | 5.308 | 0.551% |
| gil262 | 2385.804 | 12.050 | Uniform-500 | 2389.772 | 12.070 | 0.166% |
| pr264 | 49135.005 | 6.200 | Uniform-500 | 49135.005 | 6.200 | 0.000% |
| a280 | 2586.770 | 9.238 | Uniform-500 | 2586.770 | 9.238 | 0.000% |
| pr299 | 48194.920 | 6.638 | Uniform-500 | 48823.925 | 6.725 | 1.305% |
| lin318 | 42042.535 | 10.170 | Uniform-500 | 42053.026 | 10.172 | 0.025% |
| rd400 | 15275.985 | 15.344 | Uniform-500 | 15291.238 | 15.359 | 0.100% |
| fl417 | 11914.309 | 6.286 | Uniform-500 | 12044.736 | 6.354 | 1.095% |
| pr439 | 107215.302 | 8.991 | Uniform-500 | 108096.965 | 9.065 | 0.822% |
| pcb442 | 50783.548 | 13.364 | Uniform-500 | 51020.753 | 13.427 | 0.467% |
| d493 | 35018.526 | 9.350 | Uniform-500 | 35446.905 | 9.465 | 1.223% |
| u574 | 36934.771 | 12.023 | Uniform-500 | 37320.427 | 12.149 | 1.044% |
| rat575 | 6795.968 | 13.619 | Uniform-500 | 6830.574 | 13.689 | 0.509% |
| p654 | 34646.835 | 7.196 | Uniform-1000 | 34894.068 | 7.247 | 0.714% |
| d657 | 48915.630 | 12.212 | Uniform-1000 | 49246.478 | 12.294 | 0.676% |
| u724 | 41907.728 | 14.437 | Uniform-1000 | 42384.083 | 14.601 | 1.137% |
| rat783 | 8842.995 | 15.247 | Uniform-1000 | 8887.347 | 15.323 | 0.502% |
| pr1002 | 259066.663 | 16.397 | Uniform-1000 | 260668.540 | 16.498 | 0.618% |
| *mean* | – | **8.062** | – | – | **8.095** | **0.367%** |

*Table 25.* **COExpander v.s. Previous SOTA: A Summarized Comparison**. * denotes ultra-large datasets.

| PROBLEM | DATASET | PREVIOUS SOTA OR GUROBI | | | | COEXPANDER | |
|---|---|---|---|---|---|---|---|
| | | METHOD | TYPE | DROP↓ | TIME↓ | DROP↓ | TIME↓ |
| MIS | RB-SMALL | Fast-T2T (Li et al., 2024) | GP | 1.700% | 0.860s | 1.482% | 0.652s |
| MIS | RB-LARGE | DiffUCO (Sanokowski et al., 2024) | GP | 7.944% | 25.479s | 3.582% | 1.298s |
| MIS | ER-700-800 | Fast-T2T (Li et al., 2024) | GP | 7.169% | 1.961s | 5.343% | 0.703s |
| MIS | SATLIB | DIFUSCO (Sun & Yang, 2023) | GP | 0.200% | 2.962s | 0.160% | 1.056s |
| MIS* | ER-1400-1600 | Fast-T2T (Li et al., 2024) | GP | 25.417% | 6.961s | 4.911% | 4.516s |
| MIS* | ER-1400-1600 | Gurobi (60s) (Gurobi Optimization, 2023) | OR | 19.856% | 62.141s | 4.911% | 4.516s |
| MIS* | RB-GIANT | DiffUCO (Sanokowski et al., 2024) | GP | 14.352% | 339.010s | 5.221% | 14.660s |
| MIS* | RB-GIANT | Gurobi (60s) (Gurobi Optimization, 2023) | OR | 8.806% | 56.010s | 5.221% | 14.660s |
| MCl | RB-SMALL | Meta-EGN (Wang & Li, 2023) | GP | 8.300% | 0.270s | 0.434% | 0.362s |
| MCl | RB-LARGE | Meta-EGN (Wang & Li, 2023) | GP | 15.487% | 0.542s | 0.898% | 0.368s |
| MCl* | RB-GIANT | Gurobi (60s) (Gurobi Optimization, 2023) | OR | 33.917% | 60.248s | -6.424% | 1.520s |
| MVC | RB-SMALL | Meta-EGN (Wang & Li, 2023) | GP | 1.563% | 0.297s | 0.290% | 0.086s |
| MVC | RB-LARGE | Meta-EGN (Wang & Li, 2023) | GP | 4.400% | 1.025s | 0.163% | 0.515s |
| MVC* | RB-GIANT | Gurobi (60s) (Gurobi Optimization, 2023) | OR | 0.162% | 58.121s | 0.149% | 8.200s |
| MCut | BA-SMALL | DiffUCO (Sanokowski et al., 2024) | GP | 0.061% | 0.610s | -0.043% | 0.202s |
| MCut | BA-LARGE | DiffUCO (Sanokowski et al., 2024) | GP | -1.773% | 2.701s | -1.693% | 0.364s |
| MCut* | BA-GIANT | DiffUCO (Sanokowski et al., 2024) | GP | -2.358% | 10.760s | -2.276% | 1.760s |
| MCut* | BA-GIANT | Gurobi (60s) (Gurobi Optimization, 2023) | OR | 0.011% | 61.228s | -2.276% | 1.760s |
| TSP | TSP-50 | Fast-T2T (Li et al., 2024) | GP | 0.008% | 0.256s | 0.005% | 0.088s |
| TSP | TSP-50 | RL4CO (SymNCO) (Berto et al., 2024) | LC | 0.680% | 0.169s | 0.005% | 0.088s |
| TSP | TSP-100 | Fast-T2T (Li et al., 2024) | GP | 0.017% | 0.265s | 0.014% | 0.180s |
| TSP | TSP-100 | BQ-NCO (Drakulic et al., 2023) | LC | 1.484% | 0.901s | 0.014% | 0.180s |
| TSP | TSP-500 | Fast-T2T (Li et al., 2024) | GP | 0.500% | 0.469s | 0.251% | 0.656s |
| TSP | TSP-500 | BQ-NCO (Drakulic et al., 2023) | LC | 1.326% | 4.016s | 0.251% | 0.656s |
| TSP | TSP-1K | Fast-T2T (Li et al., 2024) | GP | 0.740% | 1.930s | 0.640% | 2.430s |
| TSP | TSP-1K | BQ-NCO (Drakulic et al., 2023) | LC | 1.707% | 10.336s | 0.640% | 2.430s |
| TSP* | TSP-10K | Fast-T2T (Li et al., 2024) | GP | 1.595% | 40.938s | 1.450% | 29.495s |
| ATSP | ATSP-50 | MatNet (Kwon et al., 2021) | LC | 0.3001% | 0.037s | 0.1709% | 0.101s |
| ATSP | ATSP-100 | MatNet (Kwon et al., 2021) | LC | 1.5823% | 0.067s | 0.9464% | 0.516s |
| ATSP | ATSP-200 | GOAL (Drakulic et al., 2024) | LC | 2.8627% | 4.690s | 1.5011% | 1.097s |
| ATSP | ATSP-500 | GOAL (Drakulic et al., 2024) | LC | 8.1275% | 32.239s | 1.5681% | 6.597s |
| *Summary of Main Results* | | | | | | | |
| MIS | MAIN | Global Prediction Solvers | GP | 4.253% | 7.816s | 2.642% | 0.927s |
| MCl | MAIN | Global Prediction Solvers | GP | 11.894% | 0.406s | 0.666% | 0.365s |
| MVC | MAIN | Global Prediction Solvers | GP | 2.982% | 0.661s | 0.227% | 0.301s |
| MCut | MAIN | Global Prediction Solvers | GP | -0.856% | 1.656s | -0.868% | 0.283s |
| TSP | MAIN | Global Prediction Solvers | GP | 0.316% | 0.729s | 0.228% | 0.839s |
| TSP | MAIN | Local Construction Solvers | LC | 1.299% | 3.856s | 0.228% | 0.839s |
| ATSP | MAIN | Local Construction Solvers | LC | 4.253% | 7.816s | 1.047% | 2.078s |
| **Average** | **MAIN** | **Previous SOTA NCO Solvers** | **–** | **3.807%** | **3.181s** | **0.657%** | **0.799s** |
| *Summary of Ultra-Large Scale Results* | | | | | | | |
| MIS* | Ultra-Large | Global Prediction Solvers | GP | 19.885% | 172.986s | 5.066% | 9.588s |
| MIS* | Ultra-Large | Gurobi (60s) (Gurobi Optimization, 2023) | OR | 14.331% | 59.076s | 5.066% | 9.588s |
| MCl* | Ultra-Large | Gurobi (60s) (Gurobi Optimization, 2023) | OR | 33.917% | 60.248s | -6.424% | 1.520s |
| MVC* | Ultra-Large | Gurobi (60s) (Gurobi Optimization, 2023) | OR | 0.162% | 28.121s | 0.149% | 8.200s |
| MCut* | Ultra-Large | Global Prediction Solvers | GP | -2.358% | 10.760s | -2.276% | 1.760s |
| MCut* | Ultra-Large | Gurobi (60s) (Gurobi Optimization, 2023) | OR | 0.011% | 61.228s | -2.276% | 1.760s |
| TSP* | Ultra-Large | Global Prediction Solvers | GP | 1.595% | 40.938s | 1.450% | 29.495s |

# H. Discussion of Limitation

## H.1. Limitation of Training Methods and Generalization

COExpander currently relies on supervised learning (SL), which inevitably requires a certain amount of labeled data (details are presented in Appendix E). However, it is worth noting that the traditional solvers we re-wrapped (Appendix D) support parallel processing on CPUs. Therefore, the time required to generate each training dataset ranges from several hours to tens of hours, which is an acceptable time cost.

Although we have conducted some generalization experiments across models, sizes, and distributions in Appendix G, there is still a lack of generalization experiments on problems such as TSP and ATSP under other distributions (e.g., gaussian, cluster), as well as on problems such as MIS, MCl, MVC, and MCut under other distributions (e.g., Holme&Kim (HK) (Holme & Kim, 2002), Watts&Strogatz (WS) (Watts & Strogatz, 1998)). In future work, we will complete the experiments and analysis in this regard.

*Table 26.* Complete Results on CVRP.

| Method | Type | CVRP-50 | | | CVRP-100 | | |
|---|---|---|---|---|---|---|---|
| | | Obj.↓ | Drop↓ | Time↓ | Obj.↓ | Drop↓ | Time↓ |
| HGS (Vidal et al., 2012) | Heuristics | 10.489* | 0.000±0.000% | 1.005s | 15.563* | 0.000±0.000% | 20.027s |
| RL4CO (Sym-NCO) (Berto et al., 2024) | LC | 10.769 | 3.891±1.640% | 0.087s | 16.220 | 4.241±1.131% | 0.166s |
| RL4CO (Sym-NCO) (Berto et al., 2024) + Classic-LS | LC | **10.565** | **1.910±1.435%** | 0.091s | **15.933** | **2.379±1.001%** | 0.173s |
| COExpander (S=1,$D_s$=1,$I_s$=1) | GP | 12.640 | 21.835±6.535% | 0.009s | 19.202 | 23.333±5.155% | 0.010s |
| COExpander (S=1,$D_s$=3,$I_s$=1) | AE | 11.979 | 15.407±6.898% | 0.033s | 17.497 | 12.343±5.459% | 0.047s |
| COExpander (S=1,$D_s$=1,$I_s$=1) + Classic-LS | GP | 10.871 | 4.836±5.552% | 0.013s | 16.294 | 4.698±3.120% | 0.018s |
| COExpander (S=1,$D_s$=3,$I_s$=1) + Classic-LS | AE | 10.773 | 3.903±4.052% | 0.037s | 16.224 | 4.253±1.991% | 0.055s |

| Method | Type | CVRP-200 | | | CVRP-500 | | |
|---|---|---|---|---|---|---|---|
| | | Obj.↓ | Drop↓ | Time↓ | Obj.↓ | Drop↓ | Time↓ |
| HGS (Vidal et al., 2012) | Heuristics | 19.630* | 0.000±0.000% | 60.024s | 37.154* | 0.000±0.000% | 360.376s |
| RL4CO (Sym-NCO) (Berto et al., 2024) | LC | 20.662 | 5.274±0.926% | 0.320s | 40.382 | 8.723±0.810% | 0.769s |
| RL4CO (Sym-NCO) (Berto et al., 2024) + Classic-LS | LC | **20.193** | **2.880±0.854%** | 0.341s | **38.700** | **4.173±0.606%** | 0.883s |
| COExpander (S=1,$D_s$=1,$I_s$=1) | GP | 25.064 | 27.616±5.095% | 0.059s | 47.749 | 28.509±2.920% | 0.091s |
| COExpander (S=1,$D_s$=3,$I_s$=1) | AE | 22.402 | 13.977±4.072% | 0.145s | 43.901 | 18.199±2.834% | 0.554s |
| COExpander (S=1,$D_s$=1,$I_s$=1) + Classic-LS | GP | 20.662 | 5.290±1.471% | 0.063s | 39.195 | 5.530±0.937% | 0.215s |
| COExpander (S=1,$D_s$=3,$I_s$=1) + Classic-LS | AE | 20.587 | 4.893±1.344% | 0.153s | 39.121 | 5.337±0.938% | 0.605s |

## H.2. Limitation of Resolving Complex-Constraint Problems

COExpander is the result of combining AE and CM, so its scope of application is limited by its global predictor, i.g. consistency model. To our best knowledge, before our work, consistency model (Li et al., 2024) has only been applied to two combinatorial optimization problems: TSP and MIS. At present, there is no consistency model (and even from a higher level, no global prediction solvers) that can solve combinatorial optimization problems with relatively complex constraints, such as CVRP (with demand constraints).

We have attempted to apply COExpander to **CVRP-50**, **CVRP-100**, **CVRP-200** and **CVRP-500**. Following GOAL (Drakulic et al., 2024) and NeuOpt-GIRE (Ma et al., 2023), the coordinates of the depot and clients were sampled from a uniform distribution over the unit square, consistent with the TSP setting. The demands of clients are randomly generated integers from the interval [1, 10], and the vehicle capacity is set to 40 / 50 / 80 / 100 for *CVRP-50* to *CVRP-500*, respectively. During the training phase, we use HGS (Vidal et al., 2012) to generate 1,280,000 / 640,000 / 32,000 / 12,800 samples for *CVRP-50* to *CVRP-500*, respectively, and then each model has been trained with a total of 10 training epochs. In the testing phase, for *CVRP-50* and *CVRP-100*, consistent with NeuOpt-GIRE, each test dataset has 10,000 samples. For *CVRP-200* and *CVRP-500*, we generated 100 samples each as the test set. The results are shown in Table 26.

From the experimental results, although the GP solver, i.g., COExpander (S=1, $D_s$=1, $I_s$=1) has certain advantages in speed, it is far inferior to the LC solver (we take Sym-NCO (Berto et al., 2024) as a representative of the LC solver) in terms of effectiveness. After we apply Adaptive Expansion to the GP solver for solving CVRP (determining partial complete sub-tours in each determination process to ensure the satisfiability of constraints), the performance has been improved to some extent, but it is still not satisfactory. Furthermore, we consider introducing post-processing. Following previous work (Prins, 2004), we have implemented the local search method (*Classic-LS*) for CVRP, leading to great performance

improvements for solvers of the three paradigms. Of course, we believe that solving the CVRP in this manner is far from sufficient. Therefore, we will discuss in detail in the next subsection the possibility of using the AE paradigm to solve complex-constraint problems such as the CVRP.

### H.3. Discussion on How Adaptive Expansion solves Complex-Constraint Problems

**Overview.** Since the AE paradigm is an iterative process where the future state depends only on the current state, i.e., satisfying the Markov property, it can naturally be modeled as a Markov Decision Process (MDP). Also, AE makes determinations on variables during the intermediate steps, which allows it to handle complex constraints through the determination process. Furthermore, inspired by LwD (Ahn et al., 2020), we plan to employ Proximal Policy Optimization (PPO, Schulman et al. (2017)) for model training, in order to better adapt to the relationship between actions and constraints, thereby extending the capability of AE methods on complex-constrained tasks. **Note that a feasible scheme is theoretically described as follows, while the implementation and empirical results have obviously exceeded the scope of this paper, hence reasonably remitted to future researches.**

**Modeling.** An MDP is defined by $(\mathcal{S}, \mathcal{A}, \mathcal{P}, \mathcal{R})$, where $\mathbf{s} \in \mathcal{S}$ denotes a state in the state space $\mathcal{S}$, $\mathbf{a} \in \mathcal{A}$ represents an action in the action space $\mathcal{A}$, $\mathcal{P}(\mathbf{s}_{t+1}|\mathbf{s}_t, \mathbf{a}_t)$ is the state transition distribution, and $\mathcal{R}(\mathbf{s}_t, \mathbf{a}_t)$ is the reward function. Given graph data $G$, cost function $c(\cdot, \cdot)$, we modeling the COPs on graph $G$ using AE paradigm $\text{VDP}(N, k)$ as follows:

- **State.** Every state $\mathbf{s}$ can be represented as a tuple of the current partial solution $\mathbf{x}$ and the mask $\mathcal{M}$, i.e., $\mathbf{s}_t := (\mathbf{x}_t, \mathcal{M}_t)$.
- **Action.** An action $\mathbf{a}$ can be divided into two parts: selection $m$ and assignment $z$, i.e., $\mathbf{a}_t := (m_t, z_t)$. Selection is to determine which decision variables will be fixed, while assignment is to assign values to these decision variables.
- **Transition.** Given two consecutive states $\mathbf{s}_t, \mathbf{s}_{t+1}$ and the corresponding action $\mathbf{a}_t$, the transition $\mathcal{P}(\mathbf{s}_{t+1}|\mathbf{s}_t, \mathbf{a}_t)$ is also divided into two steps as below, where $\vee$ denotes " logic or ".

$$\text{Update Partial Solution: } \mathbf{x}_{t+1}^i = \begin{cases} z_t^i & \text{if } m_t^i = 1 \\ \mathbf{x}_t^i & \text{if } m_t^i = 0 \end{cases}, \quad \text{Update Mask: } \mathcal{M}_{t+1} = \mathcal{M}_t \vee m_t;$$

- **Reward.** We define the reward function only in the final step, i.e., $\mathcal{R}(\mathbf{s}_t, \mathbf{a}_t) := \begin{cases} -c(G, \mathbf{x}_t) & \text{if } t = k-1, \\ 0 & \text{otherwise} \end{cases}$

**Training Model with PPO.** The model $f(\cdot, \cdot)$ takes the current state $\mathbf{s}_t$, graph data $G$ as inputs, and outputs a policy distribution $\pi_\theta^t := \pi_\theta(\mathbf{a}_t|\mathbf{s}_t, G) = f(G, \mathbf{s_t})$. And the assignment part $z_t$ of action $\mathbf{a}_t$ is sampled from $\pi_\theta^t$ via Bernoulli sampling and then set the determined variables as 0:

$$z_t = \text{Bernoulli}(\pi_\theta^t); \quad z_t^i = \begin{cases} z_t^i & \text{if } \mathcal{M}_t^i = 0 \\ 0 & \text{if } \mathcal{M}_t^i = 1 \end{cases}.$$

The training goal is to maximize the expected cumulative reward over the episode: $\max_\theta \mathbb{E}_{\pi_\theta} \left[ \sum_{t=0}^{k-1} \mathcal{R}(\mathbf{s}_t, \mathbf{a}_t) \right] = \max_\theta \mathbb{E}_{\pi_\theta} [-c(G, \mathbf{x}_{k-1})]$. To achieve this objective, we use PPO algorithm, which updates the policy parameters $\theta$ by minimizing the following loss function:

$$\mathcal{L}(\theta) = \mathbb{E}_t \left[ \min \left( \frac{\pi_\theta^t}{\pi_{\theta_{\text{old}}}^t} \hat{A}_t, \text{ clip} \left( \frac{\pi_\theta^t}{\pi_{\theta_{\text{old}}}^t}, 1 - \epsilon, 1 + \epsilon \right) \hat{A}_t \right) \right],$$

where $\pi_{\theta_{\text{old}}}$ is the old policy before the update; $\epsilon$ is a hyperparameter that controls the clipping range to prevent large policy updates; $\hat{A}_t$ is the estimated advantage function. In PPO, $\hat{A}_t = Q_t - V_t$. Given the discount factor $\gamma$, the action-value function is defined as $Q_t = -\gamma^{k-1-t} \cdot c(G, \mathbf{x}_{k-1})$, and the state-value function is defined as $V_t = \mathbb{E}_\theta [-c(G, \mathbf{x}_{k-1}|\mathbf{s}_t)]$.

**The Selection Part of Action.** The selection part $m$ of $\mathbf{a}$ is the core of controlling constraint satisfaction, and we need to design it specifically for the particular problem. We will take CVRP as an example to illustrate how to design $m$.

- **Symmetry.** The dimensions of $m$, $z$, and $\mathcal{M}$ can all be reshaped into $n \times n$, where $n$ is the number of nodes including the depot. Note that due to the symmetry of the CVRP problem, $z$ needs to undergo an OR operation with its transpose.
- **Pre-assignment.** *pre-assignment* is defined as $\hat{\mathbf{x}}_{i,j} = \begin{cases} z_{i,j} & \text{if } \mathcal{M}_{i,j} = 0, \\ \mathbf{x}_{i,j} & \text{if } \mathcal{M}_{i,j} = 1 \end{cases}.$

- **Check Constraint.** To check constraint, we need first to construct the vector $\mathbf{C} = \{0\}^n$ to indicate the two constraints that need to be checked in CVRP: the node degree constraint and the vehicle capacity constraint. As for the first constraint, we check sequentially whether the degree of each node is less than or equal to 2. And for the capacity constraint, we transform the adjacency matrix $\hat{\mathbf{x}}$ into a partition matrix $\hat{\mathbf{p}}$, and then we sequentially check whether the sub-path containing each node satisfies the capacity constraint. The whole process can be represented as below, where $\mathbf{d}_j$ is the normalized demand of the $j$-th node.

$$\mathbf{C}_i = \begin{cases} 1 & \text{if } \sum_{j=1}^n \hat{\mathbf{x}}_{i,j} > 2 \text{ or } \sum_{j=1}^n \mathbf{p}_{i,j} \cdot \mathbf{d}_j > 1 \\ \mathbf{C}_i & \text{otherwise} \end{cases}$$

- **Actual Assignment.** After checking the constraints, we determine the actual assignment $\hat{\mathbf{z}}$ based on the indicator vector $\mathbf{C}$ and $z$, i.e., $\hat{z}_{i,j} = \begin{cases} 0 & \text{if } \mathbf{C}_i = 1, \\ z_{i,j} & \text{otherwise} \end{cases}$. And we perform the *pre-assignment* again, $\hat{\mathbf{x}}_{i,j} = \begin{cases} \hat{z}_{i,j} & \text{if } \mathcal{M}_{i,j} = 0 \\ \mathbf{x}_{i,j} & \text{if } \mathcal{M}_{i,j} = 1 \end{cases}$.

- **Selection Part.** Based on the above operations, we can naturally obtain $m_{i,j} = \begin{cases} 1 & \text{if } \hat{z}_{i,j} = 1 \text{ or } \sum_{j=1}^n \hat{\mathbf{x}}_{i,j} = 2 \\ 0 & \text{otherwise} \end{cases}$.

