# OpenReview forum: "COExpander: Adaptive Solution Expansion for Combinatorial Optimization"
_ICML.cc/2025/Conference — ICML 2025 poster_

### Official Review · Reviewer_F5eW · 2025-03-09

**Overall Recommendation:** 4

**Summary:**

This article proposes an Adaptive Expansion (AE) paradigm for COPs, demonstrating its advantages in a series of experiments on COP problems.

## update after rebuttal

Thanks for your detailed response. All of my concerns are clearly resolved so I have raised my score to 4.

**Claims And Evidence:**

Generally clear and convincing.

**Essential References Not Discussed:**

Generally okay.

**Experimental Designs Or Analyses:**

I notice that similar to this article, a recent D&C method, UDC, also uses GNN for initial tour construction (which does not seem entirely as described in Figure 4).  Considering that your proposed COExpander requires an optimal solution as the label, can COExpander demonstrate outstanding performance compared to UDC under a similar runtime?

**Methods And Evaluation Criteria:**

Yes, make sense.

**Other Comments Or Suggestions:**

See Questions.

**Other Strengths And Weaknesses:**

**Strength:**

1. The experimental volume and workload of this article are large, and the literature review is relatively complete
2. COExpander can demonstrate good results

**Weakness:**

1. I think I have tried my best to understand, but I still cannot fully comprehend the methods proposed in this article. This article lacks a simple and clear description of the method design and intuition (e.g., what effects have been made after predicting mask and using partial construction for MIS or TSP). I will ask some of my doubts in the Question part.

**Questions For Authors:**

1. Is COExpander sensitive to specific strategies for determining operations for cop, and are there any relevant ablation experiments?
2. Is COExpander sensitive to the parameters in Eq (5), and what is the design concept of Eq (5)?
3. Will you plan to publish the code implementation?

For Weakness1:

4. In Line 273, what does it mean that the model input contains optimal solution $x^*$.
5. In Algorithm 1, you mentioned Model Inference with M. The acquisition of M seems to require the optimal solution. How was this step implemented during testing?
6. Can you provide a simple and clear description of the method design and introduction?

**Relation To Broader Scientific Literature:**

1. This article proposes an Adaptive Expansion (AE) paradigm which bridges the global prediction (GP) and local construction (LC) paradigms through the utilization of a partial state prompted solution generator.
2. This paper presents a series of benchmarks, which is beneficial for future research.

**Theoretical Claims:**

There are no theoretical claims in this paper.

---

> ### Author Rebuttal · Authors · 2025-03-31
>
> Dear Reviewer F5eW,
>
> We are more than grateful for your time and recognition of our contributions. We sincerely present our point-to-point clarification to your questions below.
>
> > **Q0-Comparing with UDC.**
>
> **A0:** First, we have cited the UDC method and explained our exclusion of further comparison in Sec.1 (line 25-26, right). Second, we provide a brief performance comparison below, showing that COExpander generally outperform UDC (reported in original paper with the *best* settings) in terms of both optimality gap and per-instance efficiency.
> |Task|Gap(UDC,%↓)|Time(UDC,s↓)|Gap(Ours,%↓)|Time(Ours,s↓)|
> |-|-|-|-|-|
> |TSP500|1.58|1.88|**0.25**|**0.66**|
> |TSP1K|1.78|3.75|**0.64**|**2.43**|
> |TSP10K|4.03|60.00|**1.45**|**29.50**|
>
> > **Q1-Sensitivity to specific strategies for determining operations.**
>
> **A1:** In fact, to mitigate the overreliance on ad-hoc designs, the decisive operators in our work are based on the simplest greedy heuristics. Take MIS for instance, as detailed in Sec. 4.2.2, we sort the predicted heatmap in descending order and sequentially (greedily) attempt to add nodes to the independent set. Once the "independence" constraint is breached, the process halts and the next round of model inference commences. **Thus, the version we propose can essentially be regarded as a baseline strategy, while future efforts on more complex determining operations are encouraged and may necessitate ablations against ours.** We will also explicitly address this point in the paper to prevent misunderstandings. Thanks for your question!
>
> > **Q2-Sensitivity to the mask probability in Eq(5).**
>
> **A2:** Please refer to our **response A2 to Reviewer UKMH**, where the effects of $\alpha$ is systematically evaluated though supplementary experiments.
>
> > **Q3-Disclosure of the code implementation.**
>
> **A3:** Sure. Rest assured that the code and datasets of this work will be fully open-sourced as soon as the conference permits. At the moment, due to the regulation that "links may only be used for figures (including tables) and captions that describe the figure (no additional text)", we are more than willing to provide our code via anonymous links once we have your confirmation that it does not violate any of the rules.
>
> > **Q4-Explanation of model input containing optimal solution $x^{*}$.**
>
> **A4:** Recall the process of diffusion mechanism, in the training stage, $x^*$ serves as the starting point to generate the noising trajectory, i.e., $x_{0:T}=x_0,x_1,\cdots,x_T$ where $x_0=x^*$, and the noised representation $x_T$ is fed to the model to learn the denoising mapping. At inference, the noised vector $x_T$ is generated by random Gaussian noise without ground truth label. We note the description in Line 273 slightly inaccurate and have updated it accordingly to mark the difference of model input for training and testing phases. Thank you for the critical point!
>
> > **Q5-Acquisition of $M$ in the testing phase.**
>
> **A5:** First of all, we'd clarify that the mask $M$ is a Boolean matrix with the same shape as the solution $x$. Its purpose is straightforward: to mark whether an edge or node has been determined (assigned 0 or 1). Hence, $M$ is initialized as a zero matrix (vector) at the beginning of tests, and there is no requirement for the optimal solution $x^*$. We've updated our scripts in Algorithm 1 to explicitly show the initial setting of the mask.
>
> > **Q6-Request for a simple and clear description of the method.**
>
> **A6:** In this paper, we put forward a novel paradigm for the neural decisive process, aiming to harness the strengths of both global prediction and local construction methods. Briefly, the process of determining the solution for a problem instance is adaptively divided into multiple rounds. During each round, the model generates a heatmap that forecasts the probability of each node/edge being chosen. Subsequently, the solution is greedily constructed in descending order of the heatmap values until the constraints are breached. All the selected nodes/edges are then marked as 1 and, together with the partial solution, fed back into the model to initiate a new round of inference. This cycle continues, with more nodes or edges being determined in each successive round, until every decision variable has been taken into account and assigned a value of either 0 or 1 in the final solution. The modek is trained with randomly generated intermediate states (e.g. 10 out of 50 nodes have been decided in MIS) to learn better predictions for the solving stage.
>
> ---
> We hope our explanations adequately address your concerns. We are more than willing to provide further clarifications if any question remains, while we'd be rather grateful if, through your valuable reconsideration, our technical  contributions merit an improved assessment overall.
>
> Best regards,
>
> The Authors

---

### Official Review · Reviewer_CvKY · 2025-03-12

**Overall Recommendation:** 3

**Summary:**

The paper proposes a learning-based approach for solving combinatorial
optimization problems that combines global and local paradigms to achieve better
performance. The authors describe their method and evaluate it empirically.

**Claims And Evidence:**

The claim to have reimplemented a range of state-of-the-art solvers is not credible.

**Essential References Not Discussed:**

n/a

**Experimental Designs Or Analyses:**

Yes, as far as possible based on the description provided.

**Methods And Evaluation Criteria:**

Yes.

**Other Comments Or Suggestions:**

The approach is novel to the best of my knowledge and seems to work well in
practice. However, the paper is too long for a conference format in my opinion
(the appendices are 18 pages compared to 11 pages for the main paper including
references).

The authors claim to have reimplemented the solvers Gurobi, KaMIS, LKH, GA-EAX,
and Concorde -- really? Are you seriously claiming to have reimplemented all of
these solvers, including the closed-source Gurobi, that are the result of
decades of implementation efforts? How did you verify that your
reimplementations are faithful to the originals? Appendix D seems to suggest
that your reimplementations are in fact faster than the originals, which would
be a huge advance in itself. Please clarify this important point.

This issue unfortunately casts doubt on the empirical results, in particular the
improvements of the proposed method over the baselines. Similar to the classical
solvers, the neural approaches were retrained in at least some cases, but it is
unclear whether the training process reflects what was used to train those
approaches originally. Thus, it is unclear to what extent improvements are due
to the proposed approach. In addition, a custom set of benchmarks is used,
making comparison with published figures difficult.

**Other Strengths And Weaknesses:**

Potentially a very novel way to approach CO.

**Questions For Authors:**

Please clarify what you implemented and where you used existing implementations.

**Relation To Broader Scientific Literature:**

Adequate.

**Theoretical Claims:**

n/a

---

> ### Author Rebuttal · Authors · 2025-03-31
>
> Dear Reviewer CvKY,
>
> We appreciate your time reviewing and acknowledging the novelty and empirical efforts of our work. There appear to be several misunderstandings, chiefly concerning the claim about the "reimplementation" of baseline solvers and existing neural methods. We offer point-by-point clarifications below.
>
> ---
> > **Q1-About paper length.**
>
> **A1:** Thank you for your concern. Admittedly, our paper has a relatively long appendix. However, we'd like to explain that: First, each part of the appendix provides supplementary support for the core idea. It includes additional related work for a broader discussion of existing NCO efforts (A), detailes of the 3 paradigms in our proposed taxonomy (B), formal problem definitions (C), introduction of our re-standardized data and re-wrapped tools (D&E), implementation details for reproduction (F), and supplementary results and analysis (G). Second, as far as we know, many works [1,2,3, etc.] have comprehensive content without their contributions being devalued by top conferences, and we've double-checked to ensure no violation of the submission rules. Thus, we believe our paper contains the necessary context to present a complete picture of our proposal, with both narrative and procedural justifications, comparable to precedents of the same tier and area.
>
> ---
> > **Q2-Doubts on the claim of reimplementation.**
> >
> **A2:** Thank you for posing this significant question. **Firstly**, we sincerely apologize for any misunderstanding. In fact, we are not "re-implementing" the renowned solvers, rather, our work center on enhancing the usability of invoking them through more unified and efficient APIs for future research. This process is more aptly described as re-"wrapping"/"encapsulating" at the engineering level, without duplicating the underlying algorithms. Hence, the issue of “faithfulness” to the original implementations does not pertain here. Below we provide a sample code how the solvers are accessed within our framework:
> ```
> solver = MISGurobiSolver()
> solver.from_txt("path/to/data")
> solver.solve()
> >>> (obj: xx, gt_obj: xx, gap: xx, std: xx)
> ```
> We believe our efforts enables an easy and consistent use of these solvers for future research.
>
> **Regarding the improvement of LKH**, as elucidated in Appendix D.2, it involves an in-depth exploration of the complex built-in settings of the LKH package. We found that the LKH3 package utilizes a parameter named `SPECIAL`, which by default disables the use of 2-/3-opt operators, thus compromising the solving quality on TSP. So, we empirically removed it for enhancing both performance and efficiency, and leave it an open question prompting further investigation into its internal mechanisms.
>
> **In summary**, we regret any confusion caused and hope our clarifications accurately convey the nature of this part in our work—aiming for improved convenience and potential performance gains from an engineering perspective. We'd like to reiterate our primary technical contribution: the proposal of a novel neural paradigm and framework for COPs, supported by comprehensive experiments.
>
> ---
>
> > **Q3-Doubts on retraining of existing neural solvers.**
>
> **A3:** Thanks for your comments. Firstly, in the field of NCO research, it is a common and encouraged practice to retrain and re-evaluate existing methods, which facilitates a fairer comparison, as directly using results from previous works can lead to discrepancies of hardware, datasets, and testing pipelines. Retraining demands significant effort. **Regarding the data, our benchmark datasets have covered the vast majority of publicly available datasets, such as TSP(Uniform), MIS(RB-SMALL) and MIS(ER-700-800)**. In cases where consistent data is currently lacking, like for ATSP, MCl, MVC, MCut, etc., we synthesize new instances **following the same generating algorithm as previous works**. Our intention is to help establish standardized benchmarks for these problems, similar to the well-established TSP benchmark. It's worth noting that the results obtained from our retrained neural approaches generally match or even surpass those reported in the original papers (see the table below). Rest assured that we shall release the code and data as soon as the conference permits via anonymous links, in any case.
>
> |Data|Method|Ori. Gap(%)|ReImpl. Gap(%)|
> |-|-|-|-|
> |MIS-SMALL|DIFUCO|4.29|3.46|
> |MVC-SMALL|Meta-EGN|2.80|1.56|
> |MCl-SMALL|Meta-EGN|7.00|8.30|
> |MCut-SMALL|DIFUCO|0.48|0.06|
> |MCut-LARGE|DIFUCO|-0.11|-1.77|
> |TSP-100|GCN|1.39|0.26|
> |TSP-100|SYM-NCO|2.88|1.75|
> |ATSP-100|MatNet|3.24|3.26|
>
> ---
> ## References
>
> [1] UDC, NeurIPS 2024 (45 pages)
>
> [2] UniCO, ICLR 2025 (41 pages)
>
> [3] UCOM2, ICML 2024 (34 pages)
>
> ---
> We hope our clarifications adequately address your concerns. We'd be rather grateful if, through your meticulous reconsideration, stronger consensus could be reached on a more favorable assessment of our work. We remain positive to further discussions!
>
> Best regards,
>
> The Authors

---

> > ### Comment · Reviewer_CvKY · 2025-04-01
> >
> > Thank you for the clarification -- this addresses my main concern and I will raise my score accordingly.

---

### Official Review · Reviewer_zhdB · 2025-03-13

**Overall Recommendation:** 3

**Summary:**

This paper introduces an adaptive approach for combinatorial optimization, where solution variables are incrementally determined using a dynamically adjusted decision step-size rather than fixed-size decisions. The method integrates global prediction (GP), producing probabilities for all variables at once, and adaptively expands partial solutions in each step. The authors demonstrate the effectiveness of their approach across several benchmarks, showing clear performance gains.

However, the core idea appears closely related to the previously introduced 'Learning What to Defer (LwD)' framework [1], which also adaptively adjusts the number of decision steps through deferred actions. To clearly establish the novelty of this paper, a direct experimental comparison against LwD is necessary. Without such comparisons, the originality of the proposed method remains uncertain. For this reason, I currently give a weak rejection and recommend that the authors provide explicit comparisons to related adaptive decision-making methods to better highlight their contribution.

[1] Ahn et al., "Learning What to Defer for Maximum Independent Sets", ICML 2020.

---
**After Rebuttal**: I increased score because authors' rebuttal addressed my concerns.

**Claims And Evidence:**

Their claim that the GP method is not expressive enough and the LC method is not very efficient is valid. Adaptive decision-making appears to be a promising alternative that addresses these limitations.

**Essential References Not Discussed:**

One critical missing reference is [1], which presents an almost identical claim and proposes an adaptive decision-making scheme that automatically stretches or shrinks decision steps using deferred actions. A direct comparison with this work is necessary, along with a clear discussion of the differences and contributions.

**[1]** Ahn et al., *"Learning What to Defer for Maximum Independent Sets,"* ICML 2020.

Additionally, for **partial solution generation**, you may refer to prior work related to GLOP:

- Kim et al., *"Learning Collaborative Policies to Solve NP-hard Routing Problems,"* NeurIPS 2021 (which introduces a "reviser" to refine partial solutions).

For **LC solvers**, you may also include the following work in the related works section, as it extends DeepACO:

- Kim et al., *"Ant Colony Sampling with GFlowNets for Combinatorial Optimization,"* AISTATS 2025.

**Experimental Designs Or Analyses:**

The experimental design is reasonable, covering several benchmarks and large-scale cases.

**Methods And Evaluation Criteria:**

Their evaluation criteria are reasonable and clearly justified.

**Other Comments Or Suggestions:**

None

**Other Strengths And Weaknesses:**

**Weaknesses:** They only verify their method on simple constrained combinatorial optimization problems. Its effectiveness on more complex constrained problems, such as CVRP, PDP, and JSSP, remains uncertain, especially since hard constraints in these problems are more naturally handled by the LC method.

**Questions For Authors:**

I have one critical question: Can this type of method effectively handle hard-constrained problems? As far as I know, such methods perform well on locally decomposable problems like MIS and simple routing problems such as TSP. However, I have yet to see a non-LC method successfully solve complex routing problems, even relatively simple ones like CVRP, which involve multiple hard constraints. I would like to hear your discussion on this issue and how your algorithm might address these challenges.

**Relation To Broader Scientific Literature:**

This strategy is related to decision-making methods in LLMs. Adaptive decision-making could potentially be useful in that context as well.

**Theoretical Claims:**

No theory here.

---

> ### Author Rebuttal · Authors · 2025-03-31
>
> Dear reviewer zhdB,
>
> We appreciate your meticulous review and constructive advice. Below we respond to the 3 major points mentioned in your comments.
>
> > **Q1: Comparison with Lwd.**
>
> **A1:** **Firstly**, it was our oversight not to compare LwD initially. Indeed, LwD is a highly relevant counterpart within the AE paradigm employing the RL learning strategy, and it merits a more in-depth discussion in our paper for its gradual "determination" of decision variables. **However**, it’s important to note that while LwD focuses predominantly on problems that are "locally decomposable" through RL-based policy learning, our COExpander represents a significant extension. It is designed to handle a broader spectrum of tasks, spanning node-selection tasks like MIS (isolated constraints), edge-selection tasks such as TSP (simple global constraints), and even more complex CVRP (detailed in Q3). This enhanced capability is underpinned by the utilization of more advanced backbone model architectures, specifically diffusion and consistency models, in combination with SL learning. We list the results comparing LwD and COExpander in [ae_solvers.png](https://anonymous.4open.science/r/COExpander-ICML-Rebuttal-0246/ae_solvers.png).
>
> **To summarise, we are more than delighted to have incorporated your recommended work into our discussion and experimental comparisons. Our collective efforts, we believe, will contribute to a more systematic establishment of the AE paradigm in the context of NCO problem-solving.**
>
> > **Q2: Suggestions on additional related works (LCP & GFACS).**
>
> **A2:** Thanks for your suggestion. We have carefully studied the works you provided and agree that LCP introduces a reviser mechanism for refining partial solutions and GFACS provides valuable insights into advanced sampling techniques for LC solvers, which are both significant approaches in the NCO community. Therefore, we have followed your advice to cite them in our related work to enhance the paper’s connection to the broader NCO research landscape.
>
> > **Q3: Applicability to more complex problems like CVRP.**
>
> **A3:** Firstly, COExpander can be directly applied to CVRP, as presented in [cvrp.png](https://anonymous.4open.science/r/COExpander-ICML-Rebuttal-0246/cvrp.png), which is the first attempt to demonstrate applicability of heatmap-based methods on VRPs, to our knowledge. To admit, there is limitation of SL methods to tackle complex constraints. Inspired by LwD, we have theoretically conceived to extend the AE paradigm to solve complex problems like VRPs via PPO.
>
> **(i) Overview.** AE paradigm is an iterative process and satisfies the Markov property, thus can be modeled as MDP. We plan to use PPO for model training, in order to better adapt to the relationship between actions and constraints. **Note that a feasible scheme is theoretically described as follows, while the implementation and empirical results have obviously exceeded the scope of this paper, hence reasonably remitted to future researches.**
>
> **(ii) Modeling.** state $s_t:=(x_t,M_t)$; action $a_t:=(m_t,z_t)$ (selection and assignment); transition $P(s_{t+1}|s_t,a_t)$ devided to $x_{t+1}^{i} = \begin{cases}z_t^i&\text{if }m_t^i=1 \\\ x_t^i&\text{else }\end{cases}$ and $M_{t+1}=M_t\lor m_t$; reward is only defined in the final step $R(s_t,a_t) := \begin{cases} -c(G, x_t) & \text{if } t=k-1,\\\ 0 & \text{else} \end{cases}$
>
> **(iii) Training Model with PPO.** Given the model $f(\cdot, \cdot)$, $\pi_\theta^t := \pi_\theta(a_t|s_t, G) = f(G, s_t)$. $z_t$ of $a_t$ is sampled from $\pi_\theta^t$ via **Bernoulli sampling** and then set the determined variables as 0. The advantage function: $\hat{A}\_t= -\gamma^{k-1-t} \cdot c(G,x_{k-1})-\mathbb{E}\_\theta [ -c(G,x_{k-1} | s_t)]$.
>
> **(iv) The Selection Part of Action.** The selection part $m$ of $a$ is the core of controlling constraint satisfaction. Take CVRP as an example.
> - Reshape $m$, $z$, and $M$ to $n\times n$, where $n$ is the number of nodes including the depot. $z$ needs to undergo an OR operation with its transpose.
> - Make pre-assignment: $\hat{x}\_{i,j}=\begin{cases}z_{i,j}&\text{if }M_{i,j}=0, \\\ x_{i,j}&\text{if }M_{i,j}=1\end{cases}$.
> - Check Constraints: $C_i=\begin{cases}1&\text{if }\sum_{j=1}^{n}\hat{x}\_{i, j}>2\text{ or }\sum_{j=1}^{n}p_{i,j}\cdot d_j>1 \\\ C_i&\text{else}\end{cases}$
> - Re-Assignment. $\hat{z}\_{i,j}=\begin{cases}1&\text{if }C_i=1,\\\ z_{i,j}& \text{else}\end{cases}$; $\hat{x}\_{i,j} = \begin{cases}\hat{z}\_{i,j}&\text{if }M_{i,j}=0,\\\ x_{i,j}&\text{else}\end{cases}$.
> - Selection Part. $m_{i,j}=\begin{cases}1&\text{if }\hat{z}\_{i,j}=1\text{ or } \sum_{j=1}^{n}\hat{x}\_{i,j}=2,\\\ 0&\text{else}\end{cases}$
>
> ---
> We hope our point-by-point clarificaitons, supplementary experiments, and positive feedbacks on your requests, have satisfactorily addressed your concerns. We remain fully committed to involving further discussions with you towards an elevated evaluation of our work.
>
> Best regards,
>
> The Authors

---

### Official Review · Reviewer_UKMH · 2025-03-13

**Overall Recommendation:** 3

**Summary:**

This paper introduces COExpander, a method to solve combinatorial optimization problems by diffusion models. Unlike previous neural methods, COExpander is informed by local partial solutions and iteratively improves upon them to obtain a final solution. The authors study 6 graph CO problems and demonstrate good performance in terms of speed and drop compared to SOTA neural approaches.

**Claims And Evidence:**

The paper performs extensive experiments in several graph CO datasets and the evidence is convincing in terms of drop.

**Essential References Not Discussed:**

At least one essential reference is not discussed, iSCO [1]. This paper proposed a sampling method for solving CO problems including most of the ones studied in this paper. iSCO outperforms KaMIS: in ER [700-800], while the proposed COExpander has a more than 5% gap compared to KaMIS, meaning that COExpander should be worse than this paper.

[1] Sun, H., Goshvadi, K., Nova, A., Schuurmans, D., & Dai, H. (2023, July). Revisiting sampling for combinatorial optimization. In International Conference on Machine Learning (pp. 32859-32874). PMLR

**Experimental Designs Or Analyses:**

Yes, I checked the main text. They are overall sound, although some comparisons were not made (see below).

**Methods And Evaluation Criteria:**

Yes

**Other Comments Or Suggestions:**

Minor points:

1. About the claim “To address this issue, we abandoned node features and adopted convolutions only on edges. To our best knowledge, this is the first global predictor with good performance for ATSP.”, GLOP (Ye et al. 2024b) already implemented such network.

2. The release code and datasets are a good contribution. However, at the moment, it is just a promise. I would invite authors to consider submitting the code through a zip file or an anonymized link in the future.

**Other Strengths And Weaknesses:**

I think the paper is original overall and a good contribution to NCO. Also, the authors made several contributions on the dataset side, including fixing solvers, which is appreciated.

In terms of weaknesses:

1. I believe the main limitation of this approach is the applicability to CO problems with mostly a single constraint. Similarly to other previous approaches, such as DIFUSCO, I do not see how the approach could benefit more practical problems such as vehicle routing problems and job shop scheduling.

2. This approach is supervised and requires optimal solutions $x^\*$ unlike RL, unsupervised learning, or sampling-based approaches as iSCO. This may not be possible for more complex problems which either do not have or whose optimal solutions are hard to obtain.

3. Writing: I found some parts confusing in the flow. For example: in abstract and intro, (self-) “adaptive/adapted step-sizes” is repeated multiple times, but in the main text (Section 4) this is not clarified. I am still unsure about the meaning (see question).

**Questions For Authors:**

1. Which part does the “adaptive” in your method stand for exactly? I could not find a single answer to this, as the term is mentioned in the abstract and introduction but not in Section 4. My understanding is that this is the number of determination steps, etc, from the ablation studies, but this is not adaptive and decided a priori. Related to this, what does the "self" in "self-adaptive" stand for?

2. How did you choose the $0.9$ in equation 5?

3. What is the meaning of encoding $y_t$ with a sinusoidal embedding layer? Are the nodes numbered in any way?

4. In Figure 3, why can more inference steps worsen performance in some problems?

5. You mentioned solving more general VRPs in future works. How would that work, given that, in my understanding, in such problems the mask may change values multiple times depending on the current tour?

**Relation To Broader Scientific Literature:**

The paper is related to the NCO literature on classical graph CO problems, i.e., problems on the more theoretical side with one constraint. The paper is an improvement over the previous Fast T2T method with some additional iterations for partial solution refinement.

**Theoretical Claims:**

No theoretical claim is present.

---

> ### Author Rebuttal · Authors · 2025-03-31
>
> Dear Reviewer UKMH,
>
> Thanks for your recognition and valuable questions.
>
> >**Q1: Interpretation of "adaptive"**
>
> **A1:** In our work, "adaptive" means exactly the fact that *the number of ascertained decision variables within one round of determination is not fixed*. Specifically, it is achieved via the autonomy of the determiantion operator (see Sec.4.2), where one determination round usually ends when the greedy operator yields invalid variable assignments, so the number of decisions per round is random. Note that $D_s$, which caused your confusion, is just a maximum limit for the **rounds**. E.g., $D_s=3$ for a task with 10 variables ensures the problem gets solved within 3 rounds, but **no prior is imposed on how the quantity of decisions is distributed across those rounds** (possibly [2,3,5] or [4,3,3]). For the term "self-", we have unified our wording to "adaptive" to describe such flexibility to avoid abuse of concepts.
>
> >**Q2: Choice of $\alpha=0.9$ in Eq.5.**
>
> **A2:** First, the model convergence and solving quality are not sensitive to $\alpha$. Second, $\alpha=0.9$ is preferred to ensure the model to be sufficiently (Pr=0.9) trained on cases to solve from scratch, while balancing the training diversity for solving from arbitrary intermediate states (Pr=0.1). Empirical results and detailed analysis are illustrated in the table and figures in [alpha_ablation.png](https://anonymous.4open.science/r/COExpander-ICML-Rebuttal-0246/alpha_ablation.png).
>
> >**Q3: Sinusoidal embedding layer**
>
> **A3:** **Generally**, we follow previous works (GCN4TSP,DIFUSCO,T2T,etc) to employ the sinusoidal embedding. It functions in a way similar to how position encoding works for Transformers. (Exactly as you speculated, it assigns numerical identifiers to nodes and edges for GCN.) **Specifically**, for node-tasks (e.g. MIS), the input graph contains only edge index ($E\in R^{|\mathcal{E}|}$), yet the (noised) solution vector pertains to nodes, i.e. $y_t\in [0,1]^{|\mathcal{V}|}$. Parallelly, for edge-tasks (e.g. TSP), only node coordinates ($V\in R^{|\mathcal{V}|\times2}$) are provided, while the solution are on the edges, i.e. $y_t\in [0,1]^{|\mathcal{E}|}$. Hence, the design rationale is to make up for the lacking part of each task type, i.e. dual initial embeddings for both nodes and edges to feed GCN in either case. Formally, $h_v=S(V),h_e=S(y_t)$ for edge-tasks; $h_v=S(y_t),h_e=S(E)$ for node-tasks. We hope this explanation resolves your confusion, and we'll update our paper with clearer notations.
>
> >**Q4: $I_s$-performance drop in Fig.3**
>
> **A4**: Unlike the multi-sampling or repeated random re-solving tricks, in our case, each inference step marks distinct time points in the denoising process, and the solution vector being denoised is utilized iteratively throughout the entire process. So, a larger $I_s$ merely implies a finer-grained denoising schedule towards the final solution ($\hat{x_0}$). Intuitively, Fig.3 showns an upward-trended solving quality as $I_s$ grows, however, **there is no hard guarantee of performance gain with greater $I_s$, esp. when $I_s$ is small (e.g. 1-5), since the inferences are sequentially linked rather than parallelly independent for "best-picking".**
>
> >**Q5: Applicability to VRP**
>
> **A5:** Please refer to **A3 to Reviewer zhdB**, due to the space limits.
>
> >**Q6: Claim of "the first good global predictor for ATSP".**
>
> **A6:** It appears to be a minor misunderstanding. GLOP is indeed a good work, though, as stated in Sec.6.2 of their paper, "we apply MatNet50 checkpoint as our reviser without retraining", implying that MatNet (NIPS21), a solver of the **local construction** (LC) type, was used as the backbone. Thus, it doesn't contradict our claim to be the first **global predictor** (GP) with good performance on ATSP. In any case, we've revised our paper with more careful wordings.
>
> > **Q7: Code release**
>
> **A7:** The regulations seem to emphasize that only figures are allowed in anonymous links during rebuttal. So we are more than willing to provide our code yet awaiting your confirmation that it does not violate any of the rules.
>
> > **Q8: Discussion of iSCO.**
>
> **A8:** We've noticed recent sampling methods like iSCO and RLSA(arXiv:2502.00277) and added them in our discussion of related work. Actually, they can be orthogonally performed on top of COExpander as a strong post-processor, where neural models serve to quickly yield high-quality initial solutions in aid of the infeasibility issue that sampling methods often encounter. Supplementary experiments on MIS (shown in the [rlsa.png](https://anonymous.4open.science/r/COExpander-ICML-Rebuttal-0246/rlsa.png)) validate the synergy of COExpander+RLSA.
>
> ---
> Hopefully our responses alleviate your concerns with satisfaction, and we stay open for any further interactions!
>
> Best regards,
>
> The Authors

---

### Decision · Program_Chairs · 2025-05-01

**Decision:**

Accept (poster)

**Comment:**

The authors provided an excellent rebuttal and effectively addressed the reviewers' concerns. This work combines global prediction with local construction methods to tackle various combinatorial optimization problems. While its applicability to more complex problems warrants further investigation, the authors presented preliminary results on CVRP during the rebuttal and demonstrated the potential of their approach. The more labeling effort required for supervised learning compared to reinforcement learning methods such as UDC could be better clarified in the updated version. All reviewers lean toward accepting this work.